# Provable Bounds for the Learnability of
# Sample-Compressible Families from Noisy Samples

**Arefe Boushehrian** [1]   **Amir Najafi** [2]

## Abstract

Learning distribution families over $\mathbb{R}^d$ is a fundamental problem in unsupervised learning and statistics. A central question in this setting is whether a given family of distributions possesses sufficient structure to be (at least) information-theoretically learnable and, if so, to characterize its sample complexity. In 2018, Ashtiani et al. (2018) reformulated sample compressibility as a structural property of distribution classes, proving that it guarantees PAC-learnability. This discovery subsequently enabled a series of recent advancements in deriving nearly tight sample complexity bounds for various high-dimensional open problems. It has been further conjectured that the converse also holds: every learnable class admits a sample compression scheme, making the two notions to be equivalent. In this work, we establish that sample compressible families remain learnable even from perturbed samples, subject to a set of minimax-necessary and sufficient conditions. In particular, we assume samples are corrupted by an additive independent noise model, and theoretically derive sample complexity bounds for general sample compressible classes in arbitrary dimensions with respect to both $\ell_2$-norm and total variation distance.

## 1. Introduction

Learning parametric families of probability distributions on $\mathbb{R}^d$ lies at the foundation of unsupervised learning and statistical inference. A fundamental question is whether a given family $\mathcal{F}$ is learnable and, if so, what sample size is required. Learnability is commonly studied within the Prob-

ably Approximately Correct (PAC) framework, in which the objective is to approximate an unknown distribution $g$ by elements of $\mathcal{F}$ to within discrepancy $\epsilon$ with probability at least $1 - \delta$, using a number of samples that scales polynomially in $\epsilon^{-1}$, $\delta^{-1}$, and a suitable complexity measure of the class $\mathcal{F}$ (Valiant, 1984; Devroye & Lugosi, 2001). Various discrepancy measures are used in the literature, including total variation (TV) distance (Devroye et al., 2018), Minkowski metrics (Devroye & Lugosi, 2001), and Wasserstein distances (Niles-Weed & Berthet, 2022). In this work, we focus exclusively on TV distance. Formal definitions of PAC learnability are deferred to Appendix A. It is worth noting that a significant line of work in this area focuses on *efficient* PAC learnability, in which the goal is to obtain learning guarantees with algorithms running in polynomial time. In contrast, the present paper is concerned with *information-theoretic* learnability, where no computational constraints are imposed and procedures are allowed to run in exponential time.

A recurring theme in this literature is the identification of structural conditions that control the complexity of $\mathcal{F}$ and thereby enable learnability. One such condition is *sample compressibility*, introduced by Littlestone and Warmuth (Littlestone & Warmuth, 1986) in the supervised setting and adapted to distribution learning by Ashtiani et al. (Ashtiani et al., 2018) (see Definition A.3). Informally, sample compressibility requires that for any $f \in \mathcal{F}$ and sufficiently large $n$, there exists an encoder which, with high probability, compresses $n$ i.i.d. samples drawn from $f$ into a compact representation consisting of a subset of $\tau$ samples together with $t$ additional bits, where typically $\tau$ and $t$ are $\widetilde{\mathcal{O}}(1)$ in $\epsilon$ and $\delta$. A decoder using only this compressed representation must reconstruct an estimator achieving PAC accuracy. The notion extends naturally to the agnostic setting (with $f \notin \mathcal{F}$), under the term *robust* sample compressibility. Sample compressibility has strong theoretical implications: it guarantees PAC learnability and often yields near-optimal sample complexity bounds for a broad class of distribution families (Ashtiani et al., 2018; Ben-David et al., 2023; Saberi et al., 2023).

In this paper, we study whether these guarantees persist when learning from *noisy samples*. Despite extensive work

[1]College of Management of Technology, EPFL, Lausanne, Switzerland (Email: arefe.boushehrian@epfl.ch) [2]Department of Computer Engineering, Sharif University of Technology, Tehran, Iran. Correspondence to: Amir Najafi <amir.najafi@sharif.edu>.

*Proceedings of the 43$^{rd}$ International Conference on Machine Learning*, Seoul, South Korea. PMLR 306, 2026. Copyright 2026 by the author(s).

on learning under noise in specific models, to the best of our knowledge, no general framework currently exists for extending learnability guarantees of an arbitrary distribution family to stochastic perturbation settings. Existing results are largely problem-specific and do not offer structural conditions under which learnability is preserved (see, e.g., (Tang & Yang, 2023; Saberi et al., 2023), and Appendix B for a comprehensive review of prior works). We show that such an extension is possible for broad classes of sample-compressible distributions. Concretely, we consider an additive noise model and identify a small set of natural assumptions (sufficient and (minimax) necessary) under which sample-compressible families remain PAC learnable from corrupted samples. Our results provide explicit sample complexity bounds that can be viewed as inflated analogues of their noise-free counterparts, where the inflation depends on the noise distribution and the compression parameters of $\mathcal{F}$. We further show that these assumptions are minimax necessary in a precise sense. Unlike fully nonparametric approaches, our bounds explicitly exploit the structural simplicity of parametric models through their compression properties, yielding rates that are substantially tighter than those obtained from general-purpose analyses.

However, establishing matching lower bounds remains an open direction. Even in the noise-free regime, no general framework currently exists for proving tight minimax lower bounds under sample compressibility alone. We conjecture that deriving such bounds would require additional problem-specific assumptions, as sample compressibility by itself does not appear sufficiently restrictive to yield matching minimax guarantees.

## 1.1. Notations

For $n \in \mathbb{N}$, we show the set $\{1, \dots, n\}$ via $[n]$. Assume a measurable space $\mathcal{X}$ and a corresponding $\sigma$-algebra $\mathcal{B}$. We usually assume $\mathcal{X} \subseteq \mathbb{R}^d$. For $p \geq 1$, two probability measures $P_1, P_2$ supported over $\mathcal{X}$ with respective Lebesgue density functions $f_1, f_2 \in L^p(\mathcal{X})$, the $\ell_p$-distance is defined as $\|f_1 - f_2\|_p \triangleq \left( \int_{\mathcal{X}} |f_1(\boldsymbol{x}) - f_2(\boldsymbol{x})|^p \, \mathrm{d}\boldsymbol{x} \right)^{\frac{1}{p}}$. In this work, we only use $p = 1, 2$. Moreover, the Total Variation (TV) distance is defined as $\mathsf{TV}(f_1, f_2) \triangleq \sup_{B \in \mathcal{B}} |P_1(B) - P_2(B)| = \frac{1}{2}\|f_1 - f_2\|_1$. For a function $g : \mathbb{R}^d \to \mathbb{R}$ with $g \in L^2(\mathbb{R}^d)$, its *Fourier* transform $\mathsf{F}\{g(\boldsymbol{x})\}(\boldsymbol{w})$ for $\boldsymbol{w} \in \mathbb{R}^d$ is defined as follows:

$$\mathsf{F}\{g(\boldsymbol{x})\}(\boldsymbol{w}) = G(\boldsymbol{w}) \triangleq \int_{\mathbb{R}^d} g(\boldsymbol{x})e^{-i\boldsymbol{x}\cdot\boldsymbol{w}}\mathrm{d}\boldsymbol{x}, \quad (1)$$

where we usually refer to $\boldsymbol{w}$ as the *frequency vector*.

## 1.2. Sample Compression (Informal)

Let $\mathcal{X} \subseteq \mathbb{R}^d$ for some $d \in \mathbb{N}$, and let $\mathcal{F}$ be a class of distributions supported on $\mathcal{X}$. A *decoder* for $\mathcal{F}$ is a deterministic

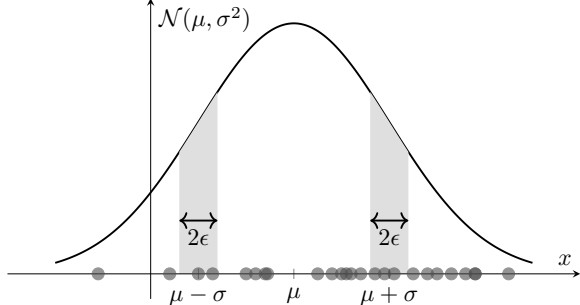

*Figure 1.* Depiction of the $(2, 0, \mathcal{O}(\frac{1}{\epsilon} \log \frac{1}{\epsilon}))$-s.c. scheme for $\mathcal{N}(\mu, \sigma^2)$. Given $\widetilde{\mathcal{O}}(\frac{1}{\epsilon})$ i.i.d. samples (shown as grey dots), with high probability, there exist two distinct samples, $X_i$ and $X_j$, falling within $(\sigma\epsilon)$-neighborhoods of $\mu - \sigma$ and $\mu + \sigma$, respectively. A simple decoder $\mathcal{N}\left((X_i + X_j)/2, (X_j - X_i)^2/4\right)$ then reconstructs an estimate of the distribution with TV error $\mathcal{O}(\epsilon)$.

function $\mathcal{J}$ that takes a finite sequence of elements of $\mathcal{X}$ and a finite sequence of bits as input, and outputs an element of $\mathcal{F}$. Specifically, $\mathcal{J} : \bigcup_{i=0}^{\infty} \mathcal{X}^i \times \bigcup_{i=0}^{\infty} \{0,1\}^i \to \mathcal{F}$.

**Definition 1.1** (Sample Compression). Let $\tau, t, m : (0,1) \to \mathbb{Z}_{\geq 0}$ be functions. A class $\mathcal{F}$ is said to admit $(\tau, t, m)$-sample compression (s.c.) if there exists a decoder $\mathcal{J}$ for $\mathcal{F}$ such that for any distribution $f \in \mathcal{F}$, the following holds: For any $\epsilon, \delta \in (0,1)$, if an i.i.d. sample set $S$ of size $n \geq m(\epsilon) \log(\frac{1}{\delta})$ is drawn from $f$, then with probability at least $1 - \delta$, there exists a sequence $\mathbf{L}$ of at most $\tau(\epsilon)$ elements of $S$ and a sequence $\mathbf{B}$ of at most $t(\epsilon)$ bits, such that $\|\mathcal{J}(\mathbf{L}, \mathbf{B}) - f\|_{\mathrm{TV}} \leq \epsilon$.

For example, the class of Gaussian distributions $\mathcal{F} = \left\{ \mathcal{N}(\mu, \sigma^2) \mid \mu \in \mathbb{R}, \sigma > 0 \right\}$ admits a $(2, 0, \mathcal{O}(\frac{1}{\epsilon} \log \frac{1}{\epsilon}))$-s.c. for any $\epsilon, \delta \in (0,1)$ (see Figure 1). Specifically, given $\mathcal{O}(\frac{1}{\epsilon} \log \frac{1}{\epsilon} \log \frac{1}{\delta})$ i.i.d. samples from any $f \in \mathcal{F}$ (with mean $\mu$ and standard deviation $\sigma$), one can guarantee that with probability at least $1 - \delta$, two distinct samples will fall within an $(\sigma\epsilon)$-neighborhood of both $\mu - \sigma$ and $\mu + \sigma$. This results in a simple scheme (see Figure 2) that $\epsilon$-estimates $f^*$ in TV error. An interesting feature of sample compression is that we do not need to explicitly identify which samples, when fed to the decoder, recover $f$. The key requirement is that the sequences of $\tau(\epsilon)$ samples and $t(\epsilon)$ bits only *exist*, without needing to specify an algorithm for finding them.

Many naturally occurring distribution families are known to admit *efficient* sample compression schemes. In a weaker sense, any PAC-learnable class of distributions is sample-compressible by at least a naive scheme. Conversely, (Ashtiani et al., 2018) demonstrated that sample compressibility guarantees PAC-learnability.

**Theorem 1.2** (Theorem 3.5 of (Ashtiani et al., 2018) (informal)). *Suppose $\mathcal{F}$ admits $(\tau, t, m)$-sample compression for some functions $\tau, t, m$. Then, there exists a deterministic*

*algorithm that, for any $\epsilon \in (0, 1)$, given at least*

$$n \geq N_{\tau,t,m}^{\mathsf{Clean}}(\epsilon, \delta) \triangleq \widetilde{\mathcal{O}}\left( m(\epsilon/6) + \frac{\tau(\epsilon/6) + t(\epsilon/6)}{\epsilon^2} \right) \quad (2)$$

*i.i.d. samples from any unknown distribution $g \in \mathcal{F}$, outputs $\widehat{f} \in \mathcal{F}$ such that $\|\widehat{f} - g\|_{\mathrm{TV}} \leq \epsilon$ with high probability. A similar bound extends to the agnostic case.*

Theorem 1.2 establishes the (information-theoretic) PAC learnability of $\mathcal{F}$ under the existence of a sample compression scheme. The complete version, including all polylogarithmic factors, is stated in Theorem C.3. However, as noted earlier, the sample-complexity bound in (2) may not be tight when compared with other approaches for proving PAC learnability. A conjecture (see Conjecture A.6) suggests that for every learnable class, there exists a tight/matching sample compression scheme and a corresponding sample complexity via Theorem 1.2. If true, this conjecture would imply that *every* information-theoretically learnable class can, without loss of generality, be characterized through an appropriate sample compression scheme. Formal definitions and guarantees are provided in Appendix A.3.

### 1.3. Formal Problem Definition

Consider a class of probability distributions $\mathcal{F}$ supported over $\mathcal{X} \subseteq \mathbb{R}^d$ for a given dimension $d \in \mathbb{N}$. For simplicity, we assume that all members of $\mathcal{F}$ admit densities. Consequently, without loss of generality, we interpret $f \in \mathcal{F}$ as a probability density function.

**Assumption 1.3** (Sample Compressibility). We assume that $\mathcal{F}$ admits a $(\tau(\epsilon), t(\epsilon), m(\epsilon))$ sample compression scheme for some $\tau, t, m : (0, 1) \to \mathbb{Z}_{\geq 0}$.

Let $G$ be a symmetric $d$-dimensional noise distribution with i.i.d. components, such as (but not limited to) a Gaussian distribution $\mathcal{N}(\mathbf{0}, \sigma^2 \mathbf{I}_d)$ for some $\sigma > 0$. The assumptions of symmetry and independence across dimensions are made for simplicity in the final formulations and can be relaxed. Given an unknown distribution $f^*$ (with $f^* \in \mathcal{F}$ in the realizable setting, but general for the agnostic case), let $\mathbf{X}_1, \ldots, \mathbf{X}_n \overset{i.i.d.}{\sim} f^*$. Our goal is to approximate $f^*$ given $\mathcal{F}$ and a set of *perturbed* samples obtained as $\widetilde{\mathbf{X}}_i = \mathbf{X}_i + \boldsymbol{\zeta}_i$, $\forall i \in [n]$, where $\boldsymbol{\zeta}_{1:n}$ are noise vectors, i.i.d. samples from $G$, independent of the original samples $\mathbf{X}_{1:n}$. Equivalently, one can assume $\widetilde{\mathbf{X}}_1, \ldots, \widetilde{\mathbf{X}}_n$ as i.i.d. samples from $f^* \star G$, where $\star$ denotes the convolution operator.

### 1.4. Summary of Results

Our main Theorem 2.10 shows that under mild conditions (Assumption 2.1), the class $\mathcal{F} \star G$ (i.e., the class of distributions $f \star G$ with $f \in \mathcal{F}$) is PAC-learnable from noisy samples within $\ell_2$-error $\epsilon > 0$, and with probability at least

$1 - \delta$, provided that

$$n \geq N_{\mathrm{clean}}\left( \frac{\epsilon}{4}, \frac{\delta}{2} \right) + \widetilde{\mathcal{O}}\left( \frac{d\tau}{\epsilon^2} \right) \cdot \log(1 + \sigma), \quad (3)$$

where $N_{\mathrm{clean}}(\epsilon, \delta)$ denotes the sample complexity of learning $\mathcal{F}$ from *clean* (noise-free) samples. Furthermore, we show that $\mathcal{F}$ itself is also PAC-learnable under noise with a constant factor of the sample complexity in (3), provided another suitable condition (Assumption 2.7) holds.

Theorem 2.10 establishes this result in the *realizable* setting; in Appendix D we extend all results to the *agnostic* case. Furthermore, Proposition 2.13 and Corollary 2.14 extend these guarantees to total variation distance. These latter results are general and may be of independent interest in broader contexts.

The sufficient conditions (Assumptions 2.1 and 2.7) are shown to be minimax necessary via Claims 2.15 and 2.17. We also verify that these assumptions hold for several important distribution families, via a series of Claims 2.2, 2.3, 2.4, 2.9, and 3.1. Our proofs are based on a novel *perturbation-quantization* technique and Fourier analysis, which naturally interface with the sample compression framework. Furthermore, in Section 3, we use our results to derive new sample complexity bounds for $k$-mixtures of uniform distributions over axis-aligned hyperrectangles in $\mathbb{R}^d$, whose sample complexity under noise corruption had previously remained open. We also obtain sample complexity bounds for learning $k$-mixtures of Gaussians under Laplace noise, a problem with applications to differential privacy that likewise remained open prior to this work.

## 2. Learnability under Additive Noise

In this section, we present our main results. The bounds are stated under the assumption that $G$ has independent and symmetrically distributed coordinates, which is solely to avoid excessively complicated formulations, and the results can be directly extended to more general scenarios. In Corollary 2.11, we also instantiate our results for two practically important noise models: Gaussian and Laplace. To establish our results, we introduce structural assumptions on the distribution class $\mathcal{F}$—namely, Assumptions 2.1 and 2.7. These assumptions are both *necessary* (in a minimax sense) and *sufficient* for learning $\mathcal{F}$ from corrupted samples.

Assumption 2.1 complements Assumption 1.3 (sample compressibility) by adding a stability condition. Specifically, it requires that $\mathcal{F}$ be not only sample-compressible but also *stably* so. This means that infinitesimally small perturbations in the input samples should not cause the decoder to produce drastically different distributions, at least in a neighborhood of *good* samples. We refer to this property as *Local Lipschitz Decodability*. See Claim 2.15 to see its minimax necessity. On the other hand, Assumption 2.7 states

that the probability density functions (pdfs) in $\mathcal{F}$ should not exhibit excessive fluctuations and must adhere to a certain degree of low-frequency behavior. Our impossibility results (e.g., Claim 2.17) theoretically show that functions with pronounced high-frequency components become increasingly difficult to recover from additive noise.

## 2.1. Local Lipschitz Decodability

**Assumption 2.1** (Local Lipschitz Decodability). Under Assumption 1.3, for a sufficiently large dataset $S$ there exists a sequence of at most $t(\epsilon)$ bits denoted by $\mathbf{B}$, and a random sequence of $\tau(\epsilon)$ samples from $S$, vectorized as $\mathbf{L} \in \mathbb{R}^{d\tau(\epsilon)}$, such that any underlying $f \in \mathcal{F}$ can be PAC-learned via a decoder $\mathcal{J}$. Additionally, assume that for any two sequences $\mathbf{B}, \mathbf{L}$ with $\mathsf{TV}\left(f, \mathcal{J}(\mathbf{L}, \mathbf{B})\right) \leq 1/2$, $\mathcal{J}$ behaves smoothly w.r.t. $\mathbf{L}$, i.e., there exists $r \geq 0$ such that any perturbed copy of $\mathbf{L}$, denoted by $\mathbf{L}' \in \mathbb{R}^{d\tau(\epsilon)}$, satisfies

$$\mathsf{TV}\left(\mathcal{J}(\mathbf{L}, \mathbf{B}), \mathcal{J}(\mathbf{L}', \mathbf{B})\right) \leq \tfrac{r}{2} \left\| \mathbf{L} - \mathbf{L}' \right\|_2. \tag{4}$$

This assumption is crucial not only from a mathematical standpoint but also in practical applications, where data samples are stored digitally and quantized using a finite number of bits. Decoders that are excessively sensitive to small perturbations are therefore impractical. The term "local" indicates that the Lipschitz property only needs to hold within a loose neighborhood around *good* samples $\mathbf{L}$.

Next, we theoretically examine several commonly used cases that adhere to this assumption, thus demonstrate that it is natural and not overly restrictive. Finally, in Claim 2.15, we establish a converse result: there exist scenarios where the assumption fails to hold, making learning from noisy samples provably impossible in such cases. The proofs for all the following claims is given in Appendix C.1.

**Claim 2.2.** Let $\mathcal{F} = \left\{ \mathcal{N}(\boldsymbol{\mu}, \sigma^2 \boldsymbol{I}) \mid \boldsymbol{\mu} \in \mathbb{R}^d, \ \sigma \geq \sigma_0 \right\}$ be a class of isotropic $d$-dimensional Gaussians with a component-wise variance of at least $\sigma_0^2$ for some $\sigma_0 > 0$. Then, $\mathcal{F}$ satisfies Assumption 2.1 with a Lipschitz constant of $r \leq \mathcal{O}\left( \frac{1}{\sigma_0 \sqrt{d \log(2d)}} \right)$.

Claim 2.2 can be extended to the general case of $\mathcal{N}(\boldsymbol{\mu}, \boldsymbol{\Sigma})$ with a positive definite $\boldsymbol{\Sigma} \succ 0$, provided that $\lambda_{\min}(\boldsymbol{\Sigma}) \geq \sigma_0 > 0$, where $\lambda_{\min}(\cdot)$ denotes the minimum eigenvalue. For brevity and readability of the proof in Section C.1, we have stated the claims in their simpler form.

**Claim 2.3.** Let $\mathcal{F} = \left\{ \mathsf{Unif}\left( \prod_i [a_i, b_i] \right) \mid a_i, b_i \in \mathbb{R}, \ b_i - a_i \geq T \right\}$ for $T > 0$, be the class of of uniform distributions over axis-aligned hyper-rectangles in $\mathbb{R}^d$ with a minimum width-per-dimension of $T > 0$. Then, $\mathcal{F}$ satisfies Assumption 2.1 with a Lipschitz constant $r \leq \frac{8d}{T}$.

The following claim is useful where learning a convex combination of a finite number of distributions from a given family—also known as a finite mixture—is required.

**Claim 2.4** (Conservation of Lipschitz Decodability under $k$-Mixture). *Let $\mathcal{F}$ admit a sample compression scheme with at least one Lipschitz decoder according to Assumption 2.1, and a corresponding constant $r \geq 0$. For $k \in \mathbb{N}$, consider the class of $k$-mixtures of $\mathcal{F}$ defined as $k-\mathrm{Mix}\left(\mathcal{F}\right) \triangleq \left\{ \sum_{i=1}^{k} \alpha_i f_i \,\middle|\, f_i \in \mathcal{F}, \ \alpha_i \geq 0, \ \sum_{i=1}^{k} \alpha_i = 1 \right\}.$ Then, $k-\mathrm{Mix}\left(\mathcal{F}\right)$ is also sample compressible (already proved by (Ashtiani et al., 2018)) and admits at least one locally Lipschitz decoder with a corresponding constant of $r\sqrt{k}$.*

## 2.2. Sample Compressibility of $\mathcal{F} \star G$

We present the following proposition, which is a key component of our main result in Theorem 2.10. This proposition states that if a distribution class $\mathcal{F}$ admits sample compression over $\subseteq \mathbb{R}^d$ with at least one decoder satisfying Assumption 2.1, then the noisy version $\mathcal{F} \star G \triangleq \{ f \star G \mid f \in \mathcal{F} \}$ is also sample compressible.

**Proposition 2.5** (Sample Compressibility of Noisy $\mathcal{F}$). *For $d \in \mathbb{N}$, assume $\mathcal{F}$ be a class of $d$-dimensional distributions satisfying Assumption 1.3 for functions $\tau, t, m : (0, 1) \to \mathbb{N}$, and let Assumption 2.1 hold for a bounded constant $r \geq 0$. Also, let $G$ be the density of an isotropic noise over $\mathbb{R}^d$ with a component-wise CDF of $\Phi_G : \mathbb{R} \to [0, 1]$. Then, $\mathcal{F} \star G$ admits*

$$\left[ \tau\left(\tfrac{\epsilon}{2}\right), t\left(\tfrac{\epsilon}{2}\right) + d\tau\left(\tfrac{\epsilon}{2}\right) \log_2 \left( 1 + \right. \right. \tag{5}$$
$$\left. \left. \frac{r}{\epsilon} \sqrt{d\tau\left(\tfrac{\epsilon}{2}\right)} \left| \Phi_G^{-1}\left( \frac{\delta}{4dm\left(\tfrac{\epsilon}{2}\right) \log \tfrac{2}{\delta}} \right) \right| \right), m\left(\tfrac{\epsilon}{2}\right) \right]$$

*-sample compression, for any $\epsilon \in (0, 1)$.*

Full proof is presented in Appendix C, however, we give a brief sketch of the proof here. Our methodology is based on *denoising* the samples before applying the available decoder from $\mathcal{F}$. Specifically, we assume access to noisy samples $\widetilde{\boldsymbol{X}}_i \triangleq \boldsymbol{X}_i + \boldsymbol{\zeta}_i$, where $\boldsymbol{\zeta}_i$ are the noise vectors drawn from $G$. Our goal is to approximate each noise vector $\boldsymbol{\zeta}_i$ with a *quantized* surrogate $\boldsymbol{\zeta}_i'$, ensuring that $\left\| \boldsymbol{\zeta}_i - \boldsymbol{\zeta}_i' \right\|_2 \leq \mathcal{O}(2^{-B})$ for a given $B \geq 1$. This way, we can use $\widetilde{\boldsymbol{X}}_i - \boldsymbol{\zeta}_i'$ as a partially denoised version of $\boldsymbol{X}_i$, with a small remaining residual noise due to non-ideal quantization. Since the values of $\boldsymbol{\zeta}_i$ are random and thus unknown, we approximate them by considering all high-probability values, which results in an overhead of $\mathcal{O}(B \times \tau(\epsilon))$ additional bits in the original sample compression scheme for $\mathcal{F}$. The small residuals can then be handles using local Lipschitz decodability assumption. Proposition 2.5 followed by Theorem 1.2 ensures the *PAC-learnability* of $f^* * G$ for any

$f^* \in \mathcal{F}$ via at most $N_{\text{clean}} + \widetilde{\mathcal{O}}(d\tau/\epsilon^2)\log(1 + r\sigma)$ i.i.d. noisy samples.

*Remark* 2.6 (Two Examples of $\Phi_G^{-1}$). For a product noise distribution $G$, let $\Phi_G$ denote its component-wise CDF. Consider the following two examples: Let $G \triangleq \mathcal{N}(\mathbf{0}, \sigma^2 \boldsymbol{I}_d)$ for some $\sigma > 0$, a common choice in many practical applications. Using Mill's ratio for the Gaussian distribution, we obtain: $\left|\Phi_G^{-1}(\Delta)\right| \leq \sigma\sqrt{2\log\left(\frac{1}{\sqrt{2\pi}\Delta}\right)}$, $\forall \Delta < 1/\sqrt{2\pi}$. Now, suppose each $G_i \triangleq \text{Laplace}(b)$ for some $b > 0$, a well-known choice, particularly in differential privacy research (Dwork, 2006; Muthukrishnan & Kalyani, 2025). Then, we have: $\left|\Phi_G^{-1}(\Delta)\right| = b\log\frac{1}{2\Delta}$, $\quad \forall \Delta \leq 1/2$.

## 2.3. $\ell_2$-**Learnability of** $\mathcal{F}$

What remains is to prove the learnability of $f^*$ itself. First, let us state a seemingly counter-intuitive fact: When $G$ is a known noise pdf, and given mild identifiability conditions on $\mathcal{F}$ with respect to $G$, it follows that for any $f_1, f_2 \in \mathcal{F}$, the condition $\mathsf{TV}(f_1 \star G, f_2 \star G) = 0$ implies $f_1 = f_2$. However, the PAC-learnability of $f^* \star G$ does not necessarily imply that $f^*$ can also be learned, at least in TV error. In other words, an *absolute* zero TV distance between $f_1 \star G$ and $f_2 \star G$ is fundamentally different from a *limiting* zero. Mathematically, there may exist a sequence $\{f_n\}_{n\in\mathbb{N}} \subset \mathcal{F}$ and a single density function $f^* \in \mathcal{F}$ such that

$$\lim_{n\to\infty} \mathsf{TV}(f_n \star G, f^* \star G) = 0,$$
$$\text{but} \quad \lim_{n\to\infty} \mathsf{TV}(f_n, f^*) > 0. \tag{6}$$

Before proving the existence of such sequences and similar to (Saberi et al., 2023), let us first introduce a *sufficient* condition on $\mathcal{F}$ that, as will become evident in Theorem 2.10, prevents this pathological phenomenon:

**Assumption 2.7** (Low-Frequency Property). For a distribution family $\mathcal{F}$, assume there exist $\alpha \geq 0$ and $\xi < 1$ such that for each $p, q \in \mathcal{F}$, with respective Fourier transforms $P, Q$, we have

$$\frac{1}{(2\pi)^d}\int_{\|\boldsymbol{w}\|_2 \geq \alpha}|P(\boldsymbol{w}) - Q(\boldsymbol{w})|^2\,\mathrm{d}\boldsymbol{w} \leq \xi\|p - q\|_2^2. \tag{7}$$

More generally, $\xi$ can be a function of $\varepsilon \triangleq \|p - q\|_2$, even with $\lim_{\varepsilon\to 0^+} \xi(\varepsilon) = 0$. However, we must have $\xi(\varepsilon) < 1$ for all $\varepsilon > 0$. Let $\mathsf{P}(\mathcal{F})$ denote the set of all $(\alpha, \xi)$ pairs that correspond to the above inequality for a class $\mathcal{F}$.

By Parseval's theorem, $\|p - q\|_2 = (2\pi)^{-d}\|P - Q\|_2$ for all $p, q \in \mathcal{F}$. However, Assumption 2.7 restricts attention to the high-frequency energy component of $\|P - Q\|_2$, integrating only over the region $\|\boldsymbol{w}\|_2 \geq \alpha$. In return, the total energy is expected to decrease by a factor of $\xi < 1$, implying that a non-negligible fraction of the $\ell_2$-energy of

$p - q$ is concentrated in lower frequencies. Several results in this section rely on this assumption to guarantee the recoverability of $f^*$ in terms of the $\ell_2$-norm or TV, provided that $f^* \star G$ is learnable. Before presenting our main result in Theorem 2.10, we first demonstrate that Assumption 2.7 holds in various practically relevant scenarios.

**Claim 2.8.** *For $\sigma_0 > 0$, assume the restricted Gaussian family $\mathcal{F} = \left\{\mathcal{N}(\boldsymbol{\mu}, \sigma^2\boldsymbol{I}) | \boldsymbol{\mu} \in \mathbb{R}^d, \sigma \geq \sigma_0\right\}$. Then, Assumption 2.7 holds with $\mathsf{P}(\mathcal{F}) \supseteq \left\{\left(\alpha, 2^{(d/2+2)}e^{-\sigma_0^2\alpha^2/2}\right) | \alpha > \frac{1}{\sigma_0}\sqrt{(d+4)\log 2}\right\}$.*

Similar to Claim 2.2, the results can be extended to the more general case of $\mathcal{N}(\boldsymbol{\mu}, \boldsymbol{\Sigma})$ with $\lambda_{\min}(\boldsymbol{\Sigma}) \geq \sigma_0$, however, this might complicate the proofs.

**Claim 2.9** ($k$-Mixtures of Uniform Measures over $\mathbb{R}$). *For $k \in \mathbb{N}$ and bandwidth $T > 0$, consider the following class of distributions: $\mathcal{F} = \left\{f : x \to \frac{\mathbb{1}(a\leq x\leq b)}{b-a} | b - a \geq T\right\}$. Then, letting $\varepsilon \triangleq \|p - q\|_2$, we have $\mathsf{P}(k-\text{Mix}(\mathcal{F})) \supseteq \left\{\left(\alpha, 1 - \zeta\left(\frac{\alpha T^2\varepsilon^2}{2(4k-1)}\right)\right) | \alpha > 0\right\}$, where function $\zeta(\cdot)$ is defined as $\zeta(h) \triangleq \frac{2}{\pi}\int_0^h \frac{\sin^2 u}{u^2}\,\mathrm{d}u$, $\forall h \geq 0$.*

Proofs are given in Appendix C.2. Claim 2.9 relies on a key property of the Fourier decay of indicator functions over convex bodies or shapes with smooth boundaries (see (Brandolini et al., 2003)), which in one dimension reduces to intervals. Claim 2.9 naturally extends to $\mathbb{R}^d$, covering a broad class of uniform distributions over convex or smooth bodies—such as polygons and hyperellipses—with several applications in learning high-dimensional shapes from noisy uniform samples (Boissonnat et al., 2007; Najafi et al., 2021; Saberi et al., 2023). While deriving nearly-tight sample complexity bounds for such classes lies beyond the scope of this paper, we point to it as a compelling direction for future work. Meanwhile, in Section 3, we utilize this claim to derive (for the first time) a sample complexity bound for learning $k$-mixtures of $d$-dimensional uniform distributions under noise perturbations. The following theorem establishes a bound in recovering $f^*$ in $\ell_2$-norm:

**Theorem 2.10** (Main Result). *Let $\mathcal{F}$ be a distribution family over $\mathcal{X} \subseteq \mathbb{R}^d$ satisfying Assumption 1.3 with a scheme $(\tau, t, m)$, and Assumption 2.1 with a bounded constant $r \geq 0$. Moreover, let Assumption 2.7 hold for the set of pairs $\mathsf{P}(\mathcal{F}) = \{(\alpha, \xi)\}$. Assume $G \in L^2(\mathcal{X})$ be a symmetric product measure with component-wise CDF of $\Phi_G$. Define*

$$B_G(\alpha) \triangleq \frac{\inf_{\|\boldsymbol{\omega}\|_2 \leq \alpha}|\mathsf{F}\{G\}(\boldsymbol{\omega})|}{\alpha^d \cdot \mathsf{Vol}\left(\mathbb{B}_2^d(1)\right)}$$

*for $\alpha > 0$, where $\mathbb{B}_2^d(1)$ denoting the $\ell_2$ ball of radius $1$ in $\mathbb{R}^d$. For any unknown $f^* \in \mathcal{F}$ and $\epsilon, \delta \in (0, 1)$, assume we*

*have $n$ i.i.d. samples from $f^* * G$ with*

$$n \geq N^{\mathsf{Clean}}_{\tau,t,m}(6\epsilon, \delta/2) + \mathcal{O}\left(\frac{d\tau(\epsilon)}{\epsilon^2} \log\left(m(\epsilon) \log \frac{1}{\delta}\right)\right.$$
$$\left. \log\left(\frac{r}{\epsilon}\sqrt{d\tau(\epsilon)} \left|\Phi_G^{-1}\left(\frac{\delta}{8dm(\epsilon)\log\frac{4}{\delta}}\right)\right|\right)\right),$$

*where $N^{\mathsf{Clean}}_{\tau,t,m}(\epsilon, \delta)$ is the sample complexity of the noiseless regime, as defined in Theorem 1.2 (full details in Theorem C.3). Then, there exists an algorithm that takes the $n$ perturbed samples as input, and outputs $\widehat{f} \in \mathcal{F}$ such that the following bound holds with probability at least $1 - \delta$:*

$$\|\widehat{f} - f^*\|_2 \leq \epsilon \cdot \left(\inf_{(\alpha,\xi)\in\mathsf{P}(\mathcal{F})} \frac{24}{\sqrt{B_G(\alpha)(1-\xi)}}\right). \quad (8)$$

The full proof is given in Appendix C. Let us investigate two special cases of Gaussian and multi-dimensional Laplace noise distributions as candidates for $G$ (proof is given in Appendix C, as well).

**Corollary 2.11** (Gaussian and Laplace Noise Models)**.** *Consider the setting of Theorem 2.10. Assume two scenarios for noise distribution $G$:* i) *Gaussian noise $G \triangleq \mathcal{N}\left(\mathbf{0}, \sigma^2\mathbf{I}_d\right)$ for some $\sigma > 0$, and* ii) *multi-dimensional Laplace noise $G_i \triangleq \mathrm{Laplace}(\sigma)$, $i \in [d]$. Then, assuming*

$$n \geq \mathcal{O}\left(N^{\mathsf{Clean}}_{\tau,t,m}(\epsilon, \delta)\right) + \widetilde{\mathcal{O}}\left(\frac{d\tau(\epsilon)}{\epsilon^2}\right) \log(1 + \sigma r), \quad (9)$$

*with probability at least $1 - \delta$, the $\ell_2$ error $\|\widehat{f} - f^*\|_2$ corresponding to cases* i) *and* ii) *is respectively bounded as*

i) $\|\widehat{f} - f^*\|_2 \leq \epsilon \displaystyle\inf_{(\alpha,\xi)\in\mathsf{P}(\mathcal{F})} 24\sqrt{\dfrac{\mathsf{Vol}(\mathbb{B}_2^d(1))\alpha^d e^{(\sigma\alpha)^2}}{1-\xi}},$

ii) $\|\widehat{f} - f^*\|_2 \leq \epsilon \displaystyle\inf_{(\alpha,\xi)\in\mathsf{P}(\mathcal{F})} \dfrac{24\sqrt{\mathsf{Vol}(\mathbb{B}_2^d(1))}}{\sqrt{1-\xi}}\left(\alpha + \dfrac{b^2\alpha^3}{d}\right)^{d/2}$

Theorem 2.10 essentially states that to learn $f^*$ up to a $\ell_2$ error of $\epsilon$ with high probability (at least $1 - \delta$), one requires $\widetilde{\mathcal{O}}\left(N^{\mathsf{Clean}}_{\tau,t,m}(\epsilon, \delta) + \frac{d\tau(\epsilon)}{\epsilon^2}\right)$ samples. The first term represents the *vanilla* sample complexity of learning $f^*$ from clean samples, while the second term accounts for the additional cost introduced by the presence of noise. Notably, in the case of learning $f^* \star G$, only the dimension $d$ and $\tau(\epsilon)$—the length of the decoder input samples—explicitly appear in the bound. Other factors, such as noise power (inherent in $\Phi_G$) and the local Lipschitz constant $r$, are encapsulated within polylogarithmic terms.

Moreover, the theorem asserts that learning $f^* \star G$ up to a TV error of $\epsilon$ is equivalent to learning $f^*$ up to an $\ell_2$-error proportional to $\epsilon$. The proportionality constant depends on specific properties of the noise distribution, including its variance, as well as Fourier-analytic characteristics of $\mathcal{F}$. For instance, in both Gaussian and Laplace noise models,

the resulting sample complexity may grow exponentially with the noise variance (see Corollary 2.11). This phenomenon, often referred to as *dimension blow-up*, has also been observed in related settings, such as learning high-dimensional simplices from noisy samples (Saberi et al., 2023). We conjecture that such an aggressive increase in sample complexity in low-SNR (high-noise) regimes is unavoidable when learning under the *TV distance*, since, as discussed earlier, nearly all practical algorithms exhibit severe empirical performance degradation as the noise level increases. In contrast, this phenomenon may not persist under less sensitive metrics, such as the Wasserstein distance. Investigating this possibility constitutes a promising direction for future work.

*Remark* 2.12 (Unknown Noise Variance)**.** Theorem 2.10 assumes full knowledge of the noise distribution $G$, which may be restrictive in practice. This assumption can be relaxed by considering settings in which only the family of the noise distribution is known—for example, Gaussian noise with unknown variance—while assuming that the unknown parameter is upper bounded by a known constant. In such cases, the variance parameter can be incorporated into the sample compression framework, discretized via quantization, and subsequently learned from the data. We omit the technical details of this extension in order to preserve the simplicity and readability of the presentation.

### 2.4. Guarantees on Total Variation Error

Theorem 2.10 establishes PAC-learnability in $\ell_2$-norm. However, in many scenarios, learning guarantees under the TV norm are of greater interest, as in the noise-free sample compression scheme of (Ashtiani et al., 2018). Generally, the TV error cannot be directly bounded by the $\ell_2$-error, and several impossibility results exist in this regard (see (Devroye et al., 2013)). Nonetheless, under certain sufficient—but not necessary—conditions on the tail decay rate of the PDFs in $\mathcal{F}$, it is possible to derive such bounds.

**Proposition 2.13** (Water-Filling Bound on TV Error via $\ell_2$-Norm)**.** *Let $f, g \in L^2(\mathcal{X})$, and assume $g(\boldsymbol{x}) \geq 0$ for all $\boldsymbol{x} \in \mathcal{X}$. Consider the following water-filling construction: find a Lebesgue-measurable set $A = A(\|f\|_2, g) \subseteq \mathcal{X}$ such that the followings hold:*

$$\mathsf{Vol}(A) \cdot \inf_{\boldsymbol{x}\in A} g^2(\boldsymbol{x}) + \int_{\mathcal{X}\backslash A} g^2(\boldsymbol{x})\mathrm{d}\boldsymbol{x} = \|f\|_2^2,$$

$$\text{and} \quad g(\boldsymbol{x}) \geq g(\boldsymbol{y}), \quad \forall(\boldsymbol{x}\in A, \ \boldsymbol{y}\notin A). \quad (10)$$

*Then, assuming $|f(\boldsymbol{x})| \leq g(\boldsymbol{x})$ for all $\boldsymbol{x} \in \mathcal{X}$, the following upper bound holds:*

$$\|f\|_1 \leq \mathsf{Vol}(A) \inf_{\boldsymbol{x}\in A(\|f\|_2,g)} g(\boldsymbol{x}) + \int_{\mathcal{X}\backslash A(\|f\|_2,g)} g(\boldsymbol{x})\mathrm{d}\boldsymbol{x}.$$

As it becomes evident during the proof of Proposition 2.13 (see Appendix C), the above procedure to determine the set

$A$ corresponds to a water-filling construction. The following corollary (also proved in Appendix C) illustrates two specific tail decay conditions—serving as concrete choices for the function $g$ in Proposition 2.13—and derives explicit bounds on the TV error when recovering $f^* \in \mathcal{F}$ based on $n$ noisy samples drawn from $f^* * G$.

**Corollary 2.14** (Bounded Support or Sub-Gaussianity). *Assume the setting of Theorem 2.10, and consider the following cases:* i) *Suppose that $\mathcal{F}$ has bounded support; that is, there exists $R > 0$ such that $\mathrm{supp}(f) \subseteq [-R, R]^d$ for all $f \in \mathcal{F}$. Then,*

$$\mathsf{TV}(\widehat{f}, f^*) \leq (2R)^{d/2} \big\| \widehat{f} - f^* \big\|_2. \qquad (11)$$

ii) *Suppose every $f \in \mathcal{F}$ satisfies a sub-Gaussian bound: there exist constants $C_1, \gamma > 0$ such that for all $\boldsymbol{x} \in \mathbb{R}^d$, $f(\boldsymbol{x}) \leq C_1 \exp(-\gamma \|\boldsymbol{x} - \boldsymbol{\mu}\|_2^2)$, where $\boldsymbol{\mu}$ is the mean of $f$. Then,*

$$\mathsf{TV}(\widehat{f}, f^*) \leq C_2 \big\| \widehat{f} - f^* \big\|_2 \cdot \log^{d/2} \left( \frac{1}{\big\| \widehat{f} - f^* \big\|_2} \right) \quad (12)$$

*for some constant $C_2$ depending on $C_1$, $\gamma$, and $d$.*

### 2.5. Minimax Necessity of Assumptions 2.1 and 2.7

Next, we present an example of a sample-compressible family that lacks the local Lipschitz decodability property for any finite $r \geq 0$ and is provably unlearnable in the PAC sense.

**Claim 2.15** (Minimax Necessity of Assumption 2.1). *Suppose $\mathcal{F}$ is the class of distributions $\{\mathcal{N}(\mu, \sigma^2) | \mu \in \mathbb{R},\ \sigma > 0\}$. This class admits $(2, 0, \widetilde{\mathcal{O}}(1/\epsilon))$-s.c., but does not satisfy Assumption 2.1 for any finite $r$. Let $G = \mathcal{N}(0, \sigma_0^2)$ for some $\sigma_0 > 0$, and define $\mathsf{A}(n, B)$ as the set of all decoders for $\mathcal{F}$ that take $n$ noisy samples and $B$ bits as input and output a corresponding $\widehat{f} \in \mathcal{F}$. Then, for any fixed $n, B \in \mathbb{N}$ and assuming $\mathbf{L} = \{\boldsymbol{X}_i\}_{i=1}^n \overset{i.i.d.}{\sim} f^* \star G$ for $f^* \in \mathcal{F}$, for $\mathscr{A} \in \mathsf{A}(n, B)$, the following holds:*

$$\inf_{\mathscr{A}} \sup_{f^* \in \mathcal{F}} \mathbb{P}\left( \min_{\mathbf{B} \in \{0,1\}^B} \mathsf{TV}\left(f^*, \mathscr{A}\left(\mathbf{L}, \mathbf{B}\right)\right) \geq \frac{1}{200} \right) \geq \frac{1}{2}.$$

The claim asserts that regardless of how large $n$ and/or $B$ are, no algorithm that receives $n$ i.i.d. samples from $f^* * G$ can learn $\mathcal{F}$ in the TV sense. The proof relies on techniques from minimax theory, specifically Le Cam's method, to establish this impossibility result.

*Remark* 2.16. Singular density functions with continuously varying degrees of freedom—such as a Gaussian distribution with an infinitesimally small variance and an arbitrary mean, or a uniform distribution with infinitesimally small bandwidth—cannot be learned in the TV error sense from samples corrupted by continuous noise, e.g., $\mathcal{N}(0, \sigma_0^2)$.

While a formal proof of this claim is beyond the scope of this paper, it is still a meaningful and important observation. Intuitively, distributions that become (asymptotically) concentrated over a zero-measure region of the space, cannot be reliably learned (at least in TV error) in noisy regimes. To demonstrate the minimax-necessity of Assumption 2.7, Claim 2.17 provides an example of a distribution family that does not satisfy Assumption 2.7 and show that the pathological phenomenon of (6) can occur. It also means that the distribution family is not be PAC-learnable in TV error in a minimax sense.

**Claim 2.17** (Minimax Necessity of Assumption 2.7). *For $x \in [0, 2\pi]$, suppose the distribution class $\mathcal{F} = \left\{ f : x \to \frac{1 + (-1)^i \sin kx}{2\pi} \ \middle| \ k \in \mathbb{Z}_{\geq 0},\ i \in \{0, 1\} \right\}$. Then, $\mathcal{F}$ does not satisfy Assumption 2.7. Define $\mathsf{A}(n, B)$ as the set of all decoders for $\mathcal{F}$ that take $n$ noisy samples and $B$ bits as input and output a corresponding $\widehat{f} \in \mathcal{F}$. Then, there exists constant $c > 0$, such that for any fixed $n, B \in \mathbb{N}$, having $\mathbf{L} = \{\boldsymbol{X}_i\}_{i=1}^n \overset{i.i.d.}{\sim} f^* * G$ for $f^* \in \mathcal{F}$, and $\mathscr{A} \in \mathsf{A}(n, B)$, we have*

$$\inf_{\mathscr{A}} \sup_{f^* \in \mathcal{F}} \mathbb{P}\left( \min_{\mathbf{B} \in \{0,1\}^B} \mathsf{TV}\left(f^*, \mathscr{A}(\mathbf{L}, \mathbf{B})\right) \geq c \right) \geq \frac{1}{4}.$$

The proof of this claim can be found in Appendix C.2. We first show there exists at least one sequence $\{f_n\}_{n \in \mathbb{N}} \subset \mathcal{F}$ such that (6) happens with $G = \mathcal{N}(0, \sigma_0^2)$ for any $\sigma_0 > 0$. Due to Le Cam's lemma, this immediately results in the impossibility of PAC-learning $\mathcal{F}$ in TV error.

## 3. Theoretical Example

We demonstrate how the findings from the previous parts of this work come together to solve a theoretical example that, to the best of our knowledge, has not been previously addressed. In our example, we consider the problem of learning Uniform Mixture Models (UMMs). For $T > 0$, consider the class of distributions $\mathcal{F}$, consisting of uniform distributions over axis-aligned hyper-rectangles in $\mathbb{R}^d$, defined as (also see Claim 2.3):

$$\mathcal{F} = \left\{ \boldsymbol{x} \mapsto \prod_{i=1}^d \frac{\mathbb{1}(a_i \leq x_i \leq b_i)}{b_i - a_i} \ \middle| \ \boldsymbol{a}, \boldsymbol{b} \in \mathbb{R}^d,\ b_i - a_i \geq T \right\}.$$

For any $k \in \mathbb{N}$, we aim to analyze the sample complexity of $k$-mixtures of $\mathcal{F}$, also known as $k$-uniform mixture models or $k$-UMMs, denoted $k - \mathrm{Mix}(\mathcal{F})$ under the perturbation model considered thus far (see Claim 2.4 for a formal definition).

This family is widely employed for modeling piecewise constant probability density functions (Browne et al., 2011; Brunot, 2019), yet its sample complexity under noise corruption has remained open (Najafi et al., 2021; Brunot, 2019).

In fact, any density function with mild continuity properties (such as piecewise continuity) can be closely approximated by a $k$-UMM, provided $k$ is chosen sufficiently large. Hence, providing explicit sample complexity guarantees for learning such models under general perturbations (e.g., noise interference) is of both theoretical and practical significance. We now verify that the assumptions required by Theorem 2.10 is satisfied for this distribution class.

- Each component of the $k$-UMM $f^*$ is a product of $d$ one-dimensional uniform distributions. A one-dimensional uniform distribution over $\mathbb{R}$ admits $(2, 0, \frac{2}{\epsilon} \log \frac{2}{\delta})$-compression for any $\epsilon, \delta \in (0, 1)$. This is because identifying the minimum and maximum of the support interval suffices to reconstruct the distribution, and with at most $\frac{2}{\epsilon} \log \frac{2}{\delta}$ samples, we can guarantee the existence of two $\epsilon/2$-close surrogates for these extremes.

- It has been proved that products of compressible distributions remain compressible. In particular, according to Lemma 3.6 of (Ashtiani et al., 2018), $\mathcal{F}$ admits $\left(2d, 0, \frac{2d}{\epsilon} \log \frac{2}{\delta} \log(3d)\right)$-s.c..

- In addition, Lemma 3.7 of (Ashtiani et al., 2018) implies that the class $k-\mathrm{Mix}(\mathcal{F})$ admits $\left(2kd, k \log_2 \frac{4k}{\epsilon}, \frac{288dk}{\epsilon} \log \frac{2}{\delta} \log \frac{6k}{\epsilon} \log(3d)\right)$-s.c., which satisfies Assumption 1.3.

- Moreover, from Claim 2.3, each component of $\mathcal{F}$ satisfies Assumption 2.1 with Lipschitz constant $r \leq \frac{8d}{T}$. Consequently, by Claim 2.4, the class $k-\mathrm{Mix}(\mathcal{F})$ satisfies the same assumption with Lipschitz constant $r \leq \frac{8d}{T}\sqrt{k}$.

Finally, the following claim generalizes Claim 2.9 to $d$ dimensions. It identifies an appropriate set of pairs $(\alpha, \xi(\cdot)) \in \mathsf{P}(k-\mathrm{Mix}(\mathcal{F}))$, which completes the requirements for applying Theorem 2.10 and Corollary 2.11.

**Claim 3.1** (Extension of Claim 2.9 to $d$ dimensions). *Let $d, k \in \mathbb{N}$ and minimal bandwidth $T > 0$. Let the class $\mathcal{F}$ be a $d$-dimensional UMM with bandwidth $T$. Then, for $\varepsilon \triangleq \|p - q\|_2$, the class $k-\mathrm{Mix}(\mathcal{F})$ satisfies Assumption 2.7 with:*

$$\mathsf{P}(k-\mathrm{Mix}(\mathcal{F})) \supseteq \left\{ \left(\alpha,\ 1 - \zeta^d\left(\frac{\alpha}{2ck\sqrt{d}}(T\varepsilon)^{2/d}\right)\right) \,\Big|\, \alpha > 0 \right\},$$

*where $\zeta(\cdot)$ is defined as in Claim 2.9, and $c > 0$ is a universal constant.*

Proof is given in Appendix C.2. We now present our main result through the following proposition:

**Proposition 3.2** (Learnability of $k$-UMMs from Noisy Samples). *Consider a target distribution $f^* \in k-\mathrm{Mix}(\mathcal{F})$, and assume we have access to $n$ i.i.d. samples corrupted by*

*additive Gaussian noise: $\zeta_1, \ldots, \zeta_n \overset{i.i.d.}{\sim} \mathcal{N}(\mathbf{0}, \sigma^2 \mathbf{I}_d)$ for a sufficiently large $\sigma > 0$. Then, for any $\epsilon, \delta > 0$, there exists an estimator $\widehat{f}$ such that upon having*

$$n \geq \mathcal{O}\left(\frac{d^2 k}{\epsilon^2} \log^2\left(\frac{rdk\sigma}{\epsilon\delta}\right)\right),$$

*guarantees that*

$$\|\widehat{f} - f^*\|_2^2 \leq \frac{24\epsilon}{T}\left(\pi ck\sqrt{d}\right)^{d/2} \sqrt{\mathsf{Vol}(\mathbb{B}_2^d(1))}$$

*with probability at least $1 - \delta$.*

The proof is provided in Appendix E and follows directly by applying the steps outlined in Theorem 2.10 and Corollary 2.11 to the specific properties of the $k$-UMMs established above. When $\sigma$ is small (i.e., $\sigma \to 0$), the resulting bounds become too intricate to express in closed form due to the behavior of $\zeta(\cdot)$ in Claim 3.1. An interested reader is referred to the proof for the precise characterization of the bounds in the small-$\sigma$ regime.

A simple rearrangement reveals that a very similar sample complexity and error bound to those of Proposition 3.2 also apply for the case of having a $\mathrm{Laplace}(b)$-distributed noise instead of Gaussian noise.

## 4. Conclusions

In this work, we extended the theoretical framework of sample compressibility to accommodate perturbed data, encompassing stochastic noise corruption. We demonstrated that, under mild and general assumptions, sample-compressible distribution families remain learnable, with a quantifiable inflation in sample complexity due to perturbations. Along the way, we showed that many well-known parametric distribution families satisfy our assumptions with reasonable constants and coefficients, while also establishing minimax impossibility results that highlight the necessity of our conditions.

The core technical contribution is a new perturbation quantization technique that aligns naturally with the structure of sample compression, offering a perspective not attainable through traditional learnability frameworks such as Valiant's PAC model (Valiant, 1984). Nevertheless, assuming the sample compression conjecture—which posits the equivalence of PAC learnability and sample-compressibility—our results carry broader generality. Our quantization strategy integrates seamlessly with existing compression schemes, enabling robust learning guarantees under both $\ell_2$ and total variation distance metrics. As concrete illustrations of our methods, we resolved a previously open problem: the learnability of finite mixtures of uniform distributions (with a bounded minimum bandwidth per dimension) under noisy perturbations.

## 4.1. Open Problems

Several important directions remain open for future investigation. We highlight a few examples:

- While our assumptions for learnability under perturbations are shown to be minimax-necessary, we only prove necessity in a limited (minimax) sense. A key open problem is to complete the necessity side of our results by characterizing conditions under which no PAC-learnability is achievable. In particular, it is unclear whether our conditions can be relaxed or fully characterized in a general necessary-and-sufficient form.

- Another challenging question is whether some of the imposed assumptions (e.g., Local Lipschitz Decodability and the Low-Frequency Property) can be derived from one another. Establishing logical implications between these structural properties may help simplify or unify the current framework. However, such results would likely hinge on some deep conjectures such as the one asserting that PAC-learnability implies the existence of an efficient sample compression scheme—a conjecture which was suggested by (Ashtiani et al., 2018) and still unresolved.

- Our analysis is information-theoretic in nature. Extending these results to efficient (polynomial-time) algorithms, particularly in high dimensions and under adversarial conditions, remains an open challenge.

## Acknowledgements

We thank the anonymous reviewers and area chair for their constructive feedback and helpful suggestions.

## Impact Statement

This paper presents theoretical work whose primary goal is to advance the foundational understanding of distribution learning from noisy samples. Our results provide information-theoretic guarantees for when sample-compressible distribution families remain learnable under additive perturbations, and clarify structural conditions under which such learning is possible. The work is methodological in nature and does not introduce new datasets, deployed systems, or application-specific decision-making tools. While the results may eventually inform the design and analysis of more robust learning methods, any societal consequences would arise only through downstream applications. We do not anticipate direct negative societal impacts from this work.

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

# A. Preliminaries and Formal Definitions

## A.1. PAC Learnability Definition

We formally define PAC learnability below.

**Definition A.1** (PAC Learnability: Realizable and Agnostic Settings). Let $\mathcal{F}$ be a class of probability distributions on a measurable space $\mathcal{X}$, and let $g$ be a target distribution on $\mathcal{X}$. Let $\mathcal{D} = \boldsymbol{X}_1, \ldots, \boldsymbol{X}_n$ denote $n$ i.i.d. samples drawn from $g$. For $\epsilon, \delta > 0$, the class $\mathcal{F}$ is said to be *PAC learnable* under total variation distance if there exist an algorithm $\mathscr{A}$ and a polynomial function $\mathrm{poly}(\cdot, \cdot)$ such that, whenever $n \geq \mathrm{poly}(\epsilon^{-1}, \delta^{-1})$, the estimator $\widehat{f} = \mathscr{A}(\mathcal{D}) \in \mathcal{F}$ satisfies

$$\mathsf{TV}(g, \widehat{f}) \leq C \inf_{f \in \mathcal{F}} \mathsf{TV}(g, f) + \epsilon$$

with probability at least $1 - \delta$ for some constant $C \geq 1$. If $g \in \mathcal{F}$, the setting is *realizable*, in which case

$$\mathsf{TV}(f^*, \widehat{f}) \leq \epsilon \quad \text{for any } f^* \in \mathcal{F}.$$

If $g \notin \mathcal{F}$, the setting is *agnostic*, and the goal is to compete with the optimal approximation to $g$ within $\mathcal{F}$ up to the constant factor $C \geq 1$. In some families $C = 1$ is achievable, while for others no procedure can attain $C = 1$ in general.

## A.2. KL Divergence

The KL divergence between $f_1$ and $f_2$ is defined as

$$\mathsf{KL}\left(f_1 \| f_2\right) \triangleq \mathbb{E}_{P_1}\left[\log\left(f_1(\boldsymbol{x}) / f_2(\boldsymbol{x})\right)\right].$$

We have $\mathsf{KL}\left(f_1 \| f_2\right) = \infty$ if $f_1$ is not absolutely continuous w.r.t. $f_2$.

## A.3. Sample Compression

Following (Ashtiani et al., 2018), we define the distribution decoder and sample compression scheme as follows. Let $\mathcal{X}$ be a measurable space, typically $\mathcal{X} \subseteq \mathbb{R}^d$ for some dimension $d \in \mathbb{N}$, and let $\mathcal{F}$ be a class of distributions supported on $\mathcal{X}$. The *decoders* for $\mathcal{F}$ are defined as:

**Definition A.2** (Distribution Decoder). A distribution decoder for $\mathcal{F}$ is a deterministic function $\mathcal{J}$ that takes a finite sequence of elements of $\mathcal{X}$ and a finite sequence of bits as input, and outputs an element of $\mathcal{F}$. Specifically, $\mathcal{J} : \bigcup_{i=0}^{\infty} \mathcal{X}^i \times \bigcup_{i=0}^{\infty} \{0, 1\}^i \to \mathcal{F}$.

**Definition A.3** (Sample Compression). Let $\tau, t, m : (0, 1) \to \mathbb{Z}_{\geq 0}$ be functions. A class $\mathcal{F}$ is said to admit $(\tau, t, m)$-sample compression (s.c.) if there exists a decoder $\mathcal{J}$ for $\mathcal{F}$ such that for any distribution $f \in \mathcal{F}$, the following holds: For any $\epsilon, \delta \in (0, 1)$, if an i.i.d. sample set $S$ of size $n \geq m(\epsilon) \log\left(\frac{1}{\delta}\right)$ is drawn from $f$, then with probability at least $1 - \delta$, there exists a sequence $\mathbf{L}$ of at most $\tau(\epsilon)$ elements of $S$ and a sequence $\mathbf{B}$ of at most $t(\epsilon)$ bits, such that $\|\mathcal{J}(\mathbf{L}, \mathbf{B}) - f\|_{\mathrm{TV}} \leq \epsilon$.

For example, the class of Gaussian distributions $\mathcal{F} = \left\{\mathcal{N}(\mu, \sigma^2) \mid \mu \in \mathbb{R}, \sigma > 0\right\}$ admits a $(2, 0, \mathcal{O}(\frac{1}{\epsilon} \log \frac{1}{\epsilon}))$-s.c. for any $\epsilon, \delta \in (0, 1)$ (see Figure 2). Specifically, given $\mathcal{O}(\frac{1}{\epsilon} \log \frac{1}{\epsilon} \log \frac{1}{\delta})$ i.i.d. samples from any $f \in \mathcal{F}$ (with mean $\mu$ and standard deviation $\sigma$), one can guarantee that with probability at least $1 - \delta$, two distinct samples will fall within an $\epsilon$-neighborhood of both $\mu - \sigma$ and $\mu + \sigma$. This results in a simple scheme (see Figure 2) that $\epsilon$-estimates $f^*$ according to TV error. An interesting feature of sample compression is that we do not need to explicitly identify which samples, when fed to the decoder, recover $f$. The key requirement is that the sequences of $\tau(\epsilon)$ samples and $t(\epsilon)$ bits only *exist*, without needing to specify an algorithm for finding them.

Many naturally occurring and practical distribution families are known to admit *efficient* sample compression schemes. In a weaker sense, any PAC-learnable class of distributions is sample-compressible by at least a naive scheme. Conversely, (Ashtiani et al., 2018) demonstrated that sample compressibility guarantees PAC-learnability (albeit potentially through an exponential-time algorithm). The core idea behind their proof is simple: we divide the training dataset into two non-overlapping partitions. Since we need to determine a sequence of at most $\tau(\epsilon)$ elements out of $m(\epsilon) \log \frac{1}{\delta}$ samples (where repetition is allowed and order matters) and at most $t(\epsilon)$ bits, using the decoder $\mathcal{J}$ and the first partition, we can generate a finite set of at most $(m(\epsilon) \log \frac{1}{\delta})^{\tau(\epsilon)} 2^{t(\epsilon)}$ candidate distributions, with at least one of them guaranteed to be close to the true distribution $f$. We then apply a multiple hypothesis testing procedure with the second partition to approximate $f$. Specifically, we use Theorem A.4 (from (Ashtiani et al., 2018), originally from (Devroye & Lugosi, 2001)) to guarantee that a multiple hypothesis testing procedure using enough samples will recover $f$ up to a small error.

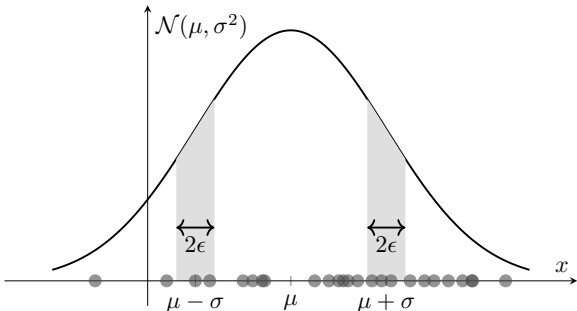

*Figure 2.* Depiction of the $(2, 0, \mathcal{O}(\frac{1}{\epsilon} \log \frac{1}{\epsilon}))$-sample compression scheme for $\mathcal{N}(\mu, \sigma^2)$ over $\mathbb{R}$. Given $\mathcal{O}(\frac{1}{\epsilon} \log \frac{1}{\epsilon} \log \frac{1}{\delta})$ i.i.d. samples from this Gaussian (shown as grey dots), with probability at least $1 - \delta$, there exist two distinct samples, $X_i$ and $X_j$, falling within $\epsilon$-neighborhoods of $\mu - \sigma$ and $\mu + \sigma$, respectively. A simple decoder, defined as $\mathcal{N}\left((X_i + X_j)/2, (X_j - X_i)^2/4\right)$, then reconstructs an estimate of the distribution with TV error $\mathcal{O}(\epsilon)$. Importantly, the indices $i$ and $j$ need not be known — it suffices that such samples exist within the dataset.

**Theorem A.4** (Theorem 3.4 of (Ashtiani et al., 2018)). *There exists a deterministic algorithm that, given candidate distributions $f_1, \ldots, f_M$, parameters $\epsilon, \delta > 0$, and $\log\left(M^2/\delta\right)/(2\epsilon^2)$ i.i.d. samples from an unknown distribution $g$ (not necessarily in $\mathcal{F}$), outputs an index $j \in [M]$ such that $\|f_j - g\|_1 \leq \min_{i \in [M]} \|f_i - g\|_1 + 4\epsilon$, with probability at least $1 - \delta$.*

**Corollary A.5** (Theorem 3.5 of (Ashtiani et al., 2018)). *Suppose $\mathcal{F}$ admits $(\tau, t, m)$-sample compression for some functions $\tau, t, m : (0, 1) \to \mathbb{N}$. Then, there exists a deterministic algorithm that, for any $\epsilon \in (0, 1)$, given at least*

$$n \geq N_{\tau,t,m}^{\mathsf{Clean}}(\epsilon, \delta) \triangleq \widetilde{\mathcal{O}}\left(m\left(\epsilon/6\right) + \frac{\tau\left(\epsilon/6\right) + t\left(\epsilon/6\right)}{\epsilon^2}\right) \tag{13}$$

*i.i.d. samples from any unknown distribution $g \in \mathcal{F}$, outputs $\widehat{f} \in \mathcal{F}$ such that $\|f - g\|_{\mathrm{TV}} \leq \epsilon$ with high probability.*

Corollary A.5 establishes the (information-theoretic) PAC learnability of $\mathcal{F}$ under the existence of a sample compression scheme. The complete version, including all polylogarithmic dependencies, is stated in Theorem C.3. However, as noted earlier, the sample-complexity bound in (13) may not be tight when compared with other approaches for proving PAC learnability.

**Conjecture A.6** (Originally posed by Ashtiani et al. (Ashtiani et al., 2020)). *For every learnable class, there exists a tight sample compression scheme whose sample complexity matches the optimal PAC rate via Corollary A.5.*

If true, this conjecture would imply that *every* information-theoretically learnable class can, without loss of generality, be characterized through an appropriate sample compression scheme. Thus, $N_{\mathrm{clean}}(\epsilon, \delta)$ as introduced in (13) may coincide with the *optimal* sample complexity for $\mathcal{F}$.

## B. Previous Works

Information-theoretic distribution learning is deeply rooted in statistical learning theory. It encompasses both parametric and nonparametric settings, mixture models, and robustness under various data-corruption regimes. The literature traces back to the foundational works of Kearns (Kearns et al., 1994) and Valiant (Valiant, 1984), and includes classical approaches based on uniform convergence and Vapnik–Chervonenkis (VC) theory—such as the *minimum distance estimator* introduced by Yatracos (Yatracos, 1985)—as well as more recent frameworks that incorporate structural assumptions, for example, distributions supported on low-dimensional manifolds (Berenfeld & Hoffmann, 2021). A comprehensive account of classical methods can be found in Devroye and Lugosi (Devroye & Lugosi, 2001).

**Parametric families and mixtures.** For structured parametric families (e.g., Gaussian and exponential families), classical results provide explicit sample-complexity bounds. In particular, Yatracos (Yatracos, 1985) derives general bounds based on the "massiveness" of the hypothesis set, quantified via entropic quantities such as covering numbers. The present work builds on this perspective, but in a different direction: we focus on *sample compressibility* as the structural property that guarantees learnability.

A major line of research concerns mixture models, particularly mixtures of Gaussians. Moitra and Valiant (Moitra & Valiant, 2010) established the first polynomial-time learnability results under mild separation assumptions, and subsequent works refined and extended these guarantees across various regimes. This line of work culminated in the result of Ashtiani et al. (Ashtiani et al., 2020), who derived an upper bound for learning general Gaussian mixture models in total variation (TV) distance that nearly matches the best known lower bounds.

A complementary line of research employs information-theoretic tools such as Shannon entropy and mutual information (Bresler, 2015; Najafi et al., 2020). In particular, Bresler (Bresler, 2015) obtained near-tight sample-complexity bounds for learning high-dimensional Ising models, while Najafi et al. (Najafi et al., 2020) used similar techniques to derive bounds for learning arbitrary mixtures of high-dimensional Bernoulli models. The literature on mixture-model learning is broad and diverse, encompassing specialized algorithmic frameworks such as Fourier-based methods for mixture estimation (Qiao et al., 2022). Recent developments have also intersected with modern concerns such as privacy-preserving (differentially private) learning (Arbas et al., 2023; Bun et al., 2024).

**Nonparametric density estimation.** Nonparametric density estimation—via kernels, histograms, projection techniques, and minimax theory—yields minimax rates for smooth densities (Hölder, Sobolev, monotone/unimodal, log-concave). Classical references include Tsybakov (Tsybakov, 2009) and Devroye & Lugosi (Devroye & Lugosi, 2001). These works also provide foundations for lower bounds used to evaluate distribution-learning procedures (Bilodeau et al., 2023; Wang & Marzouk, 2022; Devroye et al., 2020), including extensions to adversarial losses (Tang & Yang, 2023). Upper- and lower-bound results in this area often give loose estimates for parametric learning, as they do not exploit low-complexity structures—a limitation largely mitigated by sample-compression methods.

**Sample Compression.** A fundamental property that broadly guarantees learnability is *sample compressibility*, introduced by Littlestone and Warmuth (Littlestone & Warmuth, 1986) and adapted for distribution learning by Ashtiani et al. (Ashtiani et al., 2018) (see Definition A.3). Sample-compression schemes provide a conceptual bridge between combinatorial structure (e.g., VC-type parameters) and sample complexity bounds. This notion has enabled nearly tight sample-complexity bounds for Gaussian mixture models (Ashtiani et al., 2018), uniform measures over simplices (Najafi et al., 2021), and other structured distributions, under both clean and noisy observations (Saberi et al., 2023; Ben-David et al., 2023).

Ashtiani et al. (Ashtiani et al., 2018) employed a sample-compression scheme to resolve a long-standing open problem, establishing nearly tight sample-complexity bounds of $\widetilde{\mathcal{O}}(kd^2/\epsilon^2)$ for learning arbitrary $d$-dimensional Gaussian mixture models with $k$ components up to total variation error $\epsilon$. Subsequent works (Najafi et al., 2021; Saberi et al., 2023) extended this approach to uniform distributions supported on high-dimensional simplices. In particular, Najafi et al. (Najafi et al., 2021) showed that simplices in $\mathbb{R}^d$ can be learned from $n \geq \mathcal{O}(d^2 \log d/\epsilon)$ samples up to total variation error $\epsilon$, while Saberi et al. (Saberi et al., 2025) established a lower bound of $\Omega(d/\epsilon)$ for this problem. Other works have generalized these ideas to structured distributions: Devroye et al. (Devroye et al., 2020) derived minimax learning rates for normal and Ising graphical models using techniques inspired by sample compression, and Turner et al. (Turner et al., 2021) proposed "coresets" in density estimation, a concept closely related to sample compression that captures structural properties of distributions to achieve similar guarantees. See also Ben et al. (Ben-David et al., 2023) for recent advances in privately learning distributions from public data using sample compression.

### B.1. Robust Density Estimation

A major line of research investigates distribution learning and estimation under data corruption, including additive independent noise models. The field of algorithmic robust statistics—encompassing robust mean and covariance estimation (Wainwright, 2019), robust regression, robust principal component analysis (PCA) (Candès et al., 2011), and robust density estimation—has advanced rapidly, yielding efficient algorithms that achieve near-optimal breakdown points and error rates even in high-dimensional regimes. In the nonparametric setting, an important line of work on learning from noisy samples arises in the context of deconvolution kernel density estimation. Delaigle (Delaigle, 2021) studied the classical independent additive noise model, later extended to cases with unknown noise distributions (Delaigle & Van Keilegom, 2021). Their analysis characterizes the asymptotic and minimax behavior of deconvolution estimators using Fourier-based techniques. However, this approach remains nonparametric and relies on broad regularity conditions such as *supersmoothness*, rather than exploiting problem-specific structural properties. The present work therefore complements this literature by emphasizing learnability through structural constraints such as sample compressibility.

**Learning from corrupted samples.** In practical scenarios, data are often corrupted by stochastic noise perturbations, making it essential to understand how such distortions affect learnability. While the supervised setting has been extensively

studied (see (Konstantinov et al., 2020) and references therein), with several impossibility results established (Charikar et al., 2017; Kearns & Li, 1988), progress in the unsupervised domain remains limited. Existing analyses either focus on broad nonparametric settings—often yielding loose minimax bounds due to weak structural assumptions (Tang & Yang, 2023; Wang & Marzouk, 2022)—or on narrowly defined parametric models such as Gaussian mixtures (Ashtiani et al., 2018), uniform measures over simplices (Saberi et al., 2023), or unimodal and bimodal distributions with specific tail behavior (Cesa-Bianchi et al., 1999; Hall et al., 2006). These approaches are inherently problem specific and rely on ad hoc techniques that do not generalize across distribution families. From the converse, or tightness, perspective, the situation is even more fragmented: proving optimality typically requires bespoke arguments tied to the structure of each individual model, with little prospect for a unified framework.

**Sample compression.** In a recent work, Saberi et al. (Saberi et al., 2023) employed a Fourier-based technique to show that, when samples are corrupted by isotropic Gaussian noise with variance $\sigma^2$, the sample complexity required to learn uniform measures over high-dimensional simplices in $\mathbb{R}^d$ scales as $\epsilon^{-2} e^{\mathcal{O}(d\sigma^2)}$. In a subsequent work, Saberi et al. (Saberi et al., 2025) proved that recovery from noise requires at least $n \geq \Omega(d^3\sigma^2/\epsilon^2)$. Their methodology is likewise grounded in sample compression. Notably, all existing studies addressing distribution recovery from noisy samples rely explicitly on structural assumptions—such as smoothness, isotropy, or related regularity conditions—on the distributions themselves or on their supports.

# C. Proofs of Section 2: Additive Noise Model

*Proof of Theorem 2.5.* First, we use Assumption 1.3 on $\mathcal{F}$, which guarantees the existence of a decoder $\mathcal{J}$ satisfying Assumption 2.1. Specifically, for any $(\epsilon, \delta) \in (0, 1)$, given $n$ i.i.d. and *clean* samples $\boldsymbol{X}_1, \ldots, \boldsymbol{X}_n$ from any $f^* \in \mathcal{F}$, the decoder outputs $\widehat{f} \in \mathcal{F}$ satisfying $\mathsf{TV}(f^*, \widehat{f}) \leq \epsilon$ with probability at least $1 - \delta/2$. The decoder $\mathcal{J}$ requires a sequence of at most $\tau(\epsilon)$ samples, at most $t(\epsilon)$ bits, and it must hold that $n \geq m(\epsilon)\log(2/\delta)$.

Throughout the proof, let $\mathbf{L} \in \mathcal{X}^{\tau(\epsilon)} \subseteq \mathbb{R}^{d\tau(\epsilon)}$ denote this sequence of samples and let $\mathbf{B} \in \{0, 1\}^{t(\epsilon)}$ represent the corresponding bit sequence. Importantly, sample compression does not require *knowing* $\mathbf{L}$ or $\mathbf{B}$ explicitly—only their *existence* is needed. The procedure for establishing learnability (Theorem A.5) ensures these sequences are found by considering all possibilities. Mathematically, we have:

$$\mathbb{P}\left(\mathsf{TV}\left(f^*, \mathcal{J}(\mathbf{L}, \mathbf{B})\right) \leq \epsilon\right) \geq 1 - \delta/2, \quad \forall f^* \in \mathcal{F}, \tag{14}$$

where the probability is taken over the randomness in generating clean samples $\boldsymbol{X}_1, \ldots, \boldsymbol{X}_n$. However, in our setting, we do not have access to clean samples; instead, we observe perturbed samples $\tilde{\boldsymbol{X}}_i = \boldsymbol{X}_i + \boldsymbol{\zeta}_i$ for all $i \in [n]$. Thus, instead of knowing $\mathbf{L}$, we only know the existence of $\mathbf{L}_\mathsf{N}$, defined as:

$$\mathbf{L}_\mathsf{N} \triangleq (\boldsymbol{X} + \boldsymbol{\zeta})_{\boldsymbol{X} \in \mathbf{L}}.$$

For $i \in [n]$ and $j \in [d]$, let $\zeta_{ij}$ denote the $j$th component of $\boldsymbol{\zeta}_i$. In order to do this, we try to approximate each $\zeta_{ij}$ by a member of a quantized grid: we quantize a symmetric interval in $\mathbb{R}$ and generate a finite set of points, denoted by $I \subset \mathbb{R}$ (with $|I| < \infty$), such that the following holds:

$$\mathbb{P}\left(\forall i, j \middle| \min_{\widehat{\zeta}_{ij} \in I} \left|\zeta_{ij} - \widehat{\zeta}_{ij}\right| \leq \eta\right) \geq 1 - \delta/2, \tag{15}$$

for some $\delta \in (0, 1)$ and $\eta \geq 0$, where $\mathbb{P}(\cdot)$ is with respect to the randomness of drawing noise vectors $\boldsymbol{\zeta}_i$. We then establish the following lemma:

**Lemma C.1.** *For any $\epsilon, \delta \in (0, 1)$, assume there exists a decoder such that* (14) *holds for at least one clean sample sequence* $\mathbf{L}$ *and a corresponding bit sequence* $\mathbf{B}$. *Also, assume there exists a grid $I$ satisfying* (15). *Then, there exist decoders* $\mathcal{J}_1, \ldots, \mathcal{J}_M$ *for $\mathcal{F}$ with $M = |I|^{d\tau(\epsilon)}$, such that:*

$$\mathbb{P}\left(\exists i \in [M] \middle| \mathsf{TV}\left(f^*, \mathcal{J}_i(\mathbf{L}_\mathsf{N}, \mathbf{B})\right) \leq \epsilon + \frac{1}{2}r\eta\sqrt{d\tau(\epsilon)}\right) \geq 1 - \delta. \tag{16}$$

*Proof.* There are $\tau(\epsilon)$ clean samples, each of dimension $d$, in $\mathbf{L}$. Let $S \subseteq [n]$ denote the indices of these samples. Since (15) holds, we know that with probability at least $1 - \delta/2$, there exists at least one combination of the grid points $\widehat{\zeta}_{ij}$ for

$i \in S, j \in [d]$ such that

$$\left| \tilde{X}_{ij} - \widehat{\zeta}_{ij} \right| = \left| X_{ij} + \zeta_{ij} - \widehat{\zeta}_{ij} \right| \leq \eta, \quad \forall i \in S, j \in [d].$$

There are $|I|$ possible ways to denoise each dimension, and with $d \times \tau(\epsilon)$ instances corresponding to different samples and dimensions, the total number of possible denoising configurations in $\mathbf{L_N}$ is given by $M = |I|^{d\tau(\epsilon)}$. Among these, at least one configuration results in a denoised sequence $\mathbf{L}'$ such that, with probability at least $1 - \delta/2$,

$$\|\mathbf{L} - \mathbf{L}'\|_\infty \leq \eta \implies \|\mathbf{L} - \mathbf{L}'\|_2 \leq \eta\sqrt{d\tau(\epsilon)}.$$

Using the Lipschitz continuity assumption (Assumption 2.1), we can then guarantee that

$$\mathsf{TV}\left(\mathcal{J}\left(\mathbf{L}, \mathbf{B}\right), \mathcal{J}\left(\mathbf{L}', \mathbf{B}\right)\right) \leq \frac{r}{2}\|\mathbf{L} - \mathbf{L}'\|_2 \leq \frac{1}{2}r\eta\sqrt{d\tau(\epsilon)}.$$

By combining the results from (14) and (15), applying a union bound over the probability of errors (i.e., $\delta/2 + \delta/2 = \delta$), and using the triangle inequality for total variation (TV) distance,

$$\mathsf{TV}\left(f^*, \mathcal{J}\left(\mathbf{L}', \mathbf{B}\right)\right) \leq \mathsf{TV}\left(f^*, \mathcal{J}\left(\mathbf{L}, \mathbf{B}\right)\right) + \mathsf{TV}\left(\mathcal{J}\left(\mathbf{L}, \mathbf{B}\right), \mathcal{J}\left(\mathbf{L}', \mathbf{B}\right)\right),$$

we obtain

$$\mathbb{P}\left(\mathsf{TV}\left(f^*, \mathcal{J}(\mathbf{L}', \mathbf{B})\right) \leq \epsilon + \frac{1}{2}r\eta\sqrt{d\tau(\epsilon)}\right) \geq 1 - \delta,$$

which completes the proof of the lemma. $\qquad\square$

The remaining task is to design a sufficiently fine grid that satisfies (15) and then use Lemma C.1 to integrate it into a new sample compression framework for $\mathcal{F} * G$. Let the grid of quantized points $I$ be given by $I = \{b_1, \ldots, b_K\}$, where $K$ denotes the size of the grid. Without loss of generality, assume $b_1 < b_2 < \cdots < b_K$. For simplicity, we assume the $b_k$ values are distributed such that each consecutive pair $b_k, b_{k+1}$ is spaced evenly. While more sophisticated grid designs exist, their impact on final sample complexity is negligible. Thus, we consider the following quantization format:

$$b_k - b_{k-1} = 2\eta, \quad \forall k \in \{2, ..., |I|\}, \tag{17}$$

$$b_{|I|} = -b_1 = \left| \Phi_G^{-1}\left(\frac{\delta}{4nd}\right) \right|. \tag{18}$$

Examining equation (18), we note that to maintain an overall quantization error probability of at most $\delta/2$, as required by (15), the error in each of the $d$ dimensions of each sample $i \in [n]$ must not exceed $\delta/(2nd)$. Consequently, the quantization range must cover:

$$\left(\Phi_G^{-1}\left(\frac{\delta}{4nd}\right), \Phi_G^{-1}\left(1 - \frac{\delta}{4nd}\right)\right),$$

which justifies the conditions in (18). Here, we have used two facts: i) the density of the noise vector has i.i.d. components, thus each dimension is distributed according to $\Phi_G$ independently, and ii) the CDF is symmetric w.r.t. origin. Both of these constraints can be removed, however, at the cost of introducing more complex notations.

The grid $I$ can be represented using $\log_2(|I|)$ bits, implying that we require

$$d\tau(\epsilon)\log_2\left(|I|\right) \leq d\tau(\epsilon)\log_2\left(1 + \frac{1}{\eta}\left|\Phi_G^{-1}\left(\frac{\delta}{4nd}\right)\right|\right)$$

bits to construct all $M$ possible augmented decoders in Lemma C.1. Additionally, we note that $n = m(\epsilon)\log(2/\delta)$. [1] This necessitates an additional set of $d\tau(\epsilon)\log_2(|I|)$ bits, denoted $\mathbf{B_{den}}$, to identify the chosen denoising scheme among the $M$ possibilities. Thus, the total number of required bits is $t(\epsilon) + d\tau(\epsilon)\log_2(|I|)$. So far, we have demonstrated that a sample compression scheme of

$$\left[\tau(\epsilon), t(\epsilon) + d\tau(\epsilon)\log_2\left(1 + \frac{1}{\eta}\left|\Phi_G^{-1}\left(\frac{\delta}{4dm(\epsilon)\log(2/\delta)}\right)\right|\right), m(\epsilon)\log(2/\delta)\right] \tag{19}$$

---

[1] Although at most $\tau(\epsilon)$ out of the total $n$ samples are used, the quantization criteria in (15) must hold for all $n$ samples, as we do not deterministically know which ones will form the chosen $\tau(\epsilon)$.

guarantees the existence of a decoder $\mathcal{J}_{\text{den}}$ that can process $m(\epsilon) \log(2/\delta)$ noisy samples from $f^* * G$ and output a density $\widehat{f} \in \mathcal{F}$ such that

$$\mathbb{P}\left(\mathsf{TV}(f^*, \widehat{f}) \leq \epsilon + \frac{1}{2}r\eta\sqrt{d\tau(\epsilon)}\right) \geq 1 - \delta. \tag{20}$$

There are multiple ways to *optimize* the tradeoff between $\eta$ and $\epsilon$ in this setting, depending on the specific choices of $\tau$, $t$, and $m$ functions, as well as the noise CDF $\Phi_G$. Since the paper already contains sufficient technical detail, we proceed with the following simplified approach to guarantee a final total variation error of at most $\epsilon$:

$$\left\{\epsilon \leftarrow \epsilon/2 \quad \text{and} \quad \frac{1}{2}r\eta\sqrt{d\tau(\epsilon/2)} \leftarrow \epsilon/2\right\} \quad \Longrightarrow \quad \eta \triangleq \frac{\epsilon}{r\sqrt{d\tau(\epsilon/2)}}, \tag{21}$$

This yields the claimed sample compression scheme already stated in the theorem, as follows:

$$\left[\tau\left(\frac{\epsilon}{2}\right), t\left(\frac{\epsilon}{2}\right) + d\tau\left(\frac{\epsilon}{2}\right)\log_2\left(1 + \frac{r\sqrt{d\tau\left(\epsilon/2\right)}}{\epsilon}\left|\Phi_G^{-1}\left(\frac{\delta}{4dm\left(\frac{\epsilon}{2}\right)\log\frac{2}{\delta}}\right)\right|\right), m\left(\frac{\epsilon}{2}\right)\log\frac{2}{\delta}\right],$$

to ensure that $\mathbb{P}(\mathsf{TV}(f^*, \widehat{f}) \leq \epsilon) \geq 1 - \delta$. The final step of the proof is to demonstrate that having $\mathsf{TV}(f^*, \widehat{f}) \leq \epsilon$ also guarantees $\mathsf{TV}(f^* * G, \widehat{f} * G) \leq \epsilon$.

**Lemma C.2.** *For any trio of probability densities $p, q, r \in L^2(\mathcal{X})$ supported over $\mathcal{X}$, we have*

$$\mathsf{TV}(p * r, q * r) \leq \mathsf{TV}(p, q).$$

*Proof.* Based on the definition of $d$-dimensional convolution, and the equivalence between TV distance and $\|\cdot\|_1/2$, we have

$$\begin{aligned}
\mathsf{TV}(p * r, q * r) &= \int_{\mathcal{X}} |(p * r)(\boldsymbol{t}) - (q * r)(\boldsymbol{t})|\, \mathrm{d}\boldsymbol{t} \\
&= \frac{1}{2}\int_{\mathcal{X}}\left|\int_{\mathcal{X}}(p(\boldsymbol{t}) - q(\boldsymbol{t}))\, r(\boldsymbol{u} - \boldsymbol{t})\mathrm{d}\boldsymbol{u}\right|\mathrm{d}\boldsymbol{t} \\
&\overset{(i)}{\leq} \frac{1}{2}\int_{\mathcal{X}}\int_{\mathcal{X}}|p(\boldsymbol{t}) - q(\boldsymbol{t})|\, r(\boldsymbol{u} - \boldsymbol{t})\mathrm{d}\boldsymbol{u}\mathrm{d}\boldsymbol{t} \\
&= \frac{1}{2}\int_{\mathcal{X}}|p(\boldsymbol{t}) - q(\boldsymbol{t})|\left(\int_{\mathcal{X}} r(\boldsymbol{u} - \boldsymbol{t})\mathrm{d}\boldsymbol{u}\right)\mathrm{d}\boldsymbol{t}
\end{aligned} \tag{22}$$

where (i) is due to triangle inequality. The final term equals to $\mathsf{TV}(p, q)$ since we have

$$\int_{\mathcal{X}} r(\boldsymbol{u} - \boldsymbol{t})\mathrm{d}\boldsymbol{u} = 1, \quad \forall \boldsymbol{t} \in \mathcal{X}.$$

This argument proves the bound. $\qquad\square$

We showed that using the claimed sample compression scheme, one can guarantee $\mathbb{P}(\mathsf{TV}(f^* * G, \widehat{f} * G) \leq \epsilon) \geq 1 - \delta$, which completes the proof. $\qquad\square$

*Proof of Theorem 2.10.* We use the result of Proposition 2.5, which assuming the $(\tau, t, m)$-sample compressibility of $\mathcal{F}$ guarantees the sample compressibility of $\mathcal{F} * G$. Specifically, for any $(\epsilon, \delta) \in (0, 1)$, $\mathcal{F} * G$ admits

$$\left[\tau\left(\frac{\epsilon}{2}\right), t\left(\frac{\epsilon}{2}\right) + d\tau\left(\frac{\epsilon}{2}\right)\log_2\left(1 + \frac{r\sqrt{d\tau\left(\epsilon/2\right)}}{\epsilon}\left|\Phi_G^{-1}\left(\frac{\delta}{4dm\left(\frac{\epsilon}{2}\right)\log\frac{2}{\delta}}\right)\right|\right), m\left(\frac{\epsilon}{2}\right)\log\frac{2}{\delta}\right]$$

-sample compression. Therefore, there exists a decoder $\mathcal{J}$ such that given $m\left(\frac{\epsilon}{2}\right)\log\frac{2}{\delta}$ i.i.d. samples from any $f^* * G \in \mathcal{F} * G$, the decoder outputs $\widehat{f} * G \in \mathcal{F} * G$ that satisfies $\mathsf{TV}(f^* * G, \widehat{f} * G) \leq \epsilon$ with probability at least $1 - \delta$.

The next step is to use a seminal proposition from (Ashtiani et al., 2018), which combines Theorem A.4 (originally from (Devroye & Lugosi, 2001)) and a number of concentration inequalities in order to prove the information-theoretic learnability of *sample compressible* distribution classes in general. The following theorem, which is full version of Theorem A.5 establishes a fundamental theoretical connection between $(\tau, t, m)$-sample compressibility and PAC-learnability of a distribution class:

**Theorem C.3** (Full version of Theorem 3.5 in (Ashtiani et al., 2018)). *Suppose $\mathcal{F}$ admits $(\tau, t, m)$-sample compression for some functions $\tau, t, m : (0, 1) \to \mathbb{N}$. For any $\epsilon, \delta \in (0, 1)$, let $\tau'(\epsilon) \triangleq \tau(\epsilon) + t(\epsilon)$. Also, define $N_{\tau,t,m}^{\mathsf{Clean}}$ as*

$$N_{\tau,t,m}^{\mathsf{Clean}}(\epsilon, \delta) \triangleq \mathcal{O}\left(m\left(\frac{\epsilon}{6}\right)\log_3\left(\frac{2}{\delta}\right) + \frac{32}{\epsilon^2}\left[\tau'\left(\frac{\epsilon}{6}\right)\log\left(m\left(\frac{\epsilon}{6}\right)\log_3\left(\frac{2}{\delta}\right)\right) + \log\left(\frac{6}{\delta}\right)\right]\right).$$

*Then, there exists a deterministic algorithm that by having $n \geq N_{\tau,t,m}^{\mathsf{Clean}}(\epsilon, \delta)$ i.i.d. samples from any unknown distribution $f^* \in \mathcal{F}$, outputs $\widehat{f} \in \mathcal{F}$ where we have $\mathsf{TV}(f^*, \widehat{f}) \leq \epsilon$, with probability of at least $1 - \delta$.*

Combining the results from Proposition 2.5 and Theorem C.3, one can deduce that there exists a deterministic algorithm, that for any $\epsilon, \delta \in (0, 1)$, upon having

$$n \geq N_{\tau,t,m}^{\mathsf{Clean}}(6\epsilon, \delta/2) \tag{23}$$
$$+ \mathcal{O}\left[\frac{2d\tau(\epsilon)}{9\epsilon^2}\log_2\left(\frac{r\sqrt{d\tau(\epsilon)}}{2\epsilon}\left|\Phi_G^{-1}\left(\frac{\delta}{8dm(\epsilon)\log\frac{4}{\delta}}\right)\right|\right)\log\left(m(\epsilon)\log_3\left(\frac{4}{\delta}\right)\right)\right]$$

i.i.d. samples from any $f^* * G \in \mathcal{F} * G$, outputs $\widehat{f} * G \in \mathcal{F} * G$ which is an $12\epsilon$-approximation of $f^* * G$ with probability at least $1 - \delta$. Mathematically, we have:

$$\mathbb{P}(\mathsf{TV}(f^* * G, \widehat{f} * G) \leq 12\epsilon) \geq 1 - \delta, \tag{24}$$

where $\mathbb{P}(\cdot)$ is with respect to the randomness of generating samples from $f^* * G$. The final step is to establish an upper bound for $\|\widehat{f} - f^*\|_2$ with probability at least $1 - \delta$, leveraging (24). As discussed in Section 2, such a relationship does not necessarily hold, since convolution with *low-frequency* noise densities (e.g., Gaussian noise) attenuates high-frequency components of the density difference $\widehat{f} - f^*$. Consequently, ensuring that $\mathsf{TV}(f^* * G, \widehat{f} * G)$ is small does not directly imply that $\widehat{f}$ is close to $f^*$ in a general sense.

However, under Assumption 2.7, we have assumed that density differences in $\mathcal{F}$ retain a non-negligible portion of their energy in low-frequency regions. This ensures that recovering $f^* * G$ also leads to recovering $f^*$ itself. The following lemma formally establishes this fact:

**Lemma C.4.** *For any $\epsilon, \delta \in (0, 1)$, assume that there exists an algorithm such that (24) holds for the output distribution $\widehat{f}$. Also, assume that Assumption 2.7 holds for a set of pairs $\mathsf{P}(\mathcal{F}) = \{(\alpha, \xi)\}$. Then, with probability of at least $1 - \delta$, the following bound holds uniformly for all $(\alpha, \xi) \in \mathsf{P}(\mathcal{F})$:*

$$\|\widehat{f} - f^*\|_2 \leq \frac{24\epsilon}{\sqrt{B_G(\alpha)(1 - \xi)}}. \tag{25}$$

*Proof.* Based on (24) and the equivalence between TV distance and $\|\cdot\|_1 / 2$, with probability at least $1 - \delta$, we have:

$$\int_{\mathbb{R}^d}\left|f^* * G(\boldsymbol{t}) - \widehat{f} * G(\boldsymbol{t})\right| \mathrm{d}\boldsymbol{t} \leq 24\epsilon. \tag{26}$$

We invoke the Hausdorff–Young inequality, which we state as a lemma.

**Lemma C.5** (Hausdorff–Young inequality). *Recall the multidimensional Fourier transform defined in (1). If $1 \leq p \leq 2$ and $p'$ denotes the conjugate exponent with $\frac{1}{p} + \frac{1}{p'} = 1$, then for any $f \in L^p(\mathbb{R}^d)$ one has*

$$(2\pi)^{-\frac{d}{p'}}\left(\int_{\mathbb{R}^d}|\mathsf{F}\{f\}(\boldsymbol{\omega})|^{p'}\mathrm{d}\boldsymbol{\omega}\right)^{1/p'} \leq \left(\int_{\mathbb{R}^d}|f(\boldsymbol{x})|^p\mathrm{d}\boldsymbol{x}\right)^{1/p}.$$

Let $P$ and $Q$ denote the Fourier transforms of $f^*$ and $\widehat{f}$, respectively. Using the fact that convolution operator turns into point-wise multiplication in the Fourier domain, and also applying Lemma C.5, we obtain the following inequalities for any $\alpha \geq 0$:

$$24\epsilon \geq \int_{\mathbb{R}^d} \left| f^* * G(\boldsymbol{t}) - \widehat{f} * G(\boldsymbol{t}) \right| \mathrm{d}\boldsymbol{t}$$

$$\overset{(i)}{\geq} \sup_{\boldsymbol{\omega} \in \mathbb{R}^d} |P(\boldsymbol{\omega})G(\boldsymbol{\omega}) - Q(\boldsymbol{\omega})G(\boldsymbol{\omega})| \tag{27}$$

where (i) is according to Lemma C.5 by setting $p = 1$ and $p' = \infty$. Therefore, we can obtain the following relations:

$$|P(\boldsymbol{\omega}) - Q(\boldsymbol{\omega})| \leq \left( \inf_{\|\boldsymbol{\omega}\|_2 \leq \alpha} |G(\boldsymbol{\omega})| \right)^{-1} |P(\boldsymbol{\omega})G(\boldsymbol{\omega}) - Q(\boldsymbol{\omega})G(\boldsymbol{\omega})|$$

$$\leq \left( \inf_{\|\boldsymbol{\omega}\|_2 \leq \alpha} |G(\boldsymbol{\omega})| \right)^{-1} \sup_{\boldsymbol{\omega} \in \mathbb{R}^d} |P(\boldsymbol{\omega})G(\boldsymbol{\omega}) - Q(\boldsymbol{\omega})G(\boldsymbol{\omega})|$$

$$\overset{(i)}{\leq} \frac{24\epsilon}{\inf_{\|\boldsymbol{\omega}\|_2 \leq \alpha} |G(\boldsymbol{\omega})|} \tag{28}$$

where (i) is due to (27). Then, by defining $\mathbb{B}_2^d(\alpha) = \{\boldsymbol{\omega} | \|\boldsymbol{\omega}\| \leq \alpha\}$, the following bound has been achieved:

$$\int_{\|\boldsymbol{\omega}\|_2 \leq \alpha} |P(\boldsymbol{\omega}) - Q(\boldsymbol{\omega})|^2 \, \mathrm{d}\boldsymbol{\omega} \leq \int_{\|\boldsymbol{\omega}\|_2 \leq \alpha} \frac{576\epsilon^2}{\left( \inf_{\|\boldsymbol{\omega}'\|_2 \leq \alpha} |G(\boldsymbol{\omega}')| \right)^2} \mathrm{d}\boldsymbol{\omega}$$

$$= \frac{576\epsilon^2}{\left( \inf_{\|\boldsymbol{\omega}\|_2 \leq \alpha} |G(\boldsymbol{\omega})| \right)^2} \mathsf{Vol}(\mathbb{B}_2^d(\alpha))$$

$$= \frac{576\epsilon^2}{\inf_{\|\boldsymbol{\omega}\|_2 \leq \alpha} |G(\boldsymbol{\omega})|^2} \mathsf{Vol}(\mathbb{B}_2^d(\alpha)) = \frac{576\epsilon^2}{B_G(\alpha)}. \tag{29}$$

where Vol is the $d$-dimensional volume (i.e., Lebesgue measure) of a set. Also, note that $\mathsf{Vol}(\mathbb{B}_2^d(\alpha))$ can be written as $\mathsf{Vol}(\mathbb{B}_2^d(1))\alpha^d$. What remains is to use Assumption 2.7. Based on the definition of $\mathsf{P}(\mathcal{F})$, for any pair $(\alpha, \xi) \in \mathsf{P}(\mathcal{F})$ (where $\alpha \geq 0$ and $\xi < 1$), we have

$$\int_{\|\boldsymbol{\omega}\|_2 \geq \alpha} |P(\boldsymbol{\omega}) - Q(\boldsymbol{\omega})|^2 \, \mathrm{d}\boldsymbol{\omega} \leq \xi \|f^* - \widehat{f}\|_2^2.$$

Therefore, the following set of relations hold:

$$\|\widehat{f} - f^*\|_2^2 = \int_{\mathbb{R}^d} \left| f^*(\boldsymbol{t}) - \widehat{f}(\boldsymbol{t}) \right|^2 \mathrm{d}\boldsymbol{t}$$

$$= \int_{\mathbb{R}^d} |P(\boldsymbol{\omega}) - Q(\boldsymbol{\omega})|^2 \, \mathrm{d}\boldsymbol{\omega}$$

$$= \int_{\|\boldsymbol{\omega}\|_2 \leq \alpha} |P(\boldsymbol{\omega}) - Q(\boldsymbol{\omega})|^2 \, \mathrm{d}\boldsymbol{\omega} + \int_{\|\boldsymbol{\omega}\|_2 \geq \alpha} |P(\boldsymbol{\omega}) - Q(\boldsymbol{\omega})|^2 \, \mathrm{d}\boldsymbol{\omega}$$

$$\overset{(i)}{\leq} \frac{576\epsilon^2}{B_G(\alpha)} + \int_{\|\boldsymbol{\omega}\|_2^{\geq} \alpha} |P(\boldsymbol{\omega}) - Q(\boldsymbol{\omega})|^2 \, \mathrm{d}\boldsymbol{\omega}$$

$$\overset{(ii)}{\leq} \frac{576\epsilon^2}{B_G(\alpha)} + \xi \|\widehat{f} - f^*\|_2^2 \tag{30}$$

where (i) holds due to (29), and (ii) holds owing to Assumption 2.7. Note that $\xi$ can itself be a function of $\|\widehat{f} - f^*\|_2$. This argument proves the bound. $\qquad \square$

Using Lemma C.4 and some simple algebra, it has been shown that we can guarantee the existence of a deterministic algorithm that by using $n$ i.i.d. samples from $f^* * G$, outputs $\widehat{f}$ which holds in the following inequality:

$$\|\widehat{f} - f^*\|_2 \leq \frac{24\epsilon}{\sqrt{B_G(\alpha)(1 - \xi)}}, \tag{31}$$

with probability at least $1 - \delta$, uniformly over all pairs $(\alpha, \xi) \in \mathsf{P}(\mathcal{F})$. Therefore, the bound also holds for the infimum, i.e.,

$$\mathbb{P}\left(\|\widehat{f} - f^*\|_2 \leq \inf_{(\alpha, \xi) \in \mathsf{P}(\mathcal{F})} \frac{24\epsilon}{\sqrt{B_G(\alpha)(1 - \xi)}}\right) \geq 1 - \delta. \tag{32}$$

In case $\xi$ is not a constant, and instead is a function of $\|\widehat{f} - f^*\|_2$, the bound turns into the following probabilistic inequality:

$$\mathbb{P}\left(\|\widehat{f} - f^*\|_2 \sqrt{1 - \xi\left(\|\widehat{f} - f^*\|_2\right)} \leq \frac{24\epsilon}{\sqrt{B_G(\alpha)}} \,\middle|\, \forall(\alpha, \xi) \in \mathsf{P}(\mathcal{F})\right) \geq 1 - \delta, \tag{33}$$

which completes the proof. $\qquad\square$

*Proof of Corollary 2.11.* There are two major components in both the sample complexity and the final $\ell_2$ error that depend on the noise distribution ($G$): the component-wise noise CDF $\Phi_G(\cdot)$ and the quantity $B_G(\cdot)$. We first determine the order of the sample complexity for both the Gaussian and Laplace cases by substituting their respective exact formulations and proving the claimed expressions.

**Gaussian Noise**: According to Remark 2.6, we have $\left|\Phi_G^{-1}(\Delta)\right| = \sigma\mathcal{O}\left(\sqrt{\log \Delta^{-1}}\right)$, for $\Delta < 1/\sqrt{2\pi}$. Substituting $\Delta = \frac{\delta}{8dm(\epsilon)\log\frac{4}{\delta}}$, we obtain

$$\left|\Phi_G^{-1}\left(\frac{\delta}{8dm(\epsilon)\log\frac{4}{\delta}}\right)\right| = \sigma\mathcal{O}\left(\sqrt{\log\left(\frac{dm(\epsilon)}{\delta}\log\frac{1}{\delta}\right)}\right) = \sigma\mathcal{O}\left(\sqrt{\log\frac{dm(\epsilon)}{\delta}}\right).$$

**Laplace Noise**: Again, based on Remark 2.6, we have $\left|\Phi_G^{-1}(\Delta)\right| = b\mathcal{O}\left(\log \Delta^{-1}\right)$. Substituting $\Delta = \frac{\delta}{8dm(\epsilon)\log\frac{4}{\delta}}$, this results in

$$\left|\Phi_G^{-1}\left(\frac{\delta}{8dm(\epsilon)\log\frac{4}{\delta}}\right)\right| = b\mathcal{O}\left(\log\left(\frac{dm(\epsilon)}{\delta}\log\frac{1}{\delta}\right)\right) = b\mathcal{O}\left(\log\frac{dm(\epsilon)}{\delta}\right).$$

Substituting these results into the sample complexity expression in Theorem 2.10, and setting $\lambda = \sigma$ for $G = \mathcal{N}(\mathbf{0}, \sigma^2 \mathbf{I}_d)$ and $\lambda = b$ for Laplace noise, we obtain the claimed order-wise sample complexity. Next, we derive explicit formulations for the final $\ell_2$ errors $\|f^* - \widehat{f}\|_2$ in both scenarios.

**Gaussian Noise**: Using the Fourier transform of an isotropic Gaussian density, we have

$$\mathsf{F}\{G\}(\boldsymbol{\omega}) = \mathsf{F}\left\{\frac{1}{(2\pi\sigma^2)^{d/2}} e^{-\|\boldsymbol{\varsigma}\|_2^2/2}\right\}(\boldsymbol{\omega}) = e^{-\sigma^2\|\boldsymbol{\omega}\|_2^2/2}. \tag{34}$$

Thus, the explicit formulation for $B_G(\cdot)$ is given by

$$B_G(\alpha) \triangleq \mathsf{Vol}^{-1}(\mathbb{B}_2^d(1))\alpha^{-d}\inf_{\|\boldsymbol{\omega}\|_2 \leq \alpha}|\mathsf{F}\{G\}(\boldsymbol{\omega})| = \mathsf{Vol}^{-1}(\mathbb{B}_2^d(1))\alpha^{-d}e^{-(\sigma\alpha)^2/2}. \tag{35}$$

Substituting this into the final result of Theorem 2.10 yields the result (i) in Corollary 2.11.

**Laplace Noise**: Following a similar approach for multivariate Laplace noise with independent and identically distributed components, and using the fact that the Fourier transform factorizes over independent dimensions, we derive the following formulation for $B_G(\cdot)$:

$$\begin{aligned}
B_G(\alpha) &\triangleq \mathsf{Vol}^{-1}(\mathbb{B}_2^d(1))\alpha^{-d}\inf_{\|\boldsymbol{\omega}\|_2 \leq \alpha}|\mathsf{F}\{G\}(\boldsymbol{\omega})| \\
&\overset{(i)}{=} \mathsf{Vol}^{-1}(\mathbb{B}_2^d(1))\alpha^{-d}\inf_{\|\boldsymbol{\omega}\|_2 \leq \alpha}\prod_{i=1}^{d}\frac{1}{1 + b^2\omega_i^2} \\
&\overset{(ii)}{=} \mathsf{Vol}^{-1}(\mathbb{B}_2^d(1))\alpha^{-d}\left(1 + \frac{(b\alpha)^2}{d}\right)^{-d},
\end{aligned} \tag{36}$$

where (i) follows from the fact that

$$\left| \mathsf{F}\left\{ G(\boldsymbol{\zeta}) = (2b)^{-d} \prod_{i=1}^{d} e^{-|\zeta_i|/b} \right\}(\boldsymbol{\omega}) \right| = \prod_{i=1}^{d} \frac{1}{1 + b^2 \zeta_i^2},$$

and (ii) holds since the infimum is attained when $\zeta_i^2 = \alpha^2/d$ for each $i \in [d]$. This completes the proof. $\qquad\square$

*Proof of Proposition 2.13.* We seek to upper bound the optimal value of the following functional optimization problem:

$$\sup_{f \in L^2(\mathcal{X})} \int |f| \quad \text{subject to} \quad f(\boldsymbol{x}) \leq g(\boldsymbol{x}), \ \forall \boldsymbol{x} \in \mathcal{X}, \quad \text{and} \quad \int f^2 = \varepsilon^2, \tag{37}$$

where $\varepsilon \triangleq \|f\|_2$ is assumed to be fixed and given. Since the objective is symmetric in $f$, the supremum is achieved when $f \geq 0$. Thus, without loss of generality, we may restrict the feasible set to non-negative functions. The problem becomes:

$$\inf_{f \in L^2(\mathcal{X})} - \int f \quad \text{subject to} \quad 0 \leq f(\boldsymbol{x}) \leq g(\boldsymbol{x}), \ \forall \boldsymbol{x} \in \mathcal{X}, \quad \text{and} \quad \int f^2 \leq \varepsilon^2. \tag{38}$$

This is a convex optimization problem: the objective is linear, the inequality constraints are convex (box constraints), and the quadratic equality constraint defines a convex level set. Hence, by standard results in convex analysis (e.g., Slater's condition), strong duality holds, and the optimal value can be characterized via the Karush–Kuhn–Tucker (KKT) conditions (Boyd & Vandenberghe, 2004). We define the Lagrangian functional as follows:

$$\mathcal{L}(f, \lambda_1, \lambda_2, \nu) \triangleq - \int f - \int \lambda_1 f + \int \lambda_2 (f - g) + \nu \left( \int f^2 - \varepsilon^2 \right), \quad \forall f \in L^2(\mathcal{X}), \tag{39}$$

where $\lambda_1(\boldsymbol{x}), \lambda_2(\boldsymbol{x}) \geq 0$ are dual variables for the pointwise lower and upper bound constraints respectively, and $\nu \geq 0$ is the dual variable for the quadratic constraint. The stationarity condition with respect to $f$ yields:

$$\nabla_f \mathcal{L}(\boldsymbol{x}) = -1 - \lambda_1(\boldsymbol{x}) + \lambda_2(\boldsymbol{x}) + 2\nu f(\boldsymbol{x}) = 0, \quad \forall \boldsymbol{x} \in \mathcal{X}, \tag{40}$$

which leads to the expression for the optimizer:

$$\text{(i)} \quad f^*(\boldsymbol{x}) = \frac{1 + \lambda_1(\boldsymbol{x}) - \lambda_2(\boldsymbol{x})}{2\nu}. \tag{41}$$

The KKT conditions also include:

$$\text{(ii)} \quad \lambda_1(\boldsymbol{x}) \geq 0, \quad \lambda_2(\boldsymbol{x}) \geq 0, \quad \nu \geq 0$$
$$\text{(iii)} \quad \lambda_1(\boldsymbol{x}) f^*(\boldsymbol{x}) = 0, \quad \lambda_2(\boldsymbol{x})(f^*(\boldsymbol{x}) - g(\boldsymbol{x})) = 0$$
$$\text{(iv)} \quad \nu \left( \int_{\mathcal{X}} f^{*2}(\boldsymbol{x}) \mathrm{d}\boldsymbol{x} - \varepsilon^2 \right) = 0. \tag{42}$$

From these conditions, we can deduce the following structural properties of $f^*$: a) On the set where $f^*(\boldsymbol{x}) < g(\boldsymbol{x})$, the upper bound is inactive, so $\lambda_2(\boldsymbol{x}) = 0$, implying $f^*(\boldsymbol{x}) = \frac{1+\lambda_1(\boldsymbol{x})}{2\nu}$. But by complementary slackness, if $f^*(\boldsymbol{x}) > 0$, then $\lambda_1(\boldsymbol{x}) = 0$, and thus $f^*(\boldsymbol{x}) = \frac{1}{2\nu}$. b) On the set where $f^*(\boldsymbol{x}) = g(\boldsymbol{x})$, the upper bound is active, so $\lambda_2(\boldsymbol{x})$ may be non-zero.

Thus, the optimal solution $f^*$ takes the value $\min\left\{ \frac{1}{2\nu}, g(\boldsymbol{x}) \right\}$ almost everywhere. This is equivalent to a "water-filling" procedure: fill the region under $g$ until the $\ell_2$ norm constraint $\|f^*\|_2 = \varepsilon$ is met. Therefore, the function $f^*$ is the outcome of water-filling the subgraph of $g$ up to a certain level which satisfies the energy constraint. The total mass (i.e., $\ell_1$ norm) of $f^*$ is maximized under this procedure, yielding the bound stated in the proposition and the proof is complete. $\qquad\square$

*Proof of Corollary 2.14.* We consider two cases: i) when $\text{supp}(f^* - \widehat{f}) \subseteq [-R, R]^d$, the result follows directly from the Cauchy–Schwarz inequality:

$$\text{TV}(\widehat{f}, f^*)^2 = \left( \int_{\mathcal{X}} |f^*(\boldsymbol{x}) - \widehat{f}(\boldsymbol{x})| \, d\boldsymbol{x} \right)^2 \leq \left( \int_{\mathcal{X}} (f^*(\boldsymbol{x}) - \widehat{f}(\boldsymbol{x}))^2 \, d\boldsymbol{x} \right) \cdot \left( \int_{[-R,R]^d} 1 \, d\boldsymbol{x} \right)$$

$$= (2R)^d \|\widehat{f} - f^*\|_2^2. \tag{43}$$

This establishes the bound in (11).

ii) Consider the case where $f^*$ is upper bounded by a Gaussian envelope, i.e., $g(\boldsymbol{x}) \triangleq C_1 \exp\left(-\gamma \|\boldsymbol{x}\|_2^2\right)$ for all $\boldsymbol{x} \in \mathcal{X} \subseteq \mathbb{R}^d$. Without loss of generality, assume the center of the Gaussian is at the origin (i.e., $\boldsymbol{\mu} = 0$ for the particular $f$ under consideration). According to Proposition 2.13, the $\ell_1$-optimal function $f^*$ under the $\ell_2$ constraint $\|f^* - \widehat{f}\|_2 = \varepsilon$ is obtained by truncating $g$ at height $1/(2\nu)$ to form the function $f^*(\boldsymbol{x}) = \min\left\{\frac{1}{2\nu}, g(\boldsymbol{x})\right\}$. This yields a superlevel set $A(\varepsilon, g) \triangleq \{\boldsymbol{x} \in \mathcal{X} : g(\boldsymbol{x}) \geq \frac{1}{2\nu}\}$ which, due to radial symmetry, is a Euclidean ball centered at the origin with radius $R$. The radius $R$ must satisfy the constraint on the $\ell_2$ norm:

$$R^d \text{Vol}\left(\mathbb{B}_2^d(1)\right) + C_1^2 \text{Vol}\left(\mathbb{B}_2^{d-1}(1)\right) \int_R^{\infty} r^{d-1} e^{-2\gamma r^2} \, dr = \varepsilon^2, \tag{44}$$

where $\mathbb{B}_2^d(1)$ denotes the unit $\ell_2$-ball in $\mathbb{R}^d$, and its volume is given by

$$\text{Vol}\left(\mathbb{B}_2^d(1)\right) = \frac{\pi^{d/2}}{\Gamma\left(\frac{d}{2} + 1\right)}.$$

The integral in (44) involves the *incomplete gamma function* and, in general, does not admit a closed-form solution. However, from standard asymptotics for the tail of the Gaussian integral (see, e.g., (Temme, 1996)), one can show:

$$R \geq \mathcal{O}\left(\sqrt{\frac{1}{\gamma} \log \frac{1}{\varepsilon}}\right),$$

where the hidden constant depends on $C_1$ and the dimension $d$. Substituting this lower bound on $R$ into the expression for the total variation bound given by Proposition 2.13, we conclude the desired result and completes the proof. □

### C.1. Proofs of Claims: Part I

*Proof of claim 2.2.* The Gaussian family in general, and the isotropic axis-aligned Gaussian family in particular, are known to be sample compressible (Ashtiani et al., 2018). It has been shown that the following decoder $\mathcal{J}$ can achieve the above-mentioned sample compression scheme: The decoder $\mathcal{J}$ maps a sample sequence $\mathbf{L} = \{\boldsymbol{X}_1, ..., \boldsymbol{X}_{\tau(\epsilon)}\}$ and bits $\boldsymbol{B}$ to a Gaussian distribution $\mathcal{N}(\widehat{\boldsymbol{\mu}}, \widehat{\sigma}^2 \boldsymbol{I}_d)$, where

$$\widehat{\boldsymbol{\mu}} = \frac{1}{\tau(\epsilon)} \sum_{i=1}^{\tau(\epsilon)} \boldsymbol{X}_i$$

and

$$\widehat{\sigma}^2 = \max\left\{\sigma_0^2, \frac{1}{\tau(\epsilon)} \sum_{i=1}^{\tau(\epsilon)} \|\boldsymbol{X}_i - \widehat{\boldsymbol{\mu}}\|^2\right\}.$$

In this regard, let $\mathbf{L}$ and $\mathbf{L}'$ be two sample sequences with $\|\mathbf{L} - \mathbf{L}'\|_2 \leq \Delta$. Without loss of generality, for each sample

$X_i \in \mathbf{L}$, let $X_i' \in \mathbf{L}'$ satisfy $\left\| X_i - X_i' \right\|_2 \le \delta_i$, where $\sum_{i=1}^{\tau(\epsilon)} \delta_i^2 = \Delta^2$. Then, the difference in means is bounded by:

$$
\begin{aligned}
\left\| \widehat{\boldsymbol{\mu}} - \widehat{\boldsymbol{\mu}}' \right\|_2 &\le \frac{1}{\tau(\epsilon)} \sum_{i=1}^{\tau(\epsilon)} \left\| X_i - X_i' \right\|_2 \\
&\le \sqrt{ \frac{1}{\tau(\epsilon)} \sum_{i=1}^{\tau(\epsilon)} \left\| X_i - X_i' \right\|_2^2 } \\
&= \frac{1}{\sqrt{\tau(\epsilon)}} \left\| \mathbf{L} - \mathbf{L}' \right\|_2 \\
&\le \frac{\Delta}{\sqrt{\tau(\epsilon)}}.
\end{aligned}
\tag{45}
$$

On the other hand, the difference between true and empirical component-wise variances, i.e., $\sigma^2$ and $\widehat{\sigma}^2$, satisfies:

$$
\left| \widehat{\sigma}^2 - (\widehat{\sigma}')^2 \right| \le \left| \frac{1}{\tau(\epsilon)} \sum_{i=1}^{\tau(\epsilon)} \left( \left\| X_i - \widehat{\boldsymbol{\mu}} \right\|_2^2 - \left\| X_i' - \widehat{\boldsymbol{\mu}}' \right\|_2^2 \right) \right|,
\tag{46}
$$

where $\left\| X_i - \widehat{\boldsymbol{\mu}} \right\|_2^2 - \left\| X_i' - \widehat{\boldsymbol{\mu}}' \right\|_2^2 = \left\| X_i' - X_i \right\|_2^2 + \left\| \widehat{\boldsymbol{\mu}} - \widehat{\boldsymbol{\mu}}' \right\|_2^2$. As a result we have:

$$
\begin{aligned}
\left| \widehat{\sigma}^2 - (\widehat{\sigma}')^2 \right| &\le \left| \frac{1}{\tau(\epsilon)} \sum_{i=1}^{\tau(\epsilon)} \left( \left\| X_i' - X_i \right\|_2^2 + \left\| \widehat{\boldsymbol{\mu}} - \widehat{\boldsymbol{\mu}}' \right\|_2^2 \right) \right| \\
&\le \frac{\Delta^2}{\tau(\epsilon)} + \frac{\Delta^2}{\tau(\epsilon)^2} \\
&\le \mathcal{O} \left( \frac{\Delta^2}{\tau(\epsilon)} \right).
\end{aligned}
\tag{47}
$$

For two isotropic Gaussian distributions $\mathcal{N}\left( \boldsymbol{\mu}, \sigma^2 \mathbf{I}_d \right)$ and $\mathcal{N}\left( \boldsymbol{\mu}', (\sigma')^2 \mathbf{I}_d \right)$, the TV distance is known to obey the following upper-bound

$$
\mathsf{TV}\left( \mathcal{N}(\boldsymbol{\mu}, \sigma^2 \mathbf{I}_d), \mathcal{N}(\boldsymbol{\mu}', (\sigma')^2 \mathbf{I}_d) \right) \le \frac{1}{2} \left( \frac{\left\| \mu - \mu' \right\|_2}{\min\{\sigma, \sigma'\}} + \frac{\left| \sigma^2 - (\sigma')^2 \right|}{\min\{\sigma^2, (\sigma')^2\}} \right).
\tag{48}
$$

Substituting into the above results, we have

$$
\begin{aligned}
\mathsf{TV}\left( \mathcal{J}(\mathbf{L}, \mathbf{B}), \mathcal{J}(\mathbf{L}', \mathbf{B}) \right) &\le \frac{\Delta}{2\sigma_0 \sqrt{\tau(\epsilon)}} + \mathcal{O} \left( \frac{\Delta^2}{\tau(\epsilon)} \right) \\
&\le \mathcal{O} \left( \frac{1}{\sigma_0 \sqrt{\tau(\epsilon)}} \right) \Delta
\end{aligned}
\tag{49}
$$

On the other hand, according to (Ashtiani et al., 2018), for Gaussian distribution family we have $\tau(\epsilon) = \min\{2, \mathcal{O}(d \log(2d))\}$, which gives us the following result:

$$
\mathsf{TV}\left( \mathcal{J}(\mathbf{L}, \mathbf{B}), \mathcal{J}(\mathbf{L}', \mathbf{B}) \right) \le \mathcal{O} \left( \frac{1}{\sigma_0 \sqrt{d \log(2d)}} \right) \left\| \mathbf{L} - \mathbf{L}' \right\|_2,
\tag{50}
$$

and the proof is complete. $\qquad\square$

*Proof of claim 2.3.* Consider the following decoder for this class of distributions: The decoder $\mathcal{J}$, given a sample sequence $\mathbf{L} = \{\boldsymbol{X}_1, ..., \boldsymbol{X}_{\tau(\epsilon)}\}$, computes the empirical *minimum* and *maximum* of the samples alongside each dimension $i \in [d]$ as follows:

$$\widehat{a}_i = \min_{j \in [\tau(\epsilon)]} \boldsymbol{X}_{j,i} \quad , \quad \widehat{b}_i = \max_{j \in [\tau(\epsilon)]} \boldsymbol{X}_{j,i}, \tag{51}$$

where $\boldsymbol{X}_{j,i}$ is the $i$th dimension of $\boldsymbol{X}_j$. Then, $\mathcal{J}$ outputs the following uniform distribution:

$$\text{Uniform}\left( \prod_i \left[\widehat{a}_i, \widehat{b}_i\right] \right),$$

which is another uniform measure over axis-aligned hyper-rectangles and thus belongs to $\mathcal{F}$. A simple analysis can reveal that this decoder achieves a $(\tau, t, m)$-sample compression scheme for $\mathcal{F}$. However, exact knowledge of the specific functions $\tau, t, m : (0, 1) \to \mathbb{N}$ is not needed for this proof. We compute them later, in Section 3.

Similar to the proof of Claim 2.2, let $\mathbf{L}$ and $\mathbf{L}'$ be two sample sequences with $\|\mathbf{L} - \mathbf{L}'\|_2 = \Delta$. Without loss of generality, for each sample $\boldsymbol{X}_i \in \mathbf{L}$, let $\boldsymbol{X}'_i \in \mathbf{L}'$ satisfy $\|\boldsymbol{X}_i - \boldsymbol{X}'_i\|_2 \leq \delta_i$, where $\sum_{i=1}^{\tau(\epsilon)} \delta_i^2 = \Delta^2$. Hence, for each sample $j \in [\tau(\epsilon)]$ and dimension $i \in [d]$, the perturbation satisfies $|\boldsymbol{X}_{j,i} - \boldsymbol{X}'_{j,i}| \leq \Delta$. It should be noted that much tighter bounds can be attained here using a more detailed analysis, however, we have sacrificed tightness for the sake of brevity and readability of the proof.

The empirical min/max values corresponding to dimension $i$ in $\mathbf{L}$ and $\mathbf{L}'$ satisfy

$$|\widehat{a}_i - \widehat{a}'_i| \leq \Delta \quad , \quad \left|\widehat{b}_i - \widehat{b}'_i\right| \leq \Delta. \tag{52}$$

Thus, the perturbed hyper-rectangle $\prod_i \left[\widehat{a}'_i, \widehat{b}'_i\right]$ differs from the original by at most $\Delta$ in each endpoint of the boundary. For two uniform distributions

$$U = \text{Uniform}\left( \prod_i \left[\widehat{a}_i, \widehat{b}_i\right] \right) \quad \text{and} \quad U' = \text{Uniform}\left( \prod_i \left[\widehat{a}'_i, \widehat{b}'_i\right] \right),$$

the TV distance is bounded as

$$\text{TV}(U, U') \leq 1 - \frac{\text{Vol}_d\big(\text{supp}(U) \cap \text{supp}(U')\big)}{\text{Vol}_d\big(\text{supp}(U) \cup \text{supp}(U')\big)}, \tag{53}$$

where $\text{supp}(\cdot)$ denotes the support of a distribution, and $\text{Vol}_d$ is the $d$-dimensional volume (i.e., Lebesgue measure) of a set. The symmetric difference in each dimension contributes additively. Hence, for a single dimension $i \in [d]$, the overlap loss is bounded by:

$$1 - \prod_{i=1}^d \left( \frac{\min\left(\widehat{b}_i, \widehat{b}'_i\right) - \max\left(\widehat{a}_i, \widehat{a}'_i\right)}{\max\left(\widehat{b}_i, \widehat{b}'_i\right) - \min\left(\widehat{a}_i, \widehat{a}'_i\right)} \right) \leq 1 - \left(1 - \frac{2\Delta}{T/2}\right)^d$$

$$\leq \frac{4d\Delta}{T}, \tag{54}$$

where we have used the fact that having assumed the samples in $\mathbf{L}$ have achieved the total variation of at most $< 1/2$, we have $\widehat{b}_i - \widehat{a}_i \geq (b_i - a_i)/2 \geq T/2$ for each $i \in [d]$. Therefore, we have

$$\text{TV}(U, U') \leq \frac{4d\Delta}{T} = \frac{8d}{T} \times \frac{1}{2} \|\mathbf{L} - \mathbf{L}'\|_2, \tag{55}$$

which completes the proof. $\square$

*Proof of Claim 2.4.* We use the proof of Lemma 4.6 in (Ashtiani et al., 2020). Assume the class $\mathcal{F}$ admits a sample compression scheme using a decoder $\mathcal{J}$ that adheres to Assumption 2.1. For each $i \in [k]$, the chosen sequences $\mathbf{L}$ and

**B** should contain subsets $\mathbf{L}_i$ and $\mathbf{B}_i$, which are used to estimate $f_i$ using decoder $\mathcal{J}$. In this regard, we can define the $k$-mixture decoder $\mathcal{J}_k$ for class $k-\mathrm{Mix}(\mathcal{F})$ as follows:

$$\mathcal{J}_k(\mathbf{L}, \mathbf{B}) = \sum_{i=1}^{k} \widehat{\alpha}_i \mathcal{J}(\mathbf{L}_i, \mathbf{B}_i) \tag{56}$$

where $\widehat{\alpha}_i$ are the quantized weights (approximations of the true latent $\alpha_i$s) which are decoded using the bits in **B**. With some abuse of notation, let $\mathbf{L} = (\mathbf{L}_1, ..., \mathbf{L}_k)$ and $\mathbf{L}' = (\mathbf{L}'_1, ..., \mathbf{L}'_k)$ where each $\mathbf{L}_i, \mathbf{L}'_i \in \mathcal{X}^{t(\epsilon)}$. Hence, they can be viewed as $kdt(\epsilon)$-dimensional Euclidean vectors. Here, **L** is the sample sequence with $\mathsf{TV}(f^*, \mathcal{J}(\mathbf{L}, \mathbf{B})) \leq 1/2$, and $\mathbf{L}'$ represents its perturbed version. Also, note that we have:

$$\|\mathbf{L} - \mathbf{L}'\|_2^2 = \sum_{i=1}^{k} \|\mathbf{L}_i - \mathbf{L}'_i\|_2^2. \tag{57}$$

According to the assumed properties of the class $\mathcal{F}$, for each component $i \in [k]$ of the latent mixture, we have:

$$\mathsf{TV}\left(\mathcal{J}(\mathbf{L}_i, \mathbf{B}_i), \mathcal{J}(\mathbf{L}'_i, \mathbf{B}_i)\right) \overset{a.s.}{\leq} \frac{r}{2} \|\mathbf{L}_i - \mathbf{L}'_i\|_2. \tag{58}$$

Due to the convexity of TV distance for mixtures and using Jensen's inequality (Boyd & Vandenberghe, 2004), we have:

$$\mathsf{TV}\left(\mathcal{J}_k(\mathbf{L}, \mathbf{B}), \mathcal{J}_k(\mathbf{L}', \mathbf{B})\right) \leq \sum_{i=1}^{k} \widehat{\alpha}_i \mathsf{TV}\left(\mathcal{J}(\mathbf{L}_i, \mathbf{B}_i), \mathcal{J}(\mathbf{L}'_i, \mathbf{B}_i)\right)$$

$$\leq \sum_{i=1}^{k} \frac{r}{2} \widehat{\alpha}_i \|\mathbf{L}_i - \mathbf{L}'_i\|_2. \tag{59}$$

Define $\boldsymbol{a} \triangleq (r/2)\,(\widehat{\alpha}_1, \ldots, \widehat{\alpha}_k)$ and $\mathbf{B} \triangleq \left(\|\mathbf{L}_1 - \mathbf{L}'_1\|_2, \ldots, \|\mathbf{L}_k - \mathbf{L}'_k\|_2\right)$. By Holder's inequality (according to Theorem 3.8 in (Rudin, 1987)), we have:

$$\frac{r}{2} \sum_{i=1}^{k} \widehat{\alpha}_i \|\mathbf{L}_i - \mathbf{L}'_i\|_2 \leq \|\boldsymbol{a}\|_2 \cdot \|\mathbf{B}\|_2$$

$$= \|\boldsymbol{a}\|_2 \cdot \|\mathbf{L} - \mathbf{L}'\|_2$$

$$\leq \frac{1}{2} r \sqrt{k} \|\mathbf{L} - \mathbf{L}'\|_2. \tag{60}$$

As a result, the following bound holds:

$$\mathsf{TV}\left(\mathcal{J}_k(\mathbf{L}, \mathbf{B}), \mathcal{J}_k(\mathbf{L}', \mathbf{B})\right) \leq \frac{1}{2} r \sqrt{k} \|\mathbf{L} - \mathbf{L}'\|_2, \tag{61}$$

and the proof is complete. $\square$

*Proof of Claim 2.15.* The proof consists of two parts: i) First, we show that $\mathcal{F}$ cannot be learned in a PAC manner in the sense of total variation (TV) distance. ii) Next, we show that $\mathcal{F}$ does not satisfy Assumption 2.1, even though it is sample compressible.

Define $\mu^* \in \mathbb{R}$ and $\sigma^* > 0$ such that $f^* = \mathcal{N}(\mu^*, (\sigma^*)^2)$. Thus, in order to learn the class $\mathcal{F}$ of distributions $\mathcal{F} = \{\mathcal{N}(\mu, \sigma^2) \mid \mu \in \mathbb{R}, \ \sigma > 0\}$. we need to estimate both $\mu^*$ and $\sigma^*$. When a series of independent Gaussian noise values distributed according to $G = \mathcal{N}(0, \sigma_0^2)$ (for some $\sigma_0 > 0$) are added to the i.i.d. samples from $f^*$, the resulting samples are equivalently drawn from $\mathcal{N}(\mu^*, (\sigma^*)^2 + \sigma_0^2)$.

Suppose there exists an algorithm (decoder for $\mathcal{F}$) $\mathscr{A} \in \mathsf{A}(n, B)$ that, using $n$ noisy samples drawn from $f^* * G$ and activating one of its $2^B$ internal states, can estimate an $\epsilon$-approximation (in TV error sense) of $f^*$ for any $f^* \in \mathcal{F}$ with high

probability. Here, we expect that $\epsilon$ asymptotically decreases as $n \to \infty$. Let us define $\widehat{f} = \mathcal{N}(\widehat{\mu}, \widehat{\sigma}^2)$ as the algorithm's estimation of $f^*$. Based on Theorem 1.3 of (Devroye et al., 2018), if we assume $\epsilon < \frac{1}{200}$, we have:

$$\frac{|\mu^* - \widehat{\mu}|}{5\sigma^*} \leq \mathsf{TV}(f^*, \widehat{f}) \leq \epsilon. \tag{62}$$

As a result, we must have $|\mu^* - \widehat{\mu}| \leq 5\epsilon\sigma^*$ as a necessary (but not sufficient) condition for the algorithm $\mathscr{A}$ to output an $\epsilon$-approximation. Therefore, the algorithm should also be capable of reliably solving the following two-point hypothesis testing problem:

- Null hypothesis $H_0$: $\widehat{\mu} \leftarrow \mu_0 \triangleq \mu^*$,

- Alternative hypothesis $H_1$: $\widehat{\mu} \leftarrow \mu_1 \triangleq \mu^* + 10\epsilon\sigma^*$.

Note that since $\mu$ in $\mathcal{F}$ is a continuous degree of freedom and can take any value in $\mathbb{R}$, both values $\mu^*$ and $\mu^* + 10\epsilon\sigma^*$ can be chosen for $f^*$. The KL divergence between $\mathcal{N}(\mu_0, (\sigma^*)^2 + \sigma_0^2)$ and $\mathcal{N}(\mu_1, (\sigma^*)^2 + \sigma_0^2)$ is:

$$\mathsf{KL}(H_0 \| H_1) = \frac{(10\epsilon\sigma^*)^2}{2(\sigma^*)^2 + 2\sigma_0^2} = \frac{50\epsilon^2(\sigma^*)^2}{(\sigma^*)^2 + \sigma_0^2}, \tag{63}$$

where with some abuse of notation we replaced the *distribution* according to hypothesis $H_i$ with $H_i$ itself (for $i = 0, 1$). For $n$ samples, due to the independence of the noisy samples, the total ($n$-sample) KL divergence is:

$$\mathsf{KL}(H_0^n \| H_1^n) = n \frac{50\epsilon^2(\sigma^*)^2}{(\sigma^*)^2 + \sigma_0^2}, \tag{64}$$

where $H_i^n$ denotes the product probability measure of $n$ independent samples from the distribution associated with hypothesis $H_i$. Using Pinsker's inequality (Lemma 2.5 in (Tsybakov, 2009)), the TV distance satisfies:

$$\mathsf{TV}(H_0^n, H_1^n) \leq \sqrt{\frac{1}{2}\mathsf{KL}(H_0^n \| H_1^n)} = \sqrt{n\frac{25\epsilon^2(\sigma^*)^2}{(\sigma^*)^2 + \sigma_0^2}} \leq \frac{5\epsilon\sigma^*}{\sigma_0}\sqrt{n}. \tag{65}$$

At this point, we can apply Le Cam's lemma (Lemma 2.3 in (Tsybakov, 2009)), and lower-bound the minimum probability of $n$-sample misclassification between $H_0$ vs. $H_1$ as:

$$\inf_{\mathscr{A} \in \mathsf{A}(n,B)} \left( \mathbb{P}_{H_0}(|\widehat{\mu}_{\mathscr{A}} - \mu_0| \geq 5\epsilon\sigma^*) + \mathbb{P}_{H_1}(|\widehat{\mu}_{\mathscr{A}} - \mu_1| \geq 5\epsilon\sigma^*) \right) \geq \frac{1}{2}(1 - \mathsf{TV}(H_0^n, H_1^n)), \tag{66}$$

where $\widehat{\mu}_{\mathscr{A}}$ denotes the value of the mean $\widehat{\mu}$ returned by algorithm $\mathscr{A}$. Due to prior discussions around the inequalities in (62), we have the following:

$$\begin{aligned}
&\inf_{\mathscr{A} \in \mathsf{A}(n,B)} \left( \sup_{f^* \in \mathcal{F}} \mathbb{P}_{f^*} \left( \mathsf{TV}(\widehat{f}, f^*) \geq \epsilon \right) \right) \\
&\geq \inf_{\mathscr{A} \in \mathsf{A}(n,B)} \left( \mathbb{P}_{H_0} \left( \mathsf{TV}(\widehat{f}, f^* \leftarrow H_0) \geq \epsilon \right) + \mathbb{P}_{H_1} \left( \mathsf{TV}(\widehat{f}, f^* \leftarrow H_1) \geq \epsilon \right) \right) \\
&\geq \frac{1}{2}(1 - \mathsf{TV}(H_0^n, H_1^n)) \\
&\geq \frac{1}{2} \left( 1 - \frac{5\epsilon\sigma^*}{\sigma_0}\sqrt{n} \right). 
\end{aligned} \tag{67}$$

Since we can asymptotically decrease $\sigma^*$ toward zero in $\mathcal{F}$, the right-hand side of the above inequality can become arbitrarily close to $1/2$. Hence, we have shown that for any $\epsilon \leq 1/200$, achieving such an error with probability at least $1/2$ is impossible for any estimator $\mathscr{A}$, regardless of how large $n$ and $B$ are.

The remaining task (i.e., part (ii) of the proof) is to show that this distribution family cannot satisfy Assumption 2.1 for any finite $r \geq 0$. Recalling Definition A.3 of sample compression, consider $\mathbf{L}$ as the sequence of $\tau(\epsilon)$ samples chosen from the

target distribution $f^* = \mathcal{N}(\mu^*, (\sigma^*)^2)$. We denote by $\mathbf{L}'$ the perturbed samples. Let $\mathcal{J}$ be any decoder for $\mathcal{F}$ that achieves a given sample compression scheme. Define

$$\widehat{\mu} \triangleq \mu_{\mathcal{J}(\mathbf{L}, \mathbf{B})},$$
$$\widehat{\mu}' \triangleq \mu_{\mathcal{J}(\mathbf{L}', \mathbf{B})}, \tag{68}$$

as the mean values returned by decoder $\mathcal{J}$ based on observing the sample sequence $\mathbf{L}$ and its perturbed version $\mathbf{L}'$, respectively.

We show that for any sequence of bits B, even if the perturbed sample set $\mathbf{L}'$ is chosen arbitrarily close to $\mathbf{L}$, the total variation (TV) error term

$$\mathsf{TV}\left(\mathcal{J}\left(\mathbf{L}, \mathbf{B}\right), \mathcal{J}\left(\mathbf{L}', \mathbf{B}\right)\right)$$

either becomes larger than any value $\leq 1/200$ for at least one $f^* \in \mathcal{F}$, or $\mu_{\mathcal{J}(\mathbf{L}, \mathbf{B})}$ must be constant with respect to $\mathbf{L}$, contradicting the assumption that $\mathcal{J}$ is a decoder for $\mathcal{F}$.

To establish this, using (62), we obtain

$$\mathsf{TV}\left(\mathcal{J}\left(\mathbf{L}, \mathbf{B}\right), \mathcal{J}\left(\mathbf{L}', \mathbf{B}\right)\right) \geq \frac{|\widehat{\mu} - \widehat{\mu}'|}{5\sigma^*}, \tag{69}$$

under the assumption that $\mathsf{TV}\left(\mathcal{J}\left(\mathbf{L}, \mathbf{B}\right), \mathcal{J}\left(\mathbf{L}', \mathbf{B}\right)\right) < \frac{1}{200}$.

Now, if $|\widehat{\mu} - \widehat{\mu}'| > 0$, one can choose $\sigma^*$ sufficiently small such that the TV distance does not fall below $1/200$. On the other hand, if $|\widehat{\mu} - \widehat{\mu}'| = 0$, since no specific assumptions were made regarding $\mathbf{L}'$, it follows that $\mu_{\mathcal{J}(\mathbf{L}, \mathbf{B})}$ is independent of the samples in $\mathbf{L}$ and thus is a constant. Consequently, $\mathcal{J}$ cannot be a decoder, completing the proof.

$\square$

### C.2. Proofs of Claims: Part II

*Proof of Claim 2.17.* First, we prove $\mathcal{F}$ does not satisfy Assumption 2.7. The proof is based on contradiction. Suppose that there exists $\alpha \geq 0$ and $\xi < 1$ such that the assumption holds. Let $p(x) = (1 + \sin(kx))/(2\pi)$ and $q(x) = (1 - \sin(kx))/(2\pi)$ for some $k \in \mathbb{Z}_{\geq 0}$. Also, assume $P, Q : \mathbb{R} \to \mathbb{C}$ represent their respective Fourier transforms. Also, note that we have

$$\begin{aligned} P(\omega) - Q(\omega) &= \int_0^{2\pi} \frac{1}{2\pi} \left(1 + \sin(kx) - 1 + \sin(kx)\right) e^{-i\omega x} \mathrm{d}x \\ &= \frac{1}{\pi} \int_0^{2\pi} \sin(kx) e^{-i\omega x} \mathrm{d}x \\ &= \frac{1}{2\pi} \int_0^{2\pi} \left(e^{ikx} - e^{-ikx}\right) e^{-i\omega x} \mathrm{d}x \\ &= -\frac{2k\sin(\pi\omega)}{\pi\left(k^2 - \omega^2\right)} e^{-i\pi\omega}. \end{aligned} \tag{70}$$

As observed, $P(\omega) - Q(\omega)$ becomes singular at $\omega = \pm k$, indicating that most of its energy (i.e., its $\ell_2$-norm) is concentrated around $\pm k$. Consequently, increasing $k$ shifts the majority of the energy in the frequency domain away from the origin or any bounded $\alpha$-neighborhood of the origin. The following argument provides a formal mathematical justification:

$$\begin{aligned} \int_{|\omega| \geq \alpha} |P(\omega) - Q(\omega)|^2 \, \mathrm{d}\omega &= \|p - q\|_2^2 - \int_{-\alpha}^{\alpha} |P(\omega) - Q(\omega)|^2 \, \mathrm{d}\omega \\ &= \|p - q\|_2^2 - \frac{4}{\pi^2} \int_{-\alpha}^{\alpha} \frac{k^2 \sin^2(\pi\omega)}{(k^2 - \omega^2)^2} \mathrm{d}\omega \\ &\geq \|p - q\|_2^2 - \frac{8k^2\alpha}{\pi^2(k^2 - \alpha^2)^2}, \end{aligned} \tag{71}$$

where for the last inequality we assumed $k > \alpha$. Then, it can be deduced that for every $\xi < 1$, there exists some $k_0 \in \mathbb{N}$ such that

$$\|p - q\|_2^2 - \frac{8k_0^2\alpha}{\pi^2(k_0^2 - \alpha^2)^2} > \xi \|p - q\|_2^2,$$

which contradicts the initial assumption that Assumption 2.7 holds for some $\alpha$ and $\xi < 1$.

In the next step, we employ Le Cam's two-point method to establish a sufficient condition ensuring that a distribution family $\mathcal{F}$ *fails* to be PAC-learnable in total variation (TV) error. Let $\mathcal{F}$ be a family of distributions on a measurable space $\mathcal{X}$, and let $G$ be a fixed noise distribution (e.g., Gaussian). Suppose there exists a sequence $\{f_N\}_{N \geq 1} \subset \mathcal{F}$ and a reference distribution $f^* \in \mathcal{F}$, together with constants $c > 0$ and $\varepsilon_N \to 0$ as $N \to \infty$, such that for $\forall N \in \mathbb{N}$:

$$\mathsf{TV}\left(f_N * G, f^* * G\right) \leq \varepsilon_N \quad \text{and} \quad \mathsf{TV}\left(f_N, f^*\right) \geq c.$$

Then $\mathcal{F}$ is not PAC-learnable from samples drawn from $f * G$ in total variation distance. More precisely, due to Le Cam's method, for any (possibly randomized) estimator $\widehat{f}$ based on $n$ i.i.d. samples from either $f^* * G$ or $f_N * G$, we have

$$\inf_{\widehat{f}} \sup_{f \in \{f^*, f_N\}} \mathbb{P}\left(\mathsf{TV}(\widehat{f}, f) \geq \tfrac{c}{2}\right) \geq \frac{1}{2}\left(1 - \mathsf{TV}\left((f^* * G)^{\otimes n}, (f_N * G)^{\otimes n}\right)\right). \tag{72}$$

Using the subadditivity of TV distance over product distributions,

$$\mathsf{TV}\left(P^{\otimes n}, Q^{\otimes n}\right) \leq n\,\mathsf{TV}(P, Q),$$

it follows that if $n\,\varepsilon_N \leq 1/2$, the minimax error is at least $1/4$. In particular, no learning algorithm with $n = o(1/\varepsilon_N)$ samples can reliably distinguish between $f^*$ and $f_N$, nor approximate the true distribution within TV distance $c/2$ with probability larger than $3/4$. Since $\varepsilon\_N$ goes to zero as $N$ is increased, for any fixed $n$, one can find $N$ such that $n\,\varepsilon_N \leq 1/2$ occurs.

What remains is to create a distribution sequence $f_1, f_2, \ldots \in \mathcal{F}$ such that $f_n * G$ converges in TV sense, but $f_n$ does not. This is straightforward by, for example, considering the following sequence:

$$f_n \triangleq \frac{1 + \sin(nx)}{2\pi}, \quad \forall x \in [0, 2\pi], \ n \in \mathbb{N}. \tag{73}$$

It can be seen that

$$\lim_{n \to \infty} f_n * G = \lim_{n \to \infty} \left(\frac{1 + \sin(nx)}{2\pi}\right) * G$$

$$= \mathsf{Uniform}\left([0, 2\pi]\right) * G + \frac{1}{2\pi} \lim_{n \to \infty} \sin(nx) * G, \tag{74}$$

where equalities are point-wise. Since $G = \mathcal{N}(0, \sigma_0^2)$, the residual function can be computed as:

$$\left\{\lim_{n \to \infty} \sin(nx) * G\right\}(x) = \frac{1}{\sigma_0\sqrt{2\pi}} \lim_{n \to \infty} \int_{-\infty}^{\infty} \sin(nu)e^{-(x-u)^2/(2\sigma_0^2)}\mathrm{d}u, \quad x \in \mathbb{R},$$

$$\overset{(i)}{=} \lim_{n \to \infty} \frac{-1}{n\sqrt{2\pi}}e^{-(x-u)^2/(2\sigma_0^2)}\Big|_{-\infty}^{\infty}$$

$$- \frac{1}{\sigma_0\sqrt{2\pi}} \lim_{n \to \infty} \int_{-\infty}^{\infty} \frac{\cos(nu)}{n}\left(\frac{u - x}{\sigma_0^2}\right)e^{-(x-u)^2/(2\sigma_0^2)}\mathrm{d}u, \tag{75}$$

which equals to zero for all $x \in \mathbb{R}$. The equality (i) comes from applying integration by part. Therefore, we have

$$\lim_{n \to \infty} \mathsf{TV}\left(f_n * G, \mathsf{Uniform}\left([0, 2\pi]\right) * G\right) = 0. \tag{76}$$

On the other hand, the distributional sequence $f_n$, $n \in \mathbb{N}$ is not a Cauchy series with respect to total variation distance, since

$$\liminf_{n \to \infty} \mathsf{TV}\left(f_{2n}, f_n\right) = \liminf_{n \to \infty} \frac{1}{2\pi} \int_0^{2\pi} |\sin(2nx) - \sin(nx)|\,\mathrm{d}x$$

$$= \frac{1}{\pi} \liminf_{n \to \infty} \int_0^{2\pi} \left|\sin\left(\frac{nx}{2}\right)\right| \cdot \left|\cos\left(\frac{3nx}{2}\right)\right|\,\mathrm{d}x$$

$$> 0. \tag{77}$$

Therefore, $\{f_n\}_{n\in\mathbb{N}}$ does not converge in the sense of total variation distance. From a different perspective, we have

$$\mathsf{TV}\left(f_n, \mathrm{Uniform}\left([0, 2\pi]\right)\right) = \frac{1}{2\pi} \int_0^{2\pi} |\sin(nx)| \, \mathrm{d}x$$
$$= 4, \tag{78}$$

for all $n \in \mathbb{N}$. Hence, $f_n$s do not converge (in TV distance) to any distribution, specially $\mathsf{Uniform}([0, 2\pi])$. This completes the proof. $\square$

*Proof of Claim 2.8.* Assume to arbitrary distributions in $\mathcal{F}$, namely $p = \mathcal{N}(\boldsymbol{\mu}_1, \sigma_1^2 \boldsymbol{I}_d)$ and $q = \mathcal{N}(\boldsymbol{\mu}_2, \sigma_2^2 \boldsymbol{I}_d)$, where their respective Fourier transforms $P, Q : \mathbb{R}^d \to \mathbb{C}$ can be written as follows:

$$P(\boldsymbol{w}) = e^{i\boldsymbol{\mu}_1^T \boldsymbol{w} - \frac{1}{2}\sigma_1^2 \|\boldsymbol{w}\|_2^2}, \quad Q(\boldsymbol{w}) = e^{i\boldsymbol{\mu}_2^T \boldsymbol{w} - \frac{1}{2}\sigma_2^2 \|\boldsymbol{w}\|_2^2}. \tag{79}$$

Without loss of generality assume that $\sigma_1 \leq \sigma_2$. Therefore, we have

$$|P(\boldsymbol{w}) - Q(\boldsymbol{w})|^2 = \left| e^{i\boldsymbol{\mu}_1^T \boldsymbol{w} - \frac{1}{2}\sigma_1^2 \|\boldsymbol{w}\|_2^2} - e^{i\boldsymbol{\mu}_2^T \boldsymbol{w} - \frac{1}{2}\sigma_2^2 \|\boldsymbol{w}\|_2^2} \right|^2$$
$$= e^{-\sigma_1^2 \|\boldsymbol{w}\|_2^2 / 2} \cdot \left| e^{-\frac{1}{4}\sigma_1^2 \|\boldsymbol{w}\|_2^2} - e^{i(\boldsymbol{\mu}_2 - \boldsymbol{\mu}_1)^T \boldsymbol{w}} e^{-\frac{1}{2}(\sigma_2^2 - \sigma_1^2/2)\|\boldsymbol{w}\|_2^2} \right|^2$$
$$\triangleq f(\|\boldsymbol{w}\|_2) \cdot g(\boldsymbol{w}). \tag{80}$$

Note that since we have $\sigma_1 \geq \sigma_0 > 0$, $f$ is strictly decreasing and both $f$ and $g$ become exponentially small as $\|\boldsymbol{w}\|_2 \to \infty$. In this regard, one can write:

$$\int_{\|\boldsymbol{w}\|_2 \geq \alpha} |P(\boldsymbol{w}) - Q(\boldsymbol{w})|^2 \, \mathrm{d}\boldsymbol{w} = \int_{\|\boldsymbol{w}\|_2 \geq \alpha} f(\|\boldsymbol{w}\|_2) g(\boldsymbol{w}) \mathrm{d}\boldsymbol{w}$$
$$\leq \int_{\mathbb{R}^d} \min\left\{ f(\|\boldsymbol{w}\|_2), f(\alpha) \right\} g(\boldsymbol{w}) \mathrm{d}\boldsymbol{w}$$
$$\leq f(\alpha) \int_{\mathbb{R}^d} g(\boldsymbol{w}) \mathrm{d}\boldsymbol{w}. \tag{81}$$

Now, it should be noted that due to Parseval's theorem, we have

$$\int_{\mathbb{R}^d} g(\boldsymbol{w}) \mathrm{d}\boldsymbol{w} = \int_{\mathbb{R}^d} \left| e^{-\frac{1}{4}\sigma_1^2 \|\boldsymbol{w}\|_2^2} - e^{i(\boldsymbol{\mu}_2 - \boldsymbol{\mu}_1)^T \boldsymbol{w}} e^{-\frac{1}{2}(\sigma_2^2 - \sigma_1^2/2)\|\boldsymbol{w}\|_2^2} \right|^2 \mathrm{d}\boldsymbol{w}$$
$$= \left\| \mathcal{N}\left( \boldsymbol{\mu}_1, \frac{\sigma_1^2}{2} \boldsymbol{I}_d \right) - \mathcal{N}\left( \boldsymbol{\mu}_2, \left( \sigma_2^2 - \frac{\sigma_1^2}{2} \right) \boldsymbol{I}_d \right) \right\|_2^2. \tag{82}$$

Therefore, so far we have shown that

$$\int_{\|\boldsymbol{w}\|_2 \geq \alpha} |P(\boldsymbol{w}) - Q(\boldsymbol{w})|^2 \, \mathrm{d}\boldsymbol{w} \leq e^{-\sigma_1^2 \alpha^2 / 2} \left\| \mathcal{N}\left( \boldsymbol{\mu}_1, \frac{\sigma_1^2}{2} \boldsymbol{I}_d \right) - \mathcal{N}\left( \boldsymbol{\mu}_2, \left( \sigma_2^2 - \frac{\sigma_1^2}{2} \right) \boldsymbol{I}_d \right) \right\|_2^2. \tag{83}$$

Next, we use the following formula to the $\ell_2$-norm between two isotropic Gaussian densities in $\mathbb{R}^d$:

$$\|p - q\|_2^2 = \left\| \mathcal{N}\left( \boldsymbol{\mu}_1, \sigma_1^2 \boldsymbol{I}_d \right) - \mathcal{N}\left( \boldsymbol{\mu}_2, \sigma_2^2 \boldsymbol{I}_d \right) \right\|_2^2$$
$$= \frac{1}{(4\pi)^{d/2}} \left( \frac{1}{\sigma_1^d} + \frac{1}{\sigma_2^d} - \frac{2^{1+d/2}}{(\sigma_1^2 + \sigma_2^2)^{d/2}} \exp\left( -\frac{\|\boldsymbol{\mu}_1 - \boldsymbol{\mu}_2\|^2}{2(\sigma_1^2 + \sigma_2^2)} \right) \right). \tag{84}$$

In this regard, for fixed parameters $d \in \mathbb{N}$, $\theta, \gamma \geq 0$, let us define the function

$$h(\sigma) = h(\sigma | \theta, \gamma, d) \triangleq (4\pi)^{d/2} \left\| \mathcal{N}\left( \boldsymbol{0}, \sigma^2 \boldsymbol{I}_d \right) - \mathcal{N}\left( \Delta\boldsymbol{\mu}, (\sigma^2 + \theta) \boldsymbol{I}_d \right) \right\|_2^2 \tag{85}$$
$$= \frac{1}{\sigma^d} + \frac{1}{(\sigma^2 + \theta)^{d/2}} - \frac{2^{1+d/2}}{(2\sigma^2 + \theta)^{d/2}} \exp\left( -\frac{\gamma}{4\sigma^2 + 2\theta} \right),$$

where $\gamma \triangleq \|\Delta\boldsymbol{\mu}\|_2^2 \geq 0$. Then, it can be seen that we have

$$\int_{\|\boldsymbol{w}\|_2 \geq \alpha} |P(\boldsymbol{w}) - Q(\boldsymbol{w})|^2 \, \mathrm{d}\boldsymbol{w} \leq e^{-\sigma_1^2 \alpha^2 / 2} \|p - q\|_2^2 \cdot \frac{h(\sigma_1/\sqrt{2})}{h(\sigma_1)}, \tag{86}$$

with $\gamma \leftarrow \|\boldsymbol{\mu}_2 - \boldsymbol{\mu}_1\|_2^2$ and $\theta \leftarrow \sigma_2^2 - \sigma_1^2$. On the other hand, algebraic analysis of (85) reveals that we always have

$$\sup_{\sigma_1 \geq \sigma_0} \sup_{\theta, \gamma \geq 0} \frac{h\left(\frac{\sigma_1}{\sqrt{2}} \middle| \gamma, \theta, d\right)}{h\left(\sigma_1 \middle| \gamma, \theta, d\right)} \leq 4 \cdot 2^{d/2}. \tag{87}$$

As a result, and considering $\sigma_1 \geq \sigma_0$, the following bound holds and the proof is complete:

$$\int_{\|\boldsymbol{w}\|_2 \geq \alpha} |P(\boldsymbol{w}) - Q(\boldsymbol{w})|^2 \, \mathrm{d}\boldsymbol{w} \leq 2^{(d/2+2)} e^{-\sigma_0^2 \alpha^2 / 2} \|p - q\|_2^2. \tag{88}$$

$\square$

*Proof of Claim 2.9.* Let $p, q \in k-\mathrm{Mix}(\mathcal{F})$. Then, there exist coefficient vectors $\boldsymbol{\alpha} = (\alpha_1, \ldots, \alpha_k)$ and $\boldsymbol{\beta} = (\beta_1, \ldots, \beta_k)$ with $\boldsymbol{\alpha}, \boldsymbol{\beta} \in \Delta^{k-1}$, such that

$$p(x) - q(x) = \sum_{i=1}^{k} \alpha_i \frac{\mathbb{1}(a_i \leq x \leq b_i)}{b_i - a_i} - \sum_{i=1}^{k} \beta_i \frac{\mathbb{1}(a_i' \leq x \leq b_i')}{b_i' - a_i'}, \tag{89}$$

where we have $a_i, b_i, a_i', b_i' \in \mathbb{R}$ for all $i \in [k]$, and $b_i - a_i, b_i' - a_i' \geq T$. We first note that

$$\|p - q\|_\infty = \sup_{x \in \mathbb{R}} |p(x) - q(x)| \leq \frac{1}{T}.$$

Let $c_1 \leq c_2 \leq \ldots \leq c_{4k}$ be (at most) $4k$ unique points representing the sorted values of $a_i, b_i, a_i'$ and $b_i'$ for all $i$. In this regard, the function $p(x) - q(x)$ is piecewise constant on intervals $(c_{i-1}, c_i)$ for $i \in [4k]$, and is particularly zero in both $(-\infty, c_1)$ and $(c_{4k}, +\infty)$. Using this fact, $p - q$ can be rewritten as the sum of (at most) $4k - 1$ separate pulse functions, as follows:

$$p(x) - q(x) = \sum_{i=1}^{4k-1} h_i \mathbb{1}\left(x \in I_i\right), \tag{90}$$

where $I_i = (c_i, c_{i+1})$, and $-1/T \leq h_i \leq 1/T$. Note that this does come at the loss of generality, since we can always assume some of $h_i$s are zero. Also, we have neglected the values of $p - q$ at the discontinuity points. Also, let $t_i \triangleq \mathrm{len}(I_i)$ denote the length of the interval $I_i$.

Assume $P(w), Q(w)$ represent the Fourier transforms of $p, q$, respectively. Then, for any $\alpha > 0$ we have

$$
\frac{1}{2\pi} \int_{|w| \geq \alpha} |P(w) - Q(w)|^2 \, \mathrm{d}w = \frac{1}{\pi} \int_\alpha^\infty |P(w) - Q(w)|^2 \, \mathrm{d}w
$$

$$
= \frac{1}{\pi} \int_\alpha^\infty \left| \sum_{i=1}^{4k-1} h_i \mathsf{F}\left\{ \mathbb{1}\left(x \in I_i\right) \right\}(w) \right|^2 \mathrm{d}w
$$

$$
\stackrel{(*)}{=} \frac{1}{\pi} \int_\alpha^\infty \sum_{i=1}^{4k-1} h_i^2 \left| \mathsf{F}\left\{ \mathbb{1}\left(x \in I_i\right) \right\}(w) \right|^2 \mathrm{d}w
$$

$$
= \frac{4}{\pi} \sum_{i=1}^{4k-1} h_i^2 \int_\alpha^\infty \frac{\sin^2\left(wt_i/2\right)}{w^2} \mathrm{d}w
$$

$$
= \frac{2}{\pi} \sum_{i=1}^{4k-1} h_i^2 t_i \left( \frac{\pi}{2} - \int_0^{\alpha t_i/2} \frac{\sin^2(u)}{u^2} \mathrm{d}u \right)
$$

$$
= \sum_{i=1}^{4k-1} h_i^2 t_i \left( 1 - \zeta\left( \frac{\alpha t_i}{2} \right) \right). \tag{91}
$$

The equality (*) holds since functions $h_i \mathbb{1}(x \in I_i)$ and $h_j \mathbb{1}(x \in I_j)$ for $i \neq j$ are *orthogonal* due to non-overlapping supports $I_i$ and $I_j$. Fourier transform, similar to any other orthonormal transformation, preserves orthogonality. Define $\varepsilon^2 \triangleq \|p - q\|_2^2$ and note that we have

$$
\varepsilon^2 = \sum_{i=1}^{4k-1} h_i^2 t_i.
$$

Then for fixed $\alpha$ and $\varepsilon$, and over varying $p, q \in k\text{-Mix}(\mathcal{F})$, we obtain:

$$
\frac{1}{2\pi} \int_{|w| \geq \alpha} |P(w) - Q(w)|^2 \, \mathrm{d}w \leq \sup_{(h_i, t_i),\, \forall i \in [4k-1]} \sum_{i=1}^{4k-1} h_i^2 t_i \left( 1 - \zeta\left( \frac{\alpha t_i}{2} \right) \right)
$$

$$
= \varepsilon^2 - \inf_{(h_i, t_i),\, \forall i \in [4k-1]} \sum_{i=1}^{4k-1} h_i^2 t_i \zeta\left( \frac{\alpha t_i}{2} \right)
$$

$$
\text{subject to} \quad \sum_{i=1}^{4k-1} h_i^2 t_i = \varepsilon^2, \quad t_i \geq 0, \; |h_i| \leq \frac{1}{T}. \tag{92}
$$

Since $\zeta$ is non-decreasing, the minimum is achieved when all $t_i$ are equal. Thus,

$$
t_i^* = t \triangleq \frac{\varepsilon^2}{\sum_{i=1}^{4k-1} h_i^2}, \quad \forall j \in [4k-1].
$$

Substituting into the bound, we have

$$
\frac{1}{2\pi} \int_{|w| \geq \alpha} |P(w) - Q(w)|^2 \, \mathrm{d}w \leq \varepsilon^2 \left( 1 - \inf_{h_1, \ldots, h_{4k-1}} \zeta\left( \frac{\alpha \varepsilon^2}{2 \sum_i h_i^2} \right) \right)
$$

$$
= \varepsilon^2 \left( 1 - \zeta\left( \frac{\alpha T^2 \varepsilon^2}{2(4k-1)} \right) \right), \tag{93}
$$

which completes the proof. $\square$

*Proof of Claim 3.1.* The proof closely follows the argument in Claim 2.9 (see Appendix C.2). Let $\boldsymbol{\alpha}, \boldsymbol{\beta} \in \Delta^{k-1}$ be $d$-dimensional discrete probability vectors, and consider an arbitrary difference measure $p - q$ with $p, q \in k\text{-Mix}(\mathcal{F})$:

$$
p(\boldsymbol{x}) - q(\boldsymbol{x}) = \sum_{i=1}^k \alpha_i \prod_{j=1}^d \frac{\mathbb{1}\left(x_j \in I_{i,j}\right)}{|I_{i,j}|} - \sum_{i=1}^k \beta_i \prod_{j=1}^d \frac{\mathbb{1}\left(x_j \in I'_{i,j}\right)}{|I'_{i,j}|}, \quad \forall \boldsymbol{x} \in \mathbb{R}^d, \tag{94}
$$

where $I_{i,j}$ and $I'_{i,j}$ for $i \in [k]$ and $j \in [d]$ are arbitrary intervals with a minimum length (i.e., Lebesgue measure) of $T$. This function is piecewise constant and nonzero over the union of at most $2k$ axis-aligned and potentially overlapping rectangles in $\mathbb{R}^d$. According to several known results in high-dimensional or computational geometry, particularly those concerning orthogonal range decomposition or boolean combinations of boxes (see, for example, (Berg et al., 2008; Chazelle, 1988; Overmars & Yap, 1991)), this sum can be decomposed into a linear combination of at most $\Theta\left(k^d\right)$ disjoint axis-aligned rectangles:

$$p(\boldsymbol{x}) - q(\boldsymbol{x}) = \sum_{i=1}^{\Theta(k^d)} h_i \prod_{j=1}^{d} \frac{\mathbb{1}\left(x_i \in \mathsf{T}_{i,j}\right)}{|t_{i,j}|}, \tag{95}$$

where $\mathsf{T}_{i,j}$ denotes a 1D interval of length $t_{i,j}$, and the coefficients satisfy $|h_i| \leq 1/T$. Proceeding analogously to Claim 2.9, we have

$$\frac{1}{(2\pi)^d} \int_{\|\boldsymbol{w}\|_2 \geq \alpha} |P(\boldsymbol{w}) - Q(\boldsymbol{w})|^2 \, \mathrm{d}\boldsymbol{w} \leq \frac{1}{(2\pi)^d} \int_{\|\boldsymbol{w}\|_\infty \geq \alpha/\sqrt{d}} |P(\boldsymbol{w}) - Q(\boldsymbol{w})|^2 \, \mathrm{d}\boldsymbol{w}$$

$$= \varepsilon^2 - \sum_{i=1}^{\Theta(k^d)} h_i^2 \prod_{j=1}^{d} \frac{2}{\pi} t_{i,j} \int_0^{\alpha t_{i,j}/2\sqrt{d}} \frac{\sin^2 u}{u^2} \, \mathrm{d}u$$

$$\leq \varepsilon^2 - \inf_{h_i, t_{i,j}} \sum_{i=1}^{\Theta(k^d)} h_i^2 \prod_{j=1}^{d} t_{i,j} \zeta\left(\frac{\alpha t_{i,j}}{2\sqrt{d}}\right)$$

$$\text{subject to} \quad \sum_{i=1}^{\Theta(k^d)} h_i^2 \prod_{j=1}^{d} t_{i,j} = \varepsilon^2. \tag{96}$$

Applying the method of Lagrange multipliers to this constrained optimization problem, the minimum is achieved when all $t_{i,j}$ are equal (refer to the proof of Claim 2.9 in Appendix C.2), yielding

$$t_{i,j}^* = \left(\frac{\varepsilon^2}{\sum_{i=1}^{\Theta(k^d)} h_i^2}\right)^{1/d} \geq \frac{1}{c}\left(\frac{T^2\varepsilon^2}{k^d}\right)^{1/d}, \quad \forall i, j, \tag{97}$$

where $c$ is a universal constant. Substituting this into the earlier bound completes the proof. $\qquad\square$

## D. Agnostic Learnability from Corrupted Samples

In this subsection, we show that one can extend all the previous propositions in the noisy setting to the *agnostic* case (see Definition A.1). However, as expected, we need the stronger (i.e., *$\rho$-robust*) version of the sample compression notion in Definition D.1.

**Definition D.1** (Robust Sample Compression: Agnostic Setting)**.** Extending the above definition, let $\rho \geq 0$. We say $\mathcal{F}$ admits $(\tau, t, m, \rho)$-robust sample compression if there exists a decoder $\mathcal{J}$ for $\mathcal{F}$ such that for any distribution $f \in \mathcal{F}$ and any distribution $g \in L^2(\mathcal{X})$ (not necessarily in $\mathcal{F}$) with $\|f - g\|_{\mathrm{TV}} \leq \rho$, the following holds: For any $\varepsilon \in (0,1)$, if a sample set $S$ is drawn from $g^{m(\varepsilon)\log(\frac{1}{\delta})}$, then with probability at least $1 - \delta$, there exists a sequence $\mathbf{L}$ of at most $\tau(\varepsilon)$ elements of $S$, and a sequence $\mathbf{B}$ of at most $t(\varepsilon)$ bits, such that $\|\mathcal{J}(\mathbf{L}, \mathbf{B}) - f\|_{\mathrm{TV}} \leq \varepsilon$.

For the $\rho$-robust setting, the sample sequence $\mathbf{L}$ and the bit sequence $\mathbf{B}$ implicitly depend on the decoder $f$. Ashtiani et al. showed that the family of general Gaussian distributions admits a robust compression scheme with robustness parameter $\rho$ as large as $2/3$. More broadly, the robust formulation of sample compression can be interpreted as an agnostic extension of the classical PAC-learnability framework.

**Proposition D.2** (Robust Sample Compressibility of Noisy $\mathcal{F}$)**.** *For $d \in \mathbb{N}$, assume $\mathcal{F}$ be a class of $d$-dimensional distributions satisfying Assumption 1.3 for functions $\tau, t, m : (0,1) \to \mathbb{N}$ and constant $\rho \geq 0$, and let Assumption 2.1 hold for a bounded constant $r \geq 0$. Also, let $G$ be the density of an isotropic noise over $\mathbb{R}^d$ with a component-wise CDF of*

$\Phi_G : \mathbb{R} \to [0, 1]$. *Then,* $\mathcal{F} * G \triangleq \{f * G \mid f \in \mathcal{F}\}$ *admits*

$$\left( \tau\left(\tfrac{\epsilon}{2}\right), t\left(\tfrac{\epsilon}{2}\right) + d\tau\left(\tfrac{\epsilon}{2}\right) \log_2\left( 1 + \frac{r}{\epsilon}\sqrt{d\tau\left(\tfrac{\epsilon}{2}\right)} \left| \Phi_G^{-1}\left( \frac{\delta}{4dm\left(\tfrac{\epsilon}{2}\right)\log\tfrac{2}{\delta}} \right) \right| \right), m\left(\tfrac{\epsilon}{2}\right), \rho \right) \tag{98}$$

*-robust sample compression, for any* $\epsilon \in (0, 1)$.

Using Proposition D.2, having Assumption 2.7 holds for $\mathcal{F} \cup f^*$ (since $f^*$ is not necessarily in $\mathcal{F}$), and assuming there exists $f \in \mathcal{F}$ with $\|f - f^*\|_1 \le \rho$ for some $\rho > 0$, we have:

**Theorem D.3** (Agnostic Main Result). *Let* $\mathcal{F}$ *be a distribution family over* $\mathcal{X} \subseteq \mathbb{R}^d$ *satisfying Assumption 1.3 with a robust sample compression scheme* $(\tau, t, m, \rho)$ *for some* $\rho > 0$*, and Assumption 2.1 with a bounded constant* $r \ge 0$*. Fix any unknown* $f^* \in L^2(\mathcal{X})$ *(not necessarily in* $\mathcal{F}$*) with* $\inf_{f \in \mathcal{F}} \|f^* - f\|_{\mathrm{TV}} \le \rho$*, and let Assumption 2.7 hold over* $\mathcal{F} \cup \{f^*\}$ *for the set of pairs* $\mathsf{P}(\mathcal{F} \cup f^*) = \{(\alpha, \xi)\}$*. Assume* $G \in L^2(\mathcal{X})$ *be a symmetric product measure with component-wise CDF of* $\Phi_G$*. Define*

$$B_G(\alpha) \triangleq \mathsf{Vol}^{-1}\left(\mathbb{B}_2^d(1)\right) \alpha^{-d} \inf_{\|\boldsymbol{\omega}\|_2 \le \alpha} |\mathsf{F}\{G\}(\boldsymbol{\omega})|$$

*for* $\alpha > 0$*, where* $\mathbb{B}_2^d(1)$ *denoting the* $\ell_2$ *ball of radius* $1$ *in* $\mathbb{R}^d$*. For all* $\epsilon, \delta \in (0, 1)$*, assume we have* $n$ *i.i.d. samples from* $f^* * G$ *with*

$$n \ge N_{\tau,t,m}^{\mathsf{Clean}}(6\epsilon, \delta/2) + \tag{99}$$
$$\mathcal{O}\left( \frac{d\tau(\epsilon)}{\epsilon^2} \log\left( \frac{r}{\epsilon}\sqrt{d\tau(\epsilon)} \left| \Phi_G^{-1}\left( \frac{\delta}{8dm(\epsilon)\log\tfrac{4}{\delta}} \right) \right| \right) \log\left( m(\epsilon)\log\left(\tfrac{1}{\delta}\right) \right) \right),$$

*where* $N_{\tau,t,m}^{\mathsf{Clean}}(\epsilon, \delta)$ *is the sample complexity of the noiseless regime, as defined in Theorem A.5 (full details in Theorem C.3). Then, there exists a deterministic algorithm that takes the* $n$ *perturbed samples as input, and outputs* $\widehat{f} \in \mathcal{F}$ *such that the following bound holds with probability at least* $1 - \delta$*:*

$$\|\widehat{f} - f^*\|_2 \le \left( \inf_{(\alpha, \xi) \in \mathsf{P}(\mathcal{F})} \frac{24}{\sqrt{B_G(\alpha)(1 - \xi)}} \right) \cdot \left( \epsilon + \max\left\{\frac{1}{4}, \frac{1}{6\rho}\right\} \inf_{f \in \mathcal{F}} \mathsf{TV}\left(f * G, f^* * G\right) \right). \tag{100}$$

The proof, which shortly follows, is based on Theorem 4.5 of (Ashtiani et al., 2020). Furthermore, note that assuming there exists $f' \in \mathcal{F}$ such that $\mathsf{TV}(f', f^*) = \inf_{f \in \mathcal{F}} \mathsf{TV}(f, f^*)$, then we have:

$$\inf_{f \in \mathcal{F}} \mathsf{TV}(f, f^*) = \mathsf{TV}(f', f^*) \ge \mathsf{TV}(f' * G, f^* * G) \ge \inf_{f \in \mathcal{F}} \mathsf{TV}(f * G, f^* * G) \tag{101}$$

Thus, we can rewrite (100) as the following simpler alternative formulation at the cost of losing some tightness:

$$\|\widehat{f} - f^*\|_2 \le \left( \inf_{(\alpha, \xi) \in \mathsf{P}(\mathcal{F})} \frac{24}{\sqrt{B_G(\alpha)(1 - \xi)}} \right) \cdot \left( \epsilon + \max\left\{\frac{1}{4}, \frac{1}{6\rho}\right\} \inf_{f \in \mathcal{F}} \mathsf{TV}\left(f, f^*\right) \right). \tag{102}$$

*Proof of Theorem D.3.* We use the result of Proposition D.2, which assuming the $(\tau, t, m)$-$k$ robust sample compressibility of $\mathcal{F}$ guarantees the $k$-robust sample compressibility of $\mathcal{F} * G$. Specifically, for any $(\epsilon, \delta) \in (0, 1)$, $\mathcal{F} * G$ admits

$$\left[ \tau\left(\tfrac{\epsilon}{2}\right), t\left(\tfrac{\epsilon}{2}\right) + d\tau\left(\tfrac{\epsilon}{2}\right) \log_2\left( 1 + \frac{r\sqrt{d\tau(\epsilon/2)}}{\epsilon} \left| \Phi_G^{-1}\left( \frac{\delta}{4dm\left(\tfrac{\epsilon}{2}\right)\log\tfrac{2}{\delta}} \right) \right| \right), m\left(\tfrac{\epsilon}{2}\right) \log\frac{2}{\delta} \right]$$

*-k robust sample compression. Therefore, there exists a decoder* $\mathcal{J}$ *such that given* $m\left(\tfrac{\epsilon}{2}\right)\log\tfrac{2}{\delta}$ *i.i.d. samples from any* $f^* * G$ *such that there exist* $g^* \in \mathcal{F}$ *such that* $\|g^* - f^*\|_1 \le k$*, the decoder outputs* $\widehat{f} * G \in \mathcal{F} * G$ *that satisfies* $\mathsf{TV}(f^* * G, \widehat{f} * G) \le \epsilon + \max\{3, \tfrac{2}{k}\} \inf_{f \in \mathcal{F}} \mathsf{TV}(f^* * G, f * G)$ *with probability at least* $1 - \delta$*.

The next step is to use a seminal proposition from (Ashtiani et al., 2018), which combines Theorem A.4 (originally from (Devroye & Lugosi, 2001)) and a number of concentration inequalities in order to prove the information-theoretic learnability of *robust sample compressible* distribution classes in general. The following theorem establishes a fundamental theoretical connection between $(\tau, t, m)$-$k$ robust sample compressibility and PAC-learnability of a distribution class:

**Theorem D.4** (Theorem 3.5 in (Ashtiani et al., 2018))**.** *Suppose $\mathcal{F}$ admits $(\tau, t, m)$-$k$ robust sample compression for some functions $\tau, t, m : (0, 1) \to \mathbb{N}$. For any $\epsilon, \delta \in (0, 1)$, let $\tau'(\epsilon) \triangleq \tau(\epsilon) + t(\epsilon)$. Also, define $N_{\tau,t,m}^{\mathsf{Clean}}$ as*

$$N_{\tau,t,m}^{\mathsf{Clean}}(\epsilon, \delta) \triangleq \mathcal{O}\left( m\left(\frac{\epsilon}{6}\right) \log_3\left(\frac{2}{\delta}\right) + \frac{32}{\epsilon^2}\left[ \tau'\left(\frac{\epsilon}{6}\right) \log\left( m\left(\frac{\epsilon}{6}\right) \log_3\left(\frac{2}{\delta}\right) \right) + \log\left(\frac{6}{\delta}\right) \right] \right).$$

*Then, there exists a deterministic algorithm that by having $n \geq N_{\tau,t,m}^{\mathsf{Clean}}(\epsilon, \delta)$ i.i.d. samples from any unknown distribution $f^*$ not necessarily in $\mathcal{F}$ such that there exists $g^* \in \mathcal{F}$ that $\|g^* - f^*\|_1 \leq k$, outputs $\widehat{f} \in \mathcal{F}$ where we have $\mathsf{TV}(f^*, \widehat{f}) \leq \epsilon + \max\{3, \frac{2}{k}\} \inf_{f \in \mathcal{F}} \mathsf{TV}(f^*, f)$, with probability of at least $1 - \delta$.*

Combining the results from Proposition D.2 and Theorem D.4, one can deduce that there exists a deterministic algorithm, that for any $\epsilon, \delta \in (0, 1)$, upon having

$$n \geq N_{\tau,t,m}^{\mathsf{Clean}}(6\epsilon, \delta/2) \tag{103}$$
$$+ \mathcal{O}\left[ \frac{2d\tau(\epsilon)}{9\epsilon^2} \log_2\left( \frac{r\sqrt{d\tau(\epsilon)}}{2\epsilon} \left| \Phi_G^{-1}\left( \frac{\delta}{8dm(\epsilon)\log\frac{4}{\delta}} \right) \right| \right) \log\left( m(\epsilon) \log_3\left(\frac{4}{\delta}\right) \right) \right]$$

i.i.d. samples from any $f^* * G$ such that there exists $g^* \in \mathcal{F}$ that $\|g^* - f^*\|_1 \leq k$, outputs $\widehat{f} * G \in \mathcal{F} * G$ which is an $12\epsilon + \max\{3, \frac{2}{k}\} \inf_{f \in \mathcal{F}} \mathsf{TV}(f^* * G, f * G)$-approximation of $f^* * G$ with probability at least $1 - \delta$. Mathematically, we have:

$$\mathbb{P}\left( \mathsf{TV}(f^* * G, \widehat{f} * G) \leq 12\epsilon + \max\{3, \frac{2}{k}\} \inf_{f \in \mathcal{F}} \mathsf{TV}(f^* * G, f * G) \right) \geq 1 - \delta, \tag{104}$$

where $\mathbb{P}(\cdot)$ is with respect to the randomness of generating samples from $f^* * G$. The final step is to establish an upper bound for $\|\widehat{f} - f^*\|_2$ with probability at least $1 - \delta$, leveraging (104). As discussed in Section 2, such a relationship does not necessarily hold, since convolution with *low-frequency* noise densities (e.g., Gaussian noise) attenuates high-frequency components of the density difference $\widehat{f} - f^*$. Consequently, ensuring that $\mathsf{TV}(f^* * G, \widehat{f} * G)$ is small does not directly imply that $\widehat{f}$ is close to $f^*$ in a general sense.

However, under Assumption 2.7 for $\mathcal{F} \cup f^*$, we have assumed that density differences in $\mathcal{F} \cup f^*$ retain a non-negligible portion of their energy in low-frequency regions. This ensures that recovering $f^* * G$ also leads to recovering $f^*$ itself. The following lemma formally establishes this fact:

**Lemma D.5.** *For any $\epsilon, \delta \in (0, 1)$, assume that there exists an algorithm such that (104) holds for the output distribution $\widehat{f}$. Also, assume that Assumption 2.7 holds for a set of pairs $\mathsf{P}(\mathcal{F} \cup \{^*\}) = \{(\alpha, \xi)\}$. Then, with probability of at least $1 - \delta$, the following bound holds uniformly for all $(\alpha, \xi) \in \mathsf{P}(\mathcal{F})$:*

$$\|\widehat{f} - f^*\|_2 \leq \frac{24\epsilon + \max\{6, \frac{4}{k}\} \inf_{f \in \mathcal{F}} \mathsf{TV}(f^* * G, f * G)}{\sqrt{B_G(\alpha)(1 - \xi)}}. \tag{105}$$

*Proof.* Based on (104) and the equivalence between TV distance and $\|\cdot\|_1 / 2$, with probability at least $1 - \delta$, we have:

$$\int_{\mathbb{R}^d} \left| f^* * G(\boldsymbol{t}) - \widehat{f} * G(\boldsymbol{t}) \right| d\boldsymbol{t} \leq 24\epsilon + \max\{6, \frac{4}{k}\} \inf_{f \in \mathcal{F}} \mathsf{TV}(f^* * G, f * G). \tag{106}$$

We invoke the Hausdorff–Young inequality, which we state as a Lemma C.5. Let $P$ and $Q$ denote the Fourier transforms of $f^*$ and $\widehat{f}$, respectively. Using the fact that convolution operator turns into point-wise multiplication in the Fourier domain, and also applying Lemma C.5, we obtain the following inequalities for any $\alpha \geq 0$:

$$24\epsilon + \max\{6, \frac{4}{k}\} \inf_{f \in \mathcal{F}} \mathsf{TV}(f^* * G, f * G) \geq \int_{\mathbb{R}^d} \left| f^* * G(\boldsymbol{t}) - \widehat{f} * G(\boldsymbol{t}) \right| d\boldsymbol{t}$$
$$\overset{(i)}{\geq} \sup_{\boldsymbol{\omega} \in \mathbb{R}^d} |P(\boldsymbol{\omega})G(\boldsymbol{\omega}) - Q(\boldsymbol{\omega})G(\boldsymbol{\omega})| \tag{107}$$

where (i) is according to Lemma C.5 by setting $p = 1$ and $p' = \infty$. Therefore, we can obtain the following relations:

$$
\begin{aligned}
|P(\boldsymbol{\omega}) - Q(\boldsymbol{\omega})| &\leq \left( \inf_{\|\boldsymbol{\omega}\|_2 \leq \alpha} |G(\boldsymbol{\omega})| \right)^{-1} |P(\boldsymbol{\omega})G(\boldsymbol{\omega}) - Q(\boldsymbol{\omega})G(\boldsymbol{\omega})| \\
&\leq \left( \inf_{\|\boldsymbol{\omega}\|_2 \leq \alpha} |G(\boldsymbol{\omega})| \right)^{-1} \sup_{\boldsymbol{\omega} \in \mathbb{R}^d} |P(\boldsymbol{\omega})G(\boldsymbol{\omega}) - Q(\boldsymbol{\omega})G(\boldsymbol{\omega})| \\
&\overset{(i)}{\leq} \frac{24\epsilon + \max\{6, \frac{4}{k}\} \inf_{f \in \mathcal{F}} \mathsf{TV}(f^* * G, f * G)}{\inf_{\|\boldsymbol{\omega}\|_2 \leq \alpha} |G(\boldsymbol{\omega})|}
\end{aligned}
\tag{108}
$$

where (i) is due to (27). Then, by defining $\mathbb{B}_2^d(\alpha) = \{\boldsymbol{\omega} | \|\boldsymbol{\omega}\| \leq \alpha\}$, the following bound has been achieved:

$$
\begin{aligned}
\int_{\|\boldsymbol{\omega}\|_2 \leq \alpha} |P(\boldsymbol{\omega}) - Q(\boldsymbol{\omega})|^2 \, d\boldsymbol{\omega} &\leq \int_{\|\boldsymbol{\omega}\|_2 \leq \alpha} \frac{\left(24\epsilon + \max\{6, \frac{4}{k}\} \inf_{f \in \mathcal{F}} \mathsf{TV}(f^* * G, f * G)\right)^2}{\left(\inf_{\|\boldsymbol{\omega}'\|_2 \leq \alpha} |G(\boldsymbol{\omega}')|\right)^2} \, d\boldsymbol{\omega} \\
&= \frac{\left(24\epsilon + \max\{6, \frac{4}{k}\} \inf_{f \in \mathcal{F}} \mathsf{TV}(f^* * G, f * G)\right)^2}{\left(\inf_{\|\boldsymbol{\omega}\|_2 \leq \alpha} |G(\boldsymbol{\omega})|\right)^2} \mathsf{Vol}(\mathbb{B}_2^d(\alpha)) \\
&= \frac{\left(24\epsilon + \max\{6, \frac{4}{k}\} \inf_{f \in \mathcal{F}} \mathsf{TV}(f^* * G, f * G)\right)^2}{\inf_{\|\boldsymbol{\omega}\|_2 \leq \alpha} |G(\boldsymbol{\omega})|^2} \mathsf{Vol}(\mathbb{B}_2^d(\alpha)) \\
&= \frac{\left(24\epsilon + \max\{6, \frac{4}{k}\} \inf_{f \in \mathcal{F}} \mathsf{TV}(f^* * G, f * G)\right)^2}{B_G(\alpha)}.
\end{aligned}
\tag{109}
$$

where Vol is the $d$-dimensional volume (i.e., Lebesgue measure) of a set. Also, note that $\mathsf{Vol}(\mathbb{B}_2^d(\alpha))$ can be written as $\mathsf{Vol}(\mathbb{B}_2^d(1))\alpha^d$. What remains is to use Assumption 2.7. Based on the definition of $\mathsf{P}(\mathcal{F})$, for any pair $(\alpha, \xi) \in \mathsf{P}(\mathcal{F})$ (where $\alpha \geq 0$ and $\xi < 1$), we have

$$
\int_{\|\boldsymbol{\omega}\|_2 \geq \alpha} |P(\boldsymbol{\omega}) - Q(\boldsymbol{\omega})|^2 \, d\boldsymbol{\omega} \leq \xi \|f^* - \widehat{f}\|_2^2.
$$

Therefore, the following set of relations hold:

$$
\begin{aligned}
\|\widehat{f} - f^*\|_2^2 &= \int_{\mathbb{R}^d} \left| f^*(\boldsymbol{t}) - \widehat{f}(\boldsymbol{t}) \right|^2 \, d\boldsymbol{t} \\
&= \int_{\mathbb{R}^d} |P(\boldsymbol{\omega}) - Q(\boldsymbol{\omega})|^2 \, d\boldsymbol{\omega} \\
&= \int_{\|\boldsymbol{\omega}\|_2 \leq \alpha} |P(\boldsymbol{\omega}) - Q(\boldsymbol{\omega})|^2 \, d\boldsymbol{\omega} + \int_{\|\boldsymbol{\omega}\|_2 \geq \alpha} |P(\boldsymbol{\omega}) - Q(\boldsymbol{\omega})|^2 \, d\boldsymbol{\omega} \\
&\overset{(i)}{\leq} \frac{\left(24\epsilon + \max\{6, \frac{4}{k}\} \inf_{f \in \mathcal{F}} \mathsf{TV}(f^* * G, f * G)\right)^2}{B_G(\alpha)} + \int_{\|\boldsymbol{\omega}\|_2 \geq \alpha} |P(\boldsymbol{\omega}) - Q(\boldsymbol{\omega})|^2 \, d\boldsymbol{\omega} \\
&\overset{(ii)}{\leq} \frac{\left(24\epsilon + \max\{6, \frac{4}{k}\} \inf_{f \in \mathcal{F}} \mathsf{TV}(f^* * G, f * G)\right)^2}{B_G(\alpha)} + \xi \|\widehat{f} - f^*\|_2^2
\end{aligned}
\tag{110}
$$

where (i) holds due to (109), and (ii) holds owing to Assumption 2.7. Note that $\xi$ can itself be a function of $\|\widehat{f} - f^*\|_2$. This argument proves the bound. $\qquad\square$

Using Lemma D.5 and some simple algebra, it has been shown that we can guarantee the existence of a deterministic algorithm that by using $n$ i.i.d. samples from $f^* * G$, outputs $\widehat{f}$ which holds in the following inequality:

$$
\|\widehat{f} - f^*\|_2 \leq \frac{24\epsilon + \max\{6, \frac{4}{k}\} \inf_{f \in \mathcal{F}} \mathsf{TV}(f^* * G, f * G)}{\sqrt{B_G(\alpha)(1 - \xi)}},
\tag{111}
$$

with probability at least $1 - \delta$, uniformly over all pairs $(\alpha, \xi) \in \mathsf{P}(\mathcal{F})$. Therefore, the bound also holds for the infimum, i.e.,

$$\mathbb{P}\left(\|\widehat{f} - f^*\|_2 \leq \inf_{(\alpha,\xi) \in \mathsf{P}(\mathcal{F})} \frac{24\epsilon + \max\{6, \frac{4}{k}\} \inf_{f \in \mathcal{F}} \mathsf{TV}(f^* * G, f * G)}{\sqrt{B_G(\alpha)(1 - \xi)}}\right) \geq 1 - \delta. \tag{112}$$

In case $\xi$ is not a constant, and instead is a function of $\|\widehat{f} - f^*\|_2$, the bound turns into the following probabilistic inequality:

$$\mathbb{P}\left(\|\widehat{f} - f^*\|_2 \sqrt{1 - \xi\left(\|\widehat{f} - f^*\|_2\right)} \leq \frac{24\epsilon + \max\{6, \frac{4}{k}\} \inf_{f \in \mathcal{F}} \mathsf{TV}(f^* * G, f * G)}{\sqrt{B_G(\alpha)}} \,\middle|\, \forall (\alpha, \xi) \in \mathsf{P}(\mathcal{F})\right) \geq 1 - \delta, \tag{113}$$

which completes the proof. $\qquad\square$

## E. Proof of Section 3: Theoretical Example

*Proof of Proposition 3.2.* Based on Theorem 2.10 and Corollary 2.11, in this problem setting, we can guarantee that there exists an algorithm such that upon having perturbed samples from $f^*$, it outputs $\widehat{f}$ that $\|\widehat{f} - f^*\|_2^2 \leq \frac{24\epsilon}{T}\left(\pi c k\sqrt{d}\right)^{d/2}\sqrt{\mathsf{Vol}(\mathbb{B}_2^d(1))}$. The required number of samples for this purpose will be

$$\begin{aligned}
n \geq{}& \mathcal{O}\left(\frac{288dk}{\epsilon}\log\frac{2}{\delta}\log\frac{6k}{\epsilon}\log(3d)\log_3\left(\frac{4}{\delta}\right)\right) \\
&+ \mathcal{O}\left(\frac{8}{9\epsilon^2}\left[\left(2kd + k\log_2\frac{4k}{\epsilon}\right)\log\left(\frac{288dk}{\epsilon}\log\frac{2}{\delta}\log\frac{6k}{\epsilon}\log(3d)\log_3\left(\frac{4}{\delta}\right)\right) + \log\left(\frac{12}{\delta}\right)\right]\right) \\
&+ \mathcal{O}\left[\frac{4d^2k}{9\epsilon^2}\log_2\left(\frac{rd\sqrt{2k}}{2\epsilon}\middle|\Phi_{\mathcal{N}(\mathbf{0},\sigma^2 \mathbf{I}_d)}^{-1}\left(\frac{\delta\epsilon}{2304kd^2\log\left(\frac{4}{\delta}\right)\log\left(\frac{2}{\delta}\right)\log\left(\frac{6k}{\epsilon}\right)\log(3d)}\right)\middle|\right)\right] \\
&\times \log\left(\frac{288dk}{\epsilon}\log\left(\frac{2}{\delta}\right)\log\left(\frac{6k}{\epsilon}\right)\log(3d)\log_3\left(\frac{4}{\delta}\right)\right).
\end{aligned}$$

In the gaussian noise case, $B_G(\alpha) = e^{-(\sigma\alpha)^2}$, therefore by having

$$\begin{aligned}
n \geq{}& N_{\tau,t,m}^{\mathsf{Clean}}(6\epsilon, \delta/2) + \\
& \mathcal{O}\left[\frac{4d^2k}{9\epsilon^2}\log_2\left(\frac{rd\sqrt{2k}}{2\epsilon}\middle|\Phi_{\mathcal{N}(\mathbf{0},\sigma^2 \mathbf{I}_d)}^{-1}\left(\frac{\delta\epsilon}{2304kd^2\log\left(\frac{4}{\delta}\right)\log\left(\frac{2}{\delta}\right)\log\left(\frac{6k}{\epsilon}\right)\log(3d)}\right)\middle|\right)\right] \\
&\times \log\left(\frac{288dk}{\epsilon}\log\left(\frac{2}{\delta}\right)\log\left(\frac{6k}{\epsilon}\right)\log(3d)\log_3\left(\frac{4}{\delta}\right)\right),
\end{aligned}$$

which can be subsequently simplified as

$$n \geq \mathcal{O}\left(\frac{d^2k}{\epsilon^2}\log^2\left(\frac{rdk\sigma}{\epsilon\delta}\right)\right) \tag{114}$$

noisy samples, there exists a deterministic algorithm that takes these perturbed samples as input, and outputs $\widehat{f} \in \mathcal{F}$ such that the following bound holds with probability at least $1 - \delta$:

$$\|\widehat{f} - f^*\|_2 \leq \epsilon \inf_{(\alpha,\xi) \in \mathsf{P}(\mathcal{F})} 24\sqrt{\frac{\mathsf{Vol}(\mathbb{B}_2^d(1))\alpha^d e^{(\sigma\alpha)^2}}{1 - \xi}} \tag{115}$$

where

$$\xi = 1 - \zeta^d\left(\frac{\alpha}{2ck\sqrt{d}}\left(T\|\widehat{f} - f^*\|_2\right)^{2/d}\right).$$

Hence, we can conclude the following bound holds for small $\frac{\alpha}{2ck\sqrt{d}}\left(T\|\widehat{f} - f^*\|_2\right)^{2/d}$ (due to the fact that $\zeta(h) = \frac{2}{\pi}h$ for small $h$):

$$\|\widehat{f} - f^*\|_2^2 \le \epsilon \inf_{\alpha > 0} \frac{24}{T} \sqrt{\mathsf{Vol}(\mathbb{B}_2^d(1))\alpha^d \left(\pi ck\sqrt{d}\right)^d \frac{e^{(\sigma\alpha)^2}}{\alpha^d}} = \frac{24\epsilon}{T}\left(\pi ck\sqrt{d}\right)^{d/2} \sqrt{\mathsf{Vol}(\mathbb{B}_2^d(1)) \inf_{\alpha > 0} e^{(\sigma\alpha)^2}}$$

due to the fact that $\inf_{\alpha > 0}$ is equal to 1, this bound will become:

$$\|\widehat{f} - f^*\|_2^2 \le \frac{24\epsilon}{T}\left(\pi ck\sqrt{d}\right)^{d/2} \sqrt{\mathsf{Vol}(\mathbb{B}_2^d(1))} \tag{116}$$

Looking back as (114), we can upper-bound $\epsilon$ based on $n$ (and other parameters $k, d, \delta$ and $\sigma$) as follows:

$$\epsilon \le \mathcal{O}\left(\sqrt{\frac{d^2 k}{n}} \log\left(\frac{n\sigma}{d^2 k\delta}\right)\right). \tag{117}$$

Plugging (117) into (116), we get

$$\|\widehat{f} - f^*\|_2 \le \mathcal{O}\left(\frac{k^{(d+1)/4}}{n^{1/4}} \sqrt{\frac{d^{d/4+1}}{T}} \log\left(\frac{n\sigma}{d^2 k\delta}\right)\right). \tag{118}$$

The approximation of $\zeta(h) \simeq \frac{2}{\pi}h$ has not caused issues, based on the fact that $\inf$ holds in the $\lim_{\alpha \to 0}$, which makes $\frac{\alpha}{2ck\sqrt{d}}\left(T\|\widehat{f} - f^*\|_2\right)^{2/d}$ tend to zero. Therefore, the approximation that we used for $\zeta(\cdot)$ holds, which completes the proof. $\square$

