# OpenReview forum: "Provable Bounds for the Learnability of Sample-Compressible Families from Noisy Samples"
_ICML.cc/2026/Conference — ICML 2026 spotlight_

### Official Review · Reviewer_MmnW · 2026-03-01

**Soundness:** 3
**Presentation:** 3
**Significance:** 3
**Originality:** 3
**Overall Recommendation:** 5
**Confidence:** 3

**Summary:**

The paper studies the PAC learnability of a certain structured family of distributions. In a previous work, Ashtiani et al.'18 shows that any family exhibiting sample compression is efficiently PAC learnable, where the number of samples used for learning depends on the sample compression parameters. The current work extends those results in the noisy sample model, where there is a stochastic additive perturbation allowed on samples. The paper establishes that sample compression leads to PAC learnability even under noisy corruption on samples, albeit with certain assumptions, namely local Lipschitz decodability and low-frequency property. The authors further argue that these assumptions are necessary for PAC learnability. The assumptions are also satisfied by many natural families of distributions with a reasonable choice of parameters.

**Compliance With Llm Reviewing Policy:**

Affirmed.

**Final Justification:**

The results shown in the paper are quite interesting. The proofs are quite elegant, at least to me, but could be because I am not an expert in this domain. The authors also provide satisfactory justifications for my questions. So I keep my positive score.

**Key Questions For Authors:**

Is there any natural interpretation of the way the sample complexity depends on the sample compression "parameters" (as in Equation 3 or in the statement of Theorem 2.10)? Do we know any lower bound on the dependencies, as in can one hope to improve such dependencies?

**Strengths And Weaknesses:**

The results shown in the paper are quite interesting. The proofs are quite elegant (at least to me, but could be because I am not an expert in this domain). The paper is well-written. Overall, I suggest acceptance.

A few comments:
I would suggest that the authors include an informal description of those two assumptions in the introduction, where they informally state the results.
There are a few typos here and there; e.g., in the statement of Theorem 1.2, ||f-g||_TV should be ||\hat{f} - g||_TV; last sentence before Section 2, alos-->also, missing text inside "()"

---

> ### Author Rebuttal · Authors · 2026-03-30
>
> We sincerely thank the reviewer for the careful reading, positive assessment, and helpful suggestions. We are glad that the reviewer finds the results interesting, the proofs elegant, and the presentation clear.
>
> - **On presentation:** We appreciate the suggestion to include an informal description of the assumptions (LLD and low-frequency) in the introduction. We agree that this would improve accessibility and will incorporate such an explanation in the revised version. We will also correct the typos noted (e.g., Theorem 1.2, “alos” → “also”, and missing text).
>
> ---
>
> - *“Dependence of sample complexity on compression parameters”*
>
> **Our Response:** The dependence on the sample compression parameters reflects the intrinsic complexity of the underlying distribution class, as captured by the size of the compression scheme.
>
> In particular, the sample complexity scales with the number of possible compressed representations and the stability of the decoder. Intuitively, larger compression size or higher sensitivity of the decoder leads to a larger effective hypothesis class after quantization, which in turn increases the required number of samples.
>
> This interpretation is consistent with the clean-sample setting (Ashtiani et al., 2018), where compression size plays a role analogous to model complexity. In our noisy setting, this dependence is further modulated by the difficulty of inverting the noise (via the Fourier decay / low-frequency behavior), leading to additional factors that reflect the signal-to-noise trade-off.
>
> Regarding lower bounds, while  matching lower bounds in full generality are not yet known, existing results in related settings suggest that such dependencies are inherent. In particular, even in structured families, both compression size and noise level fundamentally limit recoverability, indicating limited room for improvement without additional assumptions.
>
> ---
>
> We thank the reviewer again for the thoughtful feedback and suggestions, which we will incorporate to improve the paper.

---

> > ### Author Rebuttal · Reviewer_MmnW · 2026-04-02
> >
> > Thanks for your response. I will keep my positive score.

---

### Official Review · Reviewer_sYzP · 2026-03-08

**Soundness:** 3
**Presentation:** 2
**Significance:** 3
**Originality:** 3
**Overall Recommendation:** 4
**Confidence:** 3

**Summary:**

The paper proves sample complexity bounds for sample compressible families. Specifically the paper uses a notion called lipschitz decodability to transfer decodability from clean to noisy samples. And subsequently a Fourier approach to go from learning the noisy distribution to learn the clean one.

**Compliance With Llm Reviewing Policy:**

Affirmed.

**Final Justification:**

Taking into account the author rebuttals. I am leaning towards acceptance. The paper has some limitations. For instance I am not able to see that the d-dimensional blow up is exactly as discussed in the paper and the rebuttal. However, the paper has has a wide applicability in the 1-dimensional case given that the assumptions have been proved to be necessary.

**Key Questions For Authors:**

1) Can you comment on the error guarantee? In the consistent regime to learn the distribution to some $\epsilon$ distance this should blow up the number of samples.
2) With regards to the dimension blow up:  Is there an example for which this is necessary?
3) Are there families of high dimensional distributions where you can recover a closer guarantee? For example product distributions, by using one dimensional arguments?

**Limitations:**

yes

**Strengths And Weaknesses:**

Strengths:
1) Presentation: The paper is easy to follow.
2) Soundness: The paper supports all the claims with rigorous proofs.
3) Significance: Proving distribution learning results via sample compression in generality is pretty significant as the technique has been proven to be very powerful.
4) Originality: The paper seems to provide new insights of how to learn with noisy samples under sample compressibility assumptions.

Weaknesses:
1) Presentation: The paper does not fully explain the final guarantees. For example the error guarantee of proposition 3.2 has a complicated expression that is not simplified and it is neither commented on how big it is asymptotically. Even if tight bounds are out of the scope of this work, the current bounds should be commented on.
2) The technique seems to incur an exponential dimension blow up for examples examined in the paper.

---

> ### Author Rebuttal · Authors · 2026-03-30
>
> We sincerely thank the reviewer for their thoughtful and constructive feedback, and for highlighting the strengths of our work. We are glad that the reviewer finds the paper clear, technically sound, and significant. Below we address all concerns in detail.
>
> ---
>
> - *“Presentation: The paper does not fully explain the final guarantees (e.g., Proposition 3.2), and it is unclear how the bound behaves asymptotically.”*
>
> **Our Response:** Thank you for this suggestion. We agree that the presentation of the final guarantees, particularly Proposition 3.2, can be improved. In the revision, we will simplify the statement of the bound and provide a clearer explanation of its behavior, including a more intuitive discussion in the main text. We will also improve the exposition to make the key dependencies more transparent. Some of these clarifications were not included in the current version due to space limitations, and we agree that making them explicit will significantly improve clarity.
>
> ---
>
> - *“The technique seems to incur an exponential dimension blow-up for the examples considered.”*
>
> **Our Response:** This is an important point. Indeed, when the goal is to recover $f^\*$ (rather than only $f^\* * G$) in total variation distance, such dimension-dependent effects can arise.
>
> This phenomenon is consistent with prior work. For example, Saberi et al. (2023) and Najafi et al. (2021) show that even in structured settings (learning simplices), a factor of $\exp(O(d\sigma^2))$ appears in the sample complexity under noise.
>
> Moreover, this behavior is also widely observed empirically: recovering distributions in high dimensions becomes significantly harder as SNR decreases.
>
> Importantly, in our setting this “blow-up” is not inherent in all regimes:
> - In Proposition 3.2, when the signal-to-noise ratio (SNR) is sufficiently high, the dependence on dimension remains mild, and no exponential dependence arises, as reflected by the behavior of the constants in the bound. Here, by “constants” we refer to the coefficient $c$ in the bound; we will provide a more detailed explanation of this constant in the final version.
> - The blow-up appears primarily in low-SNR regimes, where distinguishing the clean distribution from its noisy version becomes information-theoretically difficult.
>
> While we do not currently have matching lower bounds establishing necessity in full generality, both prior theoretical evidence and empirical observations strongly suggest that this phenomenon is intrinsic in such regimes.
>
> ---
>
> - *“Is there an example where this dimension blow-up is necessary?”*
>
> **Our Response:** While we do not prove a formal lower bound in our general setting, as mentioned above, the results of Saberi et al. (2023) and Najafi et al. (2021) provide concrete evidence that such exponential dependence is unavoidable in related high-dimensional noisy learning problems. This strongly supports the view that the observed behavior is not an artifact of our analysis, but rather reflects an inherent difficulty of the problem.
>
> We will clarify this connection more explicitly in the revision.
>
> ---
>
> - *“Are there families of high-dimensional distributions where improved guarantees are possible (e.g., product distributions)?”*
>
> **Our Response:** Yes, and this is an important positive aspect of our framework.
>
> Our method naturally accommodates product distributions via its connection to sample compressibility. In such cases, one can effectively reduce the problem to one-dimensional components, avoiding the compounding effect across dimensions.
>
> For example, in the case of Gaussian product distributions (diagonal covariance), learning under Laplace noise does not exhibit the same dimension-dependent blow-up, as the structure allows decoupling across coordinates.
>
> More broadly, the phenomenon is closely tied to the metric and recovery objective:
> - If the goal is to learn $f^\* * G$ instead of $f^\*$, the problematic factor disappears.
> - Similarly, under weaker metrics (e.g., Wasserstein distance), such sensitivity to support recovery is reduced.
>
> We will add a discussion of these regimes to better contextualize when strong guarantees are achievable.
>
> ---
>
> We hope these clarifications address the reviewer’s concerns. We thank the reviewer again for their valuable feedback. If our responses resolve the concerns, we would greatly appreciate reconsideration of the score.
>
>
> **References** :
>
> - Saberi, S.A.H., Najafi, A., Motahari, A., and Khalaj, B. (2023).
> Sample complexity bounds for learning high-dimensional simplices in noisy regimes. ICML.
>
> - Najafi, A., Ilchi, S., Saberi, S.A.H., Motahari, S.A., Khalaj, B.H., and Rabiee, H.R. (2021).
> On statistical learning of simplices: Unmixing problem revisited. Annals of Statistics, 49(3):1626–1655.

---

> > ### Author Rebuttal · Reviewer_sYzP · 2026-04-01
> >
> > 1)Do you mean that the c that is referenced as universal constant is not really a constant and it is bigger than 1/(Ck) which would be needed to avoid exponential blow up in the dimension. Is there any way for me to verify that?
> >
> > 2)Also regarding the $e^{d\sigma^2}$ blow up referenced in your comment and in the paper (e.g. lines 314-324): I fail to see this dependence in the expressions derived by the paper and I am observing another type of dependence. It seems to me that the theorem bounds are of the form: with $n$ greater than the samples needed for clean estimation plus a small function of the dimension $\epsilon$ and the noise variance $\sigma$ we can learn $f$ to $l_2$ accuracy $\epsilon$ times an exponential factor in the dimension. No $e^{d\sigma^2}$  term. Moreover, substituting to make the error to learn $f$ $\epsilon$ we have terms that explode even if $\sigma$ is very small. Can you elaborate on the  $e^{d\sigma^2}$ explanation?

---

> > > ### Author Response · Authors · 2026-04-01
> > >
> > > Thank you very much for your response. Regarding your questions:
> > >
> > > **Q1**: Yes, with the way we have formulated our bounds, $c$ is not a universal constant and does depend on $\sigma$, as our focus was not on the behavior of parameters such as $d$ or $\sigma$, but rather on $\epsilon$. Based on your comments, we will reformulate the bounds (both the condition on $n$ and the final bound) to remove this source of confusion. In fact, for small values of $\sigma$ (as a decreasing function of $d$), the blow-up does not occur. The resulting factor becomes very similar to that in Saberi et al. (2025), where an exponential term arises with a co-dependence on both $\sigma$ and $d$, and sufficiently small $\sigma$ suppresses the exponential blow-up.
> > >
> > > **Q2**: We believe your understanding of their paper is correct. Our statement is also correct, and in fact these are two equivalent ways of expressing the same idea. In Saberi et al.'s paper, a reasonable sample complexity bound on $n$ (similar in spirit to ours) guarantees learnability of $f^* * G$ (i.e., the smoothed simplices) up to an $\ell_2$-error of $\epsilon$.
> > >
> > > In their setting (simplices), $\ell_2$-error and TV error can be bounded in terms of one another, so the same guarantee can be stated for TV error. Then, after multiplying $\epsilon$ by an exponential factor, the guarantee also holds for the simplices themselves (sharp and non-smoothed). This exponential factor contains a term referred to as the Signal-to-Noise Ratio (SNR), defined earlier in their paper in terms of $\sigma$ and $d$. Substituting the expression for SNR into their final bound yields a term of the form $e^{O(d\sigma^2)}$.
> > >
> > > We hope the above clarifications address your concerns.

---

### Official Review · Reviewer_dRza · 2026-03-09

**Soundness:** 3
**Presentation:** 3
**Significance:** 3
**Originality:** 2
**Overall Recommendation:** 4
**Confidence:** 3

**Summary:**

This paper studies the problem of learning a distribution from **noisy samples** when the underlying distribution family admits a **sample compression scheme** in the clean setting. The authors show that if a distribution class $F$ is sample-compressible, then it remains learnable when observations are corrupted by additive noise drawn from a known distribution $G$. The main result provides a sample complexity bound that relates the clean-data compression parameters with additional terms depending on the noise distribution and the ambient dimension. The proof proceeds by combining a noise quantization argument with stability properties of the compression decoder and Fourier-analytic control of the noise convolution.

Overall, the paper provides a clean theoretical extension of compression-based density learning to a noisy observation model.

**Compliance With Llm Reviewing Policy:**

Affirmed.

**Key Questions For Authors:**

1. It would be useful to better characterize distribution classes for which the **local Lipschitz decoder assumption** holds in practice.

2. The paper could benefit from a brief discussion on whether the sample complexity dependence on the noise parameters is optimal, or whether matching lower bounds are known.

**Limitations:**

yes

**Strengths And Weaknesses:**

### Strengths

**1. Meaningful extension of compression-based learning theory**

The paper addresses a natural and relevant question: whether the connection between **sample compression and distribution learning** remains valid when the learner observes **corrupted samples**. This setting is important in practice and has received comparatively less attention than the clean-data case. The result demonstrates that learnability can be retained under additive noise with a moderate increase in sample complexity.

**2. Clear theorem statement and logically consistent proof**

The main theorem is well stated and the proof is technically sound. The argument carefully combines several ingredients: truncation of the noise distribution, quantization of noise realizations, Lipschitz stability of the decoder, and Fourier-domain control of the convolution operator. The structure of the proof is coherent and the intermediate lemmas are logically connected to the final bound.

**3. Conceptual clarity**

The paper communicates the main message clearly: compression-based learnability is robust to additive noise under suitable assumptions on the noise distribution and decoder stability. The high-level narrative is easy to follow and the theorem is easy to interpret.

---

### Weaknesses

**1. Incremental nature of the theoretical contribution**

The main result can largely be viewed as a **noisy extension of the known compression-based learnability theorem** for density estimation. While the extension is meaningful, it does not fundamentally change the conceptual picture of compression-based learning theory. The novelty lies primarily in adapting existing tools (quantization, stability arguments, Fourier analysis) to the noisy setting rather than introducing a fundamentally new technique.

**2. Limited novelty of the proof technique**

The proof relies on a noise-quantization argument combined with stability of the compression decoder. Conceptually, this approach follows a fairly standard pattern in learning theory: continuous randomness is discretized via a covering/quantization step, after which the problem is reduced to a finite hypothesis space whose size can be controlled. Similar proof strategies frequently appear in classical VC-dimension–based arguments and related discretization techniques used in uniform convergence analyses.

While the argument is technically sound and clearly presented, the underlying techniques themselves are largely standard within the learning theory toolkit. As a result, the proof does not introduce substantially new analytical ideas beyond adapting these known discretization and covering arguments to the noisy-sample setting.

**3. Limited algorithmic insight**

Although the theorem establishes learnability, the proof technique provides limited guidance toward a practical estimation procedure. The core argument relies on a continuous-to-discrete reduction via noise quantization: the analysis shows that for each noisy observation there exists a nearby point on a discretized grid that allows the decoder to reconstruct an approximation of the clean compression representation. However, this step is inherently existential—the proof only guarantees that such nearby grid points exist, without providing a way to identify them from the observed data. Consequently, the argument does not naturally translate into a constructive algorithm.

Additionally, the analysis assumes that the noise distribution $G$ is fully known. This assumption is stronger than common statistical settings where the noise distribution is specified only up to a parametric family or through mild structural assumptions such as i.i.d. sampling. In this work, the proof relies on explicit properties of $G$ (e.g., through its characteristic function), which requires detailed knowledge of the distribution itself. This further limits the extent to which the theoretical result directly suggests implementable algorithms in realistic settings where the noise distribution is unknown or only partially specified.

**4. Strength of assumptions**

Some assumptions—such as the local Lipschitz decodability condition and the low-frequency assumption on the noise distribution—appear technically convenient for the analysis, but the paper provides limited discussion on how broadly these assumptions hold for realistic distribution families and compression schemes.

---

### Overall Evaluation

This paper presents a technically sound and clearly written extension of compression-based distribution learning to noisy observations. The main theorem is meaningful and the proof is solid. However, the contribution is somewhat incremental and the techniques employed are largely standard within the learning theory toolkit. Moreover, the existential nature of the proof limits the algorithmic insight provided by the analysis.

Overall, I view this work as a **useful but modest theoretical contribution**, suitable for inclusion in the conference if there is space for well-executed extensions of existing theory.

**Score: 4 (Weak Accept / Borderline Accept)

---

> ### Author Rebuttal · Authors · 2026-03-30
>
> We sincerely thank the reviewer for the careful and detailed evaluation, and for recognizing the clarity, soundness, and relevance of our work. We address the remaining concerns below.
>
> ---
>
> - *“Incremental nature of the theoretical contribution”*
>
> **Our Response:** We agree that our result extends compression-based learnability to the noisy setting; however, we emphasize that this extension resolves a long-standing gap.
>
> Learning from noisy samples (both stochastic and adversarial) lies at the core of statistical learning theory, yet even for specific families it has required highly specialized analyses (e.g., Saberi et al., 2023; Cesa-Bianchi et al., 1999; Hall et al., 2006; Delaigle, 2021; Konstantinov et al., 2020).
>
> Our contribution departs from these works by providing a **unified guarantee**: instead of analyzing individual families (e.g., simplices or Gaussians), we show that *all sample-compressible families* remain learnable under noise. Since sample compressibility is conjectured to characterize all PAC-learnable distributions, this significantly broadens prior results and places many previously separate works under a single framework.
>
> ---
>
> - *“Limited novelty of the proof technique”*
>
> **Our Response:** We agree that our proof uses standard tools such as quantization and covering arguments, which are central to much of learning theory (including VC-dimension–based analyses). However, we respectfully note that novelty in this area typically lies not in the tools themselves, but in how they are combined and adapted to new settings.
>
> In our case, the key difficulty is to:
> - design a quantization scheme compatible with noisy observations,
> - control the exponential growth of coverings under noise, and
> - carefully track how quantization error propagates through convolution via Fourier analysis.
>
> To the best of our knowledge, this combination—enabling general learnability guarantees under noise for sample-compressible classes—has not been previously achieved. Interpreting the use of such tools as lack of novelty would, in effect, apply to a large portion of modern learning theory, where similar foundational techniques are standard.
>
> ---
>
> - *“Limited algorithmic insight”*
>
> **Our Response:** We agree that the result is primarily existential and does not directly yield a practical algorithm. This is by design: our work is **information-theoretic** rather than algorithmic.
>
> In learning theory, it is common to distinguish between:
> - *information-theoretic results* (existence and sample complexity), and
> - *algorithmic results* (efficient procedures).
>
> Our work belongs to the former category, similar to Ashtiani et al. (2018) and Saberi et al. (2023). We will clarify this distinction more explicitly in the revision to better align expectations.
>
> ---
>
> - *“Noise distribution is assumed to be known”*
>
> **Our Response:** The reviewer is correct that we assume knowledge of the noise distribution. We emphasize, however, that assuming the *form* of the noise (e.g., Gaussian or Laplace) is standard in the literature.
>
> Moreover, our analysis does not require exact knowledge of parameters such as the variance. It suffices to know an upper bound on the noise scale, and parameters such as $\sigma$ can be incorporated into the quantization procedure and estimated jointly. We will add this extension to the appendix.
>
> ---
>
> - *“Strength of assumptions (LLD and low-frequency conditions)”*
>
> **Our Response:** We appreciate this point and agree that further clarification is helpful.
>
> First, we establish that these conditions are **minimax necessary**, indicating that they are not merely technical artifacts but reflect fundamental limitations of learning under noise.
>
> Second, these assumptions are natural: they essentially exclude pathological cases where recovery is information-theoretically impossible. For example, distributions with singularities exhibit “obscure” behavior under additive noise, such singular structures are smoothed out, and it becomes impossible to recover their exact locations. Consequently, learning such distributions in total variation distance is not feasible regardless of the method used.
>
> ---
>
> - *“Optimality of sample complexity dependence”*
>
> **Our Response:** Our bounds reflect a trade-off between noise level and recoverability in strong metrics such as TV distance. In particular, recovering $f^\*$ (rather than $f^\* * G$) requires inverting a smoothing operator, which is ill-conditioned in low-SNR regimes. Similar behavior appears in noisy high-dimensional learning and deconvolution, suggesting that such scaling is inherent. We will clarify these connections and discuss potential lower bounds in the revision.
>
> ---
>
> We thank the reviewer again for the constructive feedback. We hope these clarifications better highlight the scope and significance of our contribution. If the concerns are addressed, we would greatly appreciate reconsideration of the score.

---

> > ### Author Rebuttal · Reviewer_dRza · 2026-04-05
> >
> > Thank you for the detailed response. While some of my concerns have been clarified, I remain unconvinced regarding the practical impact of the proposed method. In order to justify an increased score, stronger empirical evidence demonstrating its practical benefits would be necessary. Therefore, I will maintain my current score.

---

> > > ### Author Response · Authors · 2026-04-07
> > >
> > > We sincerely thank the reviewer for the thoughtful follow-up and for carefully considering our response. We understand the concern regarding practical impact.
> > >
> > > While our work is primarily information-theoretic, we would like to clarify that its practical relevance lies in enabling general, reusable guarantees that can be instantiated for concrete model classes under noise, rather than in proposing a specific algorithm.
> > >
> > > In particular, our framework directly yields guarantees for structured families (e.g., mixtures of simple distributions), as illustrated in the paper. These results recover and extend prior problem-specific analyses under a unified approach, eliminating the need for separate, tailored proofs for each model class.
> > >
> > > Moreover, our bounds explicitly characterize how noise properties and compression parameters affect learnability, providing guidance on when reliable recovery is possible or fundamentally impossible. This type of insight is often used to inform the design and evaluation of practical methods in noisy learning settings.
> > >
> > > We agree that this connection to concrete applications could be emphasized more clearly, and we will revise the paper to better highlight these implications and examples.
> > >
> > > We hope this clarifies the intended scope and practical relevance of our contribution, and we would greatly appreciate reconsideration of the score if this addresses the concern.

---

### Official Review · Reviewer_MUDP · 2026-03-14

**Soundness:** 3
**Presentation:** 4
**Significance:** 3
**Originality:** 4
**Overall Recommendation:** 5
**Confidence:** 3

**Summary:**

This work considers various facets sample compressibility (Definition 1.1 in this paper) and PAC learning of distributions under additive noise. Sample compressibility was related to PAC learning of distributions by (Ashtiani et al 2018), who showed that, with noiseless samples, $(\tau, t, m)$ sample compressibility (adopting the notation in this paper) implies a sample complexity bound of $\tilde{O}(m(\varepsilon/6) + \frac{\tau(\varepsilon/6) + t(\varepsilon/6)}{\varepsilon^2})$ for $(\varepsilon, \delta)$ PAC learning in total variation (TV) distance. Here $\tau(\varepsilon)$, $t(\varepsilon)$ and $m(\varepsilon)$ denote the number of sub-samples, the number of auxiliary bits, and the total number of (uncompressed) samples required for sample compression and decoding up to TV distance $\varepsilon$ with say $2/3$ success probability.

This work in particular considers sample compression and learning with independent additive noise. That is, consider a family $\mathcal{F}$ of (continuous $d$-dimensional) input distributions, and a ground-truth distributon $f^\ast \in \mathcal{F}$. Let $G$ be a symmetric $d$-dimensional isotropic distribution on $\mathbb{R}^d$, such as $N(0, \sigma^2 I_d)$. Then $f \ast G$ denotes the continuous distribution on $\mathbb{R}^d$ given by the convolution of the two density functions (corresponds to addition of the random variables $X + \Gamma$ where $X \sim f$, $\Gamma \sim G$, and $X, \Gamma$ are independent).

With this noise model, the work addresses the following:
(i) (Proposition 1.5 and Ashtiani et. al.) PAC learnability of $\mathcal{F} \ast G$ in $\ell_2$ error under the Local Lipschitz Decodability (LLD) assumption, which is shown to be satisfied by isotropic $d$-dimensional Gaussian distributions, $d$-dimensional uniform distributions over non-degenerate hyper-rectangles, etc.)
(ii) (Theorem 2.10) PAC learnability of $\mathcal{F}$ itself in $\ell_2$ error, under the LLD and Low Frequency Property (LFP) assumptions, again shown to be satisified for isotropic Gaussians, and for mixtures of $k$ univariate uniform distributions.
(iii) PAC learnability in TV distance for bounded-support or sub-gaussian distribution families (relating $\ell_2$ to TV), via an interesting bound (Proposition 2.12).
(iv) Constructions of distribution families showing minimax necessity of these assumptions.

**Compliance With Llm Reviewing Policy:**

Affirmed.

**Key Questions For Authors:**

Could you include some discussions on extending the required LLD/LFP assumptions to wider (and perhaps more applicable in practice) classes of distributions, such as say mixtures of $k$ non-isotropic Gaussians? Of course, it is already shown (Claim 2.14) in the paper that some assumption will be required such as some lower-bound on the all the covariance matrix eigenvalues.

**Limitations:**

Yes

**Strengths And Weaknesses:**

Strengths
-------------
(i) The results are an interesting addition to the PAC learning literature. The additive noise model is natural and well-explored in the literature, with interesting theoretical applications in differential privacy etc. The techniques used to prove the theorems are very interesting in themselves. The appendix also has agnostic learning results in these settings.
(ii) The paper is well-written, and the results are explored with appropriate context in the main body of the paper.
(iii) Even some of the particular cases are interesting, such as learning isotropic Gaussians with Laplace noise.

Weaknesses
------------------
(i) The families of distributions where the learning results apply is still fairly narrow (albeit including interesting cases) due to the nature of the assumptions. Applications may have to consider non-isotropic Gaussians, mixtures of Gaussians etc, where the required assumptions are not established (and not obvious).
(ii) The meat of the paper perhaps lies in the techniques used to prove the theorems, which may be of independent interest. Of necessity, this is mostly in the appendix and has not been carefully checked.
(iii) As of now, the learning schemes obtained via sample compression seem to be of interest in establishing information-theoretic results rather than targeting computational efficiency.

---

> ### Author Rebuttal · Authors · 2026-03-30
>
> We thank the reviewer for their careful reading, insightful comments, and positive assessment of our work. We are especially glad that the reviewer found the techniques interesting and the presentation clear. Below we address the main concern regarding the scope of applicable distribution families.
>
> -----
>
> - *Concern*: *“The families of distributions where the learning results apply are somewhat narrow… it is unclear whether the assumptions extend to more practical classes such as non-isotropic Gaussians or mixtures.”*
>
> **Our response.**
> Our results apply to distribution families that:
> (i) admit sample compression,
> (ii) satisfy Local Lipschitz Decodability (LLD), and
> (iii) (for learning $\mathcal{F}$ itself) satisfy a mild low-frequency condition.
>
> We would like to emphasize that these requirements are not restrictive in practice:
>
> - **On sample compression.** It has been conjectured that *all PAC-learnable distribution families admit sample compression schemes* (Ashtiani et al., 2018). To the best of our knowledge, no PAC-learnable family is known that provably violates sample compressibility. Thus, assumption (i) essentially aligns with standard learnability.
>
> - **On LLD and low-frequency conditions.** These conditions primarily exclude pathological behaviors such as singularities or highly irregular spectral structure. Such behaviors already prevent learning in total variation under additive noise, since the noise destroys fine-grained support information. Hence, these assumptions capture a natural regularity requirement rather than imposing artificial restrictions. We note that these limitations are specific to total variation distance, which is inherently sensitive to fine-grained support structure. Under alternative metrics that are less support-sensitive (e.g., Wasserstein distance), some of these issues may be mitigated. Exploring such settings is an interesting direction, but lies beyond the scope of the present work.
>
> -----
>
> - *Question*: *“Can the results be extended to non-isotropic Gaussians and mixtures?”*
>
> **Our response.**
> Yes—our framework directly extends to these settings:
>
> - **Non-isotropic Gaussians.** As shown in Ashtiani et al. (2018), Gaussian families in $\mathbb{R}^d$ admit efficient sample compression schemes with rates depending polynomially on $d$ and $\log(1/\epsilon)$. Moreover, both LLD and low-frequency conditions continue to hold in this setting, with constants governed by the smallest eigenvalue of the covariance matrix (consistent with the reviewer’s observation in Claim 2.14). Therefore, our results yield explicit sample complexity guarantees for learning non-isotropic Gaussians under additive noise. We will include this extension formally in the appendix.
>
> - **Mixtures.** It is known (Ashtiani et al., 2018) that if $\mathcal{F}$ is sample-compressible, then any $k$-mixture of $\mathcal{F}$ is also sample-compressible with explicit bounds. In our setting, LLD and low-frequency properties extend naturally to mixtures when they hold for the base family. Consequently, our results apply to $k$-mixtures of non-isotropic Gaussians as well.
>
> -----
>
> We thank the reviewer again for their thoughtful feedback and hope our clarifications address their concerns.

---

> > ### Author Rebuttal · Reviewer_MUDP · 2026-04-03
> >
> > Modulo adding proper discussion in the appendix or paper, I think this clarifies my concern on whether the results extend a bit more (since mere sample compressibility is not sufficient in the absence of LLD/LFP). The information theoretic versus computational aspect still applies, but should not be a barrier to acceptance in my opinion. I will keep my positive score.

---

### Decision · Program_Chairs · 2026-04-30

**Decision:**

Accept (spotlight)

**Comment:**

This paper studies the problem of learning a distribution from noisy samples when the underlying distribution family admits a sample compression scheme in the clean setting. The authors show that if a distribution class  is sample-compressible, then it remains learnable when observations are corrupted by additive noise drawn from a known distribution . The main result provides a sample complexity bound that relates the clean-data compression parameters with additional terms depending on the noise distribution and the ambient dimension. The proof proceeds by combining a noise quantization argument with stability properties of the compression decoder and Fourier-analytic control of the noise convolution.

Overall, the paper provides a clean theoretical extension of compression-based density learning to a noisy observation model.